



# Optimization and sophistication of the super-droplet method for ultrahigh resolution cloud simulations

Toshiki Matsushima[1], Seiya Nishizawa[2], and Shin-ichiro Shima[3,2]

[1]Kobe University, Center for Planetary Science
[2]RIKEN Center for Computational Science
[3]University of Hyogo, Graduate School of Information Science

**Correspondence:** Toshiki Matsushima (toshiki@gfd-dennou.org)

**Abstract.** A particle-based cloud model was developed for ultrahigh-resolution numerical simulation of warm clouds. Simplified cloud microphysics schemes have already made meter-scale numerical experiments feasible; however, such schemes are based on empirical assumptions, and hence, they contain huge uncertainties. The super-droplet method (SDM) is promising for cloud microphysical process modeling; it is based on a particle-based approach and does not make any assumptions for the droplet size distributions. However, meter-scale numerical experiments using the SDM are not feasible even on the existing high-end supercomputers because of its high computational cost. In the present study, we optimized and sophisticated the SDM for ultrahigh resolution simulations. The contributions of our work are as follows: (1) The uniform sampling method is not suitable when dealing with a large number of super-droplets (SDs). Hence, we developed a new initialization method for sampling SDs from a real droplet population. These SDs can be used for simulating spatial resolutions between centimeter and meter scales. (2) We improved the SDM algorithm to achieve high performance by reducing data movement and simplifying loop bodies by applying the concept of effective resolution. The improved algorithms can be applied to Fujitsu A64FX processor, and most of them are also effective on other many-core CPUs and graphics processing units (GPUs). Warm bubble experiments revealed that the particle-steps per time for the improved algorithms is 57.6 times faster than those for the original SDM. In the case of shallow cumuli, the simulation times when using the new SDM with 64–128 SDs per cell are shorter than those for a bin method with 32 bins and are comparable to those for a two-moment bulk method. (3) Using supercomputer Fugaku, we demonstrated that a numerical experiment with $2\,\mathrm{m}$ resolution and 128 SDs per cell covering $13{,}824^2 \times 3{,}072\,\mathrm{m}^3$ domain is possible. The number of grids and SDs are 104 and 442 times, respectively, those of the current state-of-the-art experiment. Our numerical model exhibited perfect weak scaling up to 36,864 nodes, which account for 23% of the total system. The simulation achieves $7.97\,\mathrm{PFLOPS}$, $7.04\%$ of peak ratio for overall performance, and the simulation time for SDM is $2.86 \times 10^{13}$ particle·steps/s. Several challenges, such as optimization for mixed-phase clouds, inclusion of terrain, and long-time integrations, still remain, and our study will also contribute toward solving them. The developed model enables us to study turbulence and microphysics processes over a wide range of scales using combinations of DNS, laboratory experiments, and field studies. We believe that our approach advances the scientific understanding of clouds and contributes to reducing the uncertainties of weather simulation and climate projection.





# 1 Introduction

Shallow clouds greatly affect the Earth's energy budget, and they are one of the essential sources of uncertainty in weather prediction and climate projection (Stevens et al., 2005). Since various processes affect the behavior of clouds, understanding the individual processes and their interactions is critical. In particular, cloud droplets interact with turbulence over a wide range of scales (Bodenschatz et al., 2010) in phenomena such as entrainment and mixing and enhancement of the collisional growth of droplets. Hence, numerical modeling of these processes and model evaluation toward the quantification and reduction of uncertainty are challenges in the fields of weather and climate science.

Meanwhile, accurate numerical simulations of stratocumulus clouds are difficult because of the presence of a sharp inversion layer on the scale of several meters. Mellado et al. (2018) suggests that combining the direct numerical simulation (DNS) approach, which sets the eddy viscosity constant, and large-eddy simulation (LES) approach can accelerate research on related processes. Following their approach, Schulz and Mellado (2018) investigated the interactions between mean vertical wind shear and in-cloud turbulence driven by evaporative and radiative cooling, and Akinlabi et al. (2019) estimated turbulent kinetic energy. However, since they used saturation adjustment for calculating clouds, their results do not include the influence of detailed microphysics processes and their interactions with entrainment-mixing and supersaturation fluctuations (Cooper, 1989), which in turn affect the radiation properties.

To incorporate the details of cloud processes into such simulations, it is essential to remove the empirical assumptions on the droplet size distributions (DSD) rather than using a bulk cloud microphysics scheme. We should use a sophisticated microphysical scheme such as a bin method and a particle-based Lagrangian cloud microphysical scheme (Shima et al., 2009). If ultrahigh resolution simulations could be performed using a sophisticated microphysical scheme in large domains, we could use a DNS-based approach (Mellado et al., 2018) and compare these simulations with small-scale numerical studies (Grabowski and Wang, 2013) and observational studies on a laboratory scale (Chang et al., 2016; Shaw et al., 2020) to field measurement (Brenguier et al., 2011) scales. Such simulations may help understand the origins of the uncertainty in the clouds and their interactions with related processes. However, in reality, only relatively low-resolution simulations are possible using sophisticated microphysical schemes owing to their high computational cost. For example, Shima et al. (2020) recently extended the SDM to predict the morphology of ice particles and reported that the computational resources of the mixed-phase SDM are 30 times that of the two-moment bulk method of Seiki and Nakajima (2014). To the best of the author's knowledge, the previous studies by Sato et al. (2017, 2018) are possibly the state-of-the-art numerical experiments on the largest computational scale as yet. To investigate the sensitivity of nonprecipitating shallow cumulus to spatial resolution, they performed numerical experiments with spatial resolutions up to $6.25\,\mathrm{m}/5\,\mathrm{m}$ (horizontal/vertical) with 30 super-droplets (SDs) per cell using supercomputer K. They found that the highest spatial resolution used in their study is sufficient for achieving grid convergence of the cloud cover but not for the convergence of cloud microphysical properties. For solutions of the microphysical properties to converge with increasing spatial resolution, it is necessary to reduce the vertical grid length (Grabowski and Jarecka, 2015) for simulating the number of activated droplets accurately and to maintain the aspect ratio of the grid length close to 1 for turbulence statistics (Nishizawa et al., 2015).



Nevertheless, using a sophisticated microphysical scheme for high-resolution simulations remains a challenge. One approach
to cope with this difficulty is to await the development of faster computers. However, single-core CPU performance is not
increasing according to Moore's law anymore. Therefore, to take advantage of the state-of-the-art supercomputers, we must
adapt our numerical models to their hardware design. Another solution to overcome the challenge is to use the rapidly advancing
data scientific approaches. Seifert and Rasp (2020) developed a surrogate model of cloud microphysics from training data
using machine learning. Tong and Xue (2008) estimated the parameters of the conventional cloud microphysics models by data
assimilation to quantify and reduce parameter uncertainty. However, these methods cannot make predictions beyond outside of
the training data or enhance the representation power of the bulk cloud microphysics schemes. The Twomey-SDM proposed
by Grabowski et al. (2018) could be used to reduce the computational cost of a sophisticated model; in this SDM, only cloud
and rain droplet data are stored as SDs. However, the Twomey-SDM cannot incorporate the hysteresis effect of haze droplets
(Abade et al., 2018). Incorporating this effect is necessary for reproducing entrainment and detrainment when eddies cause the
same droplets to activate or deactivate in a short time at the cloud interface. In addition, since clouds localize at multiple levels
of hierarchy—from a single cloud to cloud clusters, appropriate load balancing is necessary for large-scale problems using
domain decomposition parallelization if the computational cost for cloud and rain droplets is high. Although such dynamic
load balancing is adopted in some plasma simulations (Nakashima et al., 2009) using the particle-in-cell method (PIC), which
solves a coupled system of particles in cells and variables at cells in the same way as the SDM, dynamic load balancing is
not a good option for weather and climate models. This is because such codes are complicated, and changes in dynamic load
balancing can affect the computational performance of other components.

In this study, we attempted ultrahigh-resolution cloud simulations with a sophisticated microphysical scheme via the opti-
mization and sophistication of the SDM. This approach is regarded a technical approach and has not been explored much though
it is a crucial approach. Our approach is based on the SDM, which is robust to the difficulties caused by dimensionality for more
complex problems, is free from the numerical broadening of the DSD and can be used even when the Smoluchowski equation
for collisional growth of droplets is invalid (see Grabowski et al. (2019); Morrison et al. (2020) and the references therein). We
focus on the optimization on Fujitsu A64FX processor, which is used in supercomputer Fugaku. We designed cache-efficient
codes and overcame the difficulties in achieving high performance for the PIC method based on the domain knowledge. To
achieve this goal, we reduced data movement and parallelization using single instruction / multiple data (SIMD) instructions
for most calculations.

In addition, there are two potentially important aspects of model improvement for ultrahigh-resolution experiments with the
SDM. One aspect is the initialization for the SDM. In the SDM, we need to sample representative droplets from many real
droplets to calculate the microphysical processes. Shima et al. (2020) used an importance sampling method to sample rare-
state SDs more frequently to improve the convergence of calculations of collision–coalescence. However, when we sample
many SDs for high-resolution experiments, the number of SDs may exceed the number of real droplets for rare-state SDs. The
second aspect is SD movement. In the SDM, the divergence at the position of SDs calculated from interpolated velocity should
be identical to divergence at the cell to guarantee consistency of SD number density and air density (Grabowski et al., 2018).
However, the interpolation used by Grabowski et al. (2018) only achieves first-order spatial accuracy, and the effect of vortical





and shear flows on the subgrid scale are not incorporated in SD movement. Using their scheme can introduce large errors in
particle mixing calculations.

The remainder of this paper is organized as follows. In Sect. 2, we describe and review the basic equations used in our
numerical model called the SCALE-SDM, target problem, and computers to be used. Section 3 describes the main contributions
in this study for the optimization and sophistication of the model. We first describe the model framework. Subsequently, we
describe the development of a new initialization method for the SDM and describe optimizations of each process in the SDM
(SD movement, activation/deactivation, and condensation/evaporation, collision and coalescence, sorting for SDs). In Sect. 4,
we evaluate the computational and physical performances of the new SCALE-SDM in two test cases. We also compare our
results with those obtained with the same numerical model using a two-moment bulk and bin methods and with those obtained
using the original SCALE-SDM. In Sect. 5, we evaluate the applicability of our model to large-scale problems by weak scaling
and discuss the detailed computational performance. Section 6 discusses the challenges for optimizing processes in mixed-
phase clouds, the inclusion of the terrain, and long-time integration. We also discuss the possibilities of achieving further high
performance in current and future computers. We summarize our main contributions in Sect. 7.

## 2    Overview of the problem

### 2.1    Governing equations

We use the fully-compressible nonhydrostatic equations as the governing equations for atmospheric flow. To simplify the
treatment of water substances in the SDM, only moist air (i.e., dry air and vapor; aerosol particles or cloud/rain droplets are
excluded) is considered in the basic equations for atmospheric flow. Although anelastic equations are often used to perform
high-resolution numerical experiments, we encountered several problems in using them in this study. First, when the domain is
decomposed using message passing interface (MPI), communications across all subdomains are unavoidable, and the network
bandwidth becomes a bottleneck for the calculations. Moreover, the time spent in collective communication increases with the
number of processes. For some cases, the network bandwidth bottleneck may be a minor problem even on large-scale computers
since the time step to integrate anelastic equations is not constrained by the high acoustic wave velocity. Nevertheless, since
we need more processes to compute wide domains in higher resolution and since we need to use the SDM, we cannot shorten
the time for collective communication easily. Second, the anelastic equations assume horizontally uniform mean fields and are
not appropriate for computing wider domains.

The basic equations are discretized using a finite volume method on the Arakawa-C grid. For solving time evolution of
dynamical variables and water-vapor mass mixing ratio, the advection terms are discretized by the fifth-order upwind difference
scheme (UD5). Since numerical diffusion is already included implicitly in UD5, we do not need explicit numerical diffusion to
stabilize the dynamical step calculations. Wang et al. (2009) encourages the use of the Flux Corrected Transport (FCT) scheme
to ensure monotonicity for mass and number mixing ratio of droplets to simulate aerosol–cloud interactions; however, we use
the FCT scheme to ensure only positive-definiteness for the water-vapor mass mixing ratio to obtain finer structures since it is
the only tracer in our case.





The time evolutions of dynamical variables during the $\Delta t$ interval are split into short time steps $\Delta t_{\mathrm{dyn}}$ associated with acoustic waves and longer time steps for tracer advection $\Delta t_{\mathrm{adv}}$ and physical processes $\Delta t_{\mathrm{phy}}$. The classical four-stage fourth-order Runge-Kutta method is used for short time steps, and the three-stage Runge-Kutta method (Wicker and Skamarock, 2002) is used for tracer advection. Unless otherwise noted, $\Delta t = \Delta t_{\mathrm{adv}} = \Delta t_{\mathrm{phy}}$. Changes in the dynamic variables caused by physical processes are calculated using tendencies, which are assumed constant during $\Delta t_{\mathrm{adv}}$.

The SDM is used as a cloud microphysics scheme. In this study, only warm cloud processes are considered: movement, activation/deactivation and condensation/evaporation, and collision–coalescence. In the SDM, each SD has a set of attributes that represent droplet characteristics. In this case, the data on SDs necessary to describe time evolution are the position in 3D space $\boldsymbol{x}$, droplet radius $R$, number of real droplets (which we refer to as multiplicity $\xi$), and the aerosol mass dissolved in a droplet $M$. The $i$th SD moves according to the wind and fall with terminal velocity:

$$\boldsymbol{v}_i = \boldsymbol{U}(\boldsymbol{x}_i) - v_i^\infty(\rho(\boldsymbol{x}_i), P(\boldsymbol{x}_i), T(\boldsymbol{x}_i), R_i)\boldsymbol{e}_z, \quad \frac{d\boldsymbol{x}_i}{dt} = \boldsymbol{v}_i, \tag{1}$$

where $\boldsymbol{U}$ is the air velocity at the position $\boldsymbol{x}$; $\rho$ is the air density; $P$ is the atmospheric pressure; $T$ is the temperature; $\boldsymbol{e}_z$ is the unit vector in vertical positive direction; $v^\infty$ is the terminal velocity; $\boldsymbol{v}$ is the velocity of the SD; $t$ is the time. The midpoint method is used for time integration to solve Eq. (1). We also need to specify a method for determining velocity $\boldsymbol{U}$ at the position of the SDs, which we will describe in Sect. 3.3.2.

Activation/deactivation and condensation/evaporation are represented by assuming that the SD radius $R$ evolves according to the Köhler theory:

$$R_i \frac{dR_i}{dt} = A(\boldsymbol{x}_i) \left[ S(\boldsymbol{x}_i) - 1 - \frac{a(T(\boldsymbol{x}_i))}{R_i} + \frac{b(M_i)}{R_i^3} \right], \tag{2}$$

where $S$ is the saturation ratio, and $A$ is a function of the temperature at the position, and it depends on the heat conductivity and vapor diffusivity. The terms $a/R$ and $b/R^3$ represent the curvature effect and the solute effect, respectively. See Shima et al. (2020) for the specific forms of $A$, $a$, and $b$ since they are not important here. A method to solve Eq. (2) is described in Sect. 3.3.3.

The collision–coalescence process is calculated using the algorithm proposed by Shima et al. (2009). If we consider all possible pairs of droplets to calculate collision–coalescence, the computational complexity is of order $O(N^2)$. However, their method considers only nonoverlapping pairs of droplets to reduce the computational complexity to the order of $O(N)$. Hence, the obtained coalescence probability is low; this parameter was corrected to make it consistent with the actual probability. Indeed, Unterstrasser et al. (2020) showed that the method proposed by Shima et al. (2009), which they referred to as the all-or-nothing algorithm with linear sampling, is suitable for problems when computational time is critical.

The Smagorinsky–Lilly type scheme with the stratification effect (Brown et al., 1994) is used as a turbulent scheme for LES. In the SDM, we do not incorporate the effect of turbulent fluctuations on movement, activation/deactivation and condensation/evaporation, and collision–coalescence. This is because the number of the additional attributes of the SD (three subgrid velocities and supersaturation fluctuation) is almost equal to the number of the original attributes, and the effect of supersaturation fluctuation on the spectral width of the DSD becomes relatively small when grid length is finer than $10\,\mathrm{m}$ (Grabowski





and Abade, 2017). As we will see later, the SDM is a memory-intensive application. In this case, the approach of increasing the number of computations while increasing spatial resolution is better than increasing the amount of memory to utilize supercomputers.

## 2.2 Target problem

**Table 1.** Comparison of model computational configurations among previous studies and this study. The last row shows the ratios of the parameters used in Sato et al. (2017) to those used in this study.

|  | # of grid points | # of SDs per cell | step (DYN) | step (MP) | grid length | # of nodes (system usage) |
|---|---|---|---|---|---|---|
| Mellado et al. (2018) | $5{,}120 \times 5{,}120 \times 1{,}280$ |  | 60,000 | 60,000 | 1.1 m |  |
| Sato et al. (2017) | $1{,}152 \times 1{,}024 \times 600$ | 30 | 450,000 | 45,000 | 6.25/5 m | 12,288 (14.8%) of the K |
| This study | $6{,}912 \times 6{,}912 \times 1{,}536$ | 128 | 782,609 | 48,913 | 2 m | 36,864 (23.2%) of the Fugaku |
| Ratio | 103.68 | 4.267 | 1.739 | 1.087 | $2.901^{-1}$ | 3 |

We describe the final target problem in this study and compare the problem size with that considered in Mellado et al.
(2018) and Sato et al. (2017); high-resolution numerical experiments on shallow clouds were performed in these studies. Mellado et al. (2018) used the numerical settings of the first research flight of the second Dynamics and Chemistry of Marine Stratocumulus field campaign (DYCOMS-II RF01) (Stevens et al., 2005) to simulate nocturnal stratocumulus. Sato et al. (2017) used the numerical settings of the Barbados Oceanographic and Meteorological Experiment (BOMEX) (Siebesma et al., 2003) to simulate shallow trade-wind cumuli. In this study, we simulated the BOMEX case but with much higher resolutions. The
main computational parameters of the two previous studies and our study are listed in Table 1. Here, the time steps for 1 h time integration are shown in the third and fourth columns.

Mellado et al. (2018) used anelastic equations with saturation adjustment for calculating clouds. They performed large-scale numerical experiments using a petascale supercomputer (Blue Gene/Q system supercomputer JUQUEEN at Jülich Supercomputing Centre). We also note that similar numerical experiments with a larger number of grid points ($5{,}120 \times 5{,}120 \times 2{,}048$) were
performed by Schulz and Mellado (2018) using the same supercomputer. Meanwhile, Sato et al. (2017) used fully-compressible equations with the SDM and performed high-resolution numerical experiments of BOMEX using petascale supercomputer K. The time steps for the dynamical process used in Sato et al. (2017) are one order of magnitude larger than those used in Mellado et al. (2018). In addition, because of the high computational cost of the SDM, Sato et al. (2017) used fewer grid points though they used 14.8% of the total system of supercomputer K. Unlike Sato et al. (2017), we performed ultrahigh resolution
numerical simulations of BOMEX with $\sqrt[3]{(6.25^2 \times 5)/2^3} \sim 2.901$ times higher resolution, 104 times more grid points, and 442 times more SDs, thereby using 23.8% of the total system of supercomputer Fugaku. The computational performance of this simulation will be described in detail in Sect. 5.2.





## 2.3 Target architecture

**Table 2.** Size and bandwidth of the cache and memory for Fujitsu A64FX processor.

| L1D cache | L1D cache BW | L2 cache(shared in CMG) | L2 cache BW | HBM2 | Memory BW |
|---|---|---|---|---|---|
| 64 KiB ×48 | 11 TB/s | 8 MiB ×4 | 3.6 TB/s | 32 GB | 1,024 GB/s |

In this study, we mainly used computers equipped with Fujitsu A64FX processors to evaluate the computational and phys-
ical performance of the new model, SCALE-SDM. In this section, we summarize the essential features and functions of the
computers.

A64FX is a CPU that adopts scalable vector extension (SVE), an extension of the Armv8.2-a instruction set architec-
ture. A64FX has 48 computing cores and two or four assistant cores. Each CPU has four NUMA nodes called the core
memory groups (CMGs). One core has an L1 cache of $64\,$KiB and can execute SVE-based 512-bit vector operations at
$2.0\,$GHz in the normal mode ($2.2\,$GHz in the boost mode) with two FMA units. Each CMG shares an L2 cache of $8\,$MiB
and has high bandwidth memory 2 (HBM2) of $32\,$GB (bandwidth of $256\,$GB/s). The theoretical peak performance per node
is $3.072\,$TFLOPS ($3.3792\,$TFLOPS for double precision (FP64)). Supercomputer Fugaku has 158,976 nodes with a 6D torus
shape $(X, Y, Z, a, b, c) = (24, 23, 24, 2, 3, 2)$, and the nodes are connected by Tofu Interconnect D. The cache and memory per-
formances, which are particularly important for this study, are summarized in Table 2. A64FX has the best power performance
among the supercomputers equipped with a many-core general-purpose CPU (Fugaku full system, $15.418\,$GFlops/W, Green500
2022/6) and has high memory bandwidth comparable to a GPU. In addition, SVE can execute not only FP64, single-precision
(FP32), and 32 bytes integer (INT32) calculations but also low-precision 16 bytes floating point number (FP16) and 16 bytes
integer (INT16) calculations.

Fugaku and FX1000 have a power management function called the Power Knob to improve the computational power per-
formance. Users can operate the Power API (Grant et al., 2016) to control the clock frequency (Normal mode: $2.0\,$GHz, Boost
mode: $2.2\,$GHz) and switch to eco-mode, which uses only one of the two floating-point pipelines.

Fugaku was designed to achieve 100 times the effective performance of K through hardware and application co-design.
The actual performance of Fugaku is 46 (50.6) times the peak performance and 30.7 times the memory bandwidth of K. In
addition, using FP32 or FP16, the amount of data calculated by single instruction and that transferred from memory doubles or
quadruples, respectively, and by optimizing a code according to its characteristics, users can potentially achieve a further two
or four times higher effective peak performance, respectively. Due to the high memory bandwidth of Fugaku, its byte per flops
ratio (B/F) is $0.33$ ($0.30$), which is not too small compared to that of the K (B/F=0.5). This is an advantage for applications in
which the memory bandwidth is crucial for performance.

Although this study describes optimizations for A64FX, most of them can be applied to many-core general-purpose CPUs
such as Intel Xeon equipped with x86-64 instruction set architecture. For such generalization, please see Sect. 3.3.1 with
the parameters in Table 2 replaced with those for the x86-64 architecture. However, optimization using accelerators such as





GPUs is beyond the scope of this study. However, since the applicability of this study to accelerators is necessary for future high-performance computing, we discuss some differences between CPU-based and GPU-based approaches.

To map CPU-based optimization to GPU-based optimization, the L1 cache of the CPU can be read as the register file
(for storing most frequently accessed data), L1 cache, and shared memory; OpenMP parallelization can be read as streaming multiprocessors parallelization for NVIDIA GPU (or Compute Unite for AMD GPU); MPI processes can be read as the number of GPUs. In addition, since the memory bandwidth of one node of A64FX is comparable to that of a single GPU (e.g., NVIDIA Tesla V100: $900$ GB/s, A100: $1,555$ GB/s), a comparison in terms of memory throughput is reasonable if we assume that all the SD information is on GPU memory. Although the approaches for cache and memory optimization of the CPU and
GPU are similar, those for calculation optimization may differ. For example, GPUs are not good for reduction calculations, such as calculating the liquid water content in a cell from the SDs in the cell. The current trend for supercomputers is to use heterogeneous systems comprising both CPUs and GPUs as they provide excellent price performance. Nevertheless, memory bandwidth is essential for weather and climate models, including the SDM. Thus, it is not easy to achieve higher performance unless the entire simulation can be handled only in GPUs.
The numerical model UWLCM (Arabas et al., 2015; Dziekan et al., 2019; Dziekan and Zmijewski, 2022) utilized GPUs for the SDM and CPUs for other processes, and Dziekan and Zmijewski (2022) achieved 10–120 times faster computations compared with CPU-only computations. Still, the time-to-solution using the SDM is 8 times longer than the bulk method. Although the CPU used had a lower bandwidth memory compared with the GPU for the dynamical core and the bulk method, we used a CPU with a higher bandwidth memory for all processes. This is an advantage when the entire simulation must be
accelerated essential to reduce the time-to-solution.

## 3 Numerical model

### 3.1 Model framework of SCALE

We used SCALE-RM (Scalable Computing for Advanced Library and Environment-Regional Model, Nishizawa et al., 2015; Sato et al., 2015) as the development platform. SCALE is a library that consists of multiple components rather than a numerical
model. Users can use it as a numerical model. It is also possible to compose unit tests and new components, and to combine them with the model easily. In addition, since only few dependencies exist among the modules, it facilitates data exchange between multiple grid systems.

We adopted the hybrid type of three- and two-dimensional (3D and 2D) domain decompositions using MPI. For 3D decomposition, we denoted the numbers of MPI processes for the $x, y$, and $z$ axes as $N_x, N_y$, and $N_z$, respectively. For 2D
decomposition, we decomposed the $x$ and $y$ axes into $N_x^{\mathrm{2D}}$ and $N_y^{\mathrm{2D}}$ domains, respectively. Here, we set $N_x^{\mathrm{2D}} = N_x \cdot N_{xl}$ and $N_y^{\mathrm{2D}} = N_y \cdot N_{yl}$ such that $N_z = N_{xl} \cdot N_{yl}$. Then, the total number of MPI processes $N$ is common, i.e., $N = N_x \cdot N_y \cdot N_z = N_x^{\mathrm{2D}} \cdot N_y^{\mathrm{2D}}$. These two types of domain decomposition were utilized depending on the type of computations. The 3D domain decomposition is suitable for dynamical processes because frequent neighborhood communications are required to integrate short time steps for acoustic waves; further, the amount of communication is less because of the small ratio of halos to the





inner grids. On the other hand, 2D domain decomposition is suitable for the SDM. As described later, since the number density of SDs is initialized proportional to the air density, the amount of computations varies vertically in a stratified atmosphere. In addition, the communication amount varies depending on whether clouds are within the domain. If 3D decomposition is used, domains without any cloud are likely, e.g., near the top and bottom boundaries; such domains may lead to a drastic load imbalance.

A drawback of the 3D domain decomposition is that it is more likely to suffer from network congestion; further, there will be hardware limitations on the number of simultaneous communications due to the increase in the number of processes in a neighborhood. The number of processes is 26 for 3D domain decomposition, while it is eight for 2D domain decomposition. In addition, the throughput of communication decreases for smaller message sizes. In this study, we eliminated all unnecessary communications from the diagonal 20 directions and pack communications for each neighborhood direction to the maximum

extent possible to gain high communication throughput. Communication time was overlapped with computation time during the dynamics process to reduce the time-to-solution.

## 3.2    Initialization of super-droplets

Although the SDM makes no prior assumptions on the DSD, the accuracy of the prediction depends on the initialization of the sampling of SDs from a vast number of real droplets. Shima et al. (2009) first used the constant multiplicity method, which

samples SDs from normalized aerosol distribution. Further, Arabas and Shima (2013), Sato et al. (2017), and Shima et al. (2020) used the uniform sampling method, in which SDs were sampled from a uniform distribution of the log of the aerosol dry radius to sample droplets that rare but important—for example, large droplets that may trigger rain. Indeed, Unterstrasser et al. (2017) showed that collision–coalescence calculations converge faster for a given number of SDs if the dynamic range of multiplicity is broader (i.e., the uniform sampling method), and it converges slower if the constant multiplicity method is

used. However, owing to the broad dynamic range of the uniform sampling method, some multiplicities obtained using this method may fall below 1 if too many SDs are used to increase the spatial resolution. In this case, since multiplicity is stored as an integer type, some SDs will be cast to 0, and the number of SDs and real droplets will decrease.

One approach to solve this problem is to allow multiplicity to be a real number (floating point number) (Unterstrasser et al., 2017). The SDM can handle discrete and continuous systems because its formulation is based on the stochastic and discrete

nature of clouds. Nevertheless, simulations using this method may not behave as discrete systems in a small coalescence volume where the Smoluchowski equations do not hold (Dziekan and Pawlowska, 2017).

Another approach to solve the deterioration of multiplicity is to cast multiplicity from a floating point number to an integer by stochastic rounding (Connolly et al., 2021). For example, let $k$ be an integer, and let us set interval $[k, k+1]$ that contains a real number $l$; then, $l$ rounds to $k$ with probability $k+1-l$ and to $k+1$ with probability $l-k$. Hence, an expected value

obtained by the stochastic rounding process is consistent with the original real number $l$. Thus, the sampling accuracy does not decrease. Although this approach cannot prevent a decrease in the SDs, it can prevent the decrease in the number of real droplets statistically.





However, we can consider that these approaches are not optimal for ultrahigh-resolution simulations. Unterstrasser et al. (2017)'s discussion was based on the result of a box model, which is a closed system and requires a large ensemble of simulations to obtain robust statistics. In practical 3D simulations, the cloud microphysics field fluctuates spatiotemporally because of cloud dynamics and statistics in finite samples. If we sample a vast number of SDs, it is more natural to use a method that is closer to the constant multiplicity method. If such a method is used, we expect rare droplets to exist only in some cells rather than in every cell—this is a more natural continuation toward discrete systems. How can we develop such an initialization method? In addition, previous studies focused on collision–coalescence, but the sensitivity of cloud microphysical variability related to condensation/evaporation to SD initialization too must be considered. Against this background, which type of initialization is better overall?

To develop a new initialization method, we considered the simple method of generating a proposal distribution that connects the uniform sampling method to the constant multiplicity method. We chose the log of aerosol dry radius $\log r$ in the interval between $r_{\min}$ to $r_{\max}$ as the random variable. We denote an initial aerosol distribution as $n(\log r)$ and its normalization as $\hat{n}(\log r)$. The relation between $\xi$, $n$, and the proposal distribution $p$ was given by Shima et al. (2020) as

$$\xi(\log r) = \frac{n(\log r)}{N_{\mathrm{SD}}p(\log r)}, \tag{3}$$

where, $N_{\mathrm{SD}}$ is the SD number concentration. In the following explanation, for simplicity, we discretize the random variable into $k$ bins and nondimensionalize the bin width to $1$.

We define a probability simplex, which is a set of discretized probability distributions as follows:

$$C_k = \left\{ \boldsymbol{a} \in R^k : a_i \geq 0, \sum_{i=1}^{k} a_i = 1 \right\}. \tag{4}$$

Let us denote the discretized probability distribution of $\hat{n}$ as $\boldsymbol{b}_1 \in C_k$ and the uniform distribution as $\boldsymbol{b}_2 \in C_k$. Then, we define an $\alpha$-weighted mean distribution $\boldsymbol{a}$ as the Fréchet mean of $\boldsymbol{b}_1$ and $\boldsymbol{b}_2$:

$$\boldsymbol{a} = \operatorname*{arg\,min}_{\boldsymbol{a} \in C_k} \left\{ (1-\alpha)\mathcal{L}(\boldsymbol{a}, \boldsymbol{b}_1) + \alpha\mathcal{L}(\boldsymbol{a}, \boldsymbol{b}_2) \right\}, \tag{5}$$

where $\mathcal{L}$ is a metric to measure the distance between two distributions. A distribution $\boldsymbol{a}$ corresponds to a discretized and nondimensionalized proposal distribution of $p$. When the argument of the optimization is a function, $L^2$ norm is often used as the metric $\mathcal{L}$. In our case, since the argument is a probability distribution, the Wasserstein distance $W_2$ (Santambrogio, 2015; Peyré et al., 2019), which is a metric that measures the distance between two probability distributions, is a more natural choice. Several methods have been proposed to obtain solutions in Eq. (5) numerically. One method is to regularize the optimization problem of Eq. (5) by using the entropic regularized Sinkhorn distance $S_\gamma$ (Cuturi, 2013; Schmitz et al., 2018) ($\gamma$ is the regularization parameter) instead of the Wasserstein distance $W_2^2$. Another method is to use displacement interpolation (McCann, 1997), which is an equivalent formulation of Eq. (5). We used the method based on the Sinkhorn distance with $\gamma = 10^{-4}$ in Sect. 5. In this section and Sect. 4, we used the displacement interpolation specialized for the case where the random variable is one-dimensional to solve Eq. (5) more accurately. The specific forms of the Wasserstein distance $W_2$, Sinkhorn distance $S_\gamma$, and displacement interpolation are described in Appendix A.



**Figure 1.** (a) Normalized aerosol distribution given by VanZanten et al. (2011) (bold black line) and proposal distributions used for sampling ($\alpha = 0$–$1$). The bold red line shows the proposal distribution used for the uniform sampling method. (b) Relationship between dry aerosol radius and multiplicity when $\Delta V = 2^3 \, \mathrm{m}^3$ and 128 SDs per cell are sampled. (c) Distribution obtained by sampling $2^{16}$ SDs using the same setup as (b) ($\Delta V = 2^3 \, \mathrm{m}^3$ and 128 SDs per cell) and sorted by multiplicity in the ascending order. (d) The distribution corresponding to (c) when $L^2$ norm is used as a metric. The dotted lines in (b) and (c) indicate $\xi = 8^0, 8^1, 8^2, 8^3$.





We verified this method of generating proposal distributions by adopting a specific aerosol distribution $n(\log r)$. We used the bimodal log-normal distribution of VanZanten et al. (2011). This distribution is composed of ammonium bisulfate with a number density of $105\,\mathrm{cm}^{-3}$. We chose the interval for the random variable as $r_{\min} = 10\,\mathrm{nm}$ and $r_{\max} = 5\,\mu\mathrm{m}$, and adopted $k = 1{,}000$ bins and $\gamma = 10^{-4}$ to calculate proposal distributions.

 The proposal probability distributions obtained using various $\alpha$ are shown in Fig. 1(a). As $\alpha$ decreases, the uniform distri-

bution gradually changes to the normalized aerosol distribution, and probabilities (frequency for sampling) near both ends of the random variable decrease.

 The relationship between the aerosol dry radius and multiplicity for cell volume $\Delta V = 2^3\,\mathrm{m}^3$ and $128\Delta V^{-1}$ SDs are shown in Fig. 1(b). Multiplicity for the large dry radius of aerosol falls below 1 for $\alpha = 1.0$ but exceeds 1 for $\alpha = 0.8$ for all samples.

 Figure 1(c) shows the multiplicities of samples, which is obtained by sorting $2^{16}$ SDs by their multiplicity. How $\alpha$ changes

the dynamic range of multiplicity and the number of $\xi < 1$ samples can be clearly observed. Since the relationship between the aerosol dry radius and multiplicity does not change relatively if we increase the spatial resolution, we indicate $\xi = 8^0$–$8^3$ by dotted lines in Fig.1(c). As $\alpha$ decreases, the dynamic range of multiplicity decreases, and the minimum log multiplicity increases by an almost constant ratio when $\alpha \geq 0.2$. When $\Delta V = 1\,\mathrm{m}^3$, the multiplicity of all samples exceeds 1 if $\alpha \leq 0.7$. Similarly, the multiplicity exceeds 1 when $\Delta V = 50^3\,\mathrm{cm}^3$ if $\alpha \leq 0.6$, and when $\Delta V = 25^3\,\mathrm{cm}^3$ if $\alpha \leq 0.5$. Since the number

of samples of $\xi < 1$ and $0.5$ account for $7.82\%$ and $6.70\%$ of total samples, respectively, many invalid SDs are sampled if the uniform sampling method is used for $2\,\mathrm{m}$ resolution.

 Figure 1(d) shows the results corresponding to Fig. 1(c) obtained for $L^2$ norm instead of $W_2^2$ to generate proposal distributions using Eq. (5). In this case, as $\alpha$ decreases, the number of $\xi < 1$ samples decreases but does not vanish ($0.413\%$ of total samples when $\alpha = 0.1$), and the dynamic range of multiplicity does not change unless $\alpha = 0.0$. Thus, these results suggest that

the manner of connecting the two distributions is critical.

 How do aerosol statistics behave if we change $\alpha$ using the above method? The probability distributions of the number and mass concentration of dry aerosol for various $\alpha$ are shown in Fig. 2. We calculated the number and mass concentrations from 128 SDs. The multiplicity was cast to an integer using stochastic rounding for $\Delta V = 2^3\,\mathrm{m}^3$. We performed $10^5$ trials to obtain the probability distributions. The statistics of real droplets, corresponding to the limit when $\alpha = 0$ and the exact expected value,

is also shown by a dotted red line in each panel of Fig. 2.

 The expected values obtained by applying the importance-sampling method does not depend on the used proposal distribution. However, the variance of the expected values depend on the ratio of the original distribution to the proposal distribution, and the variance becomes small when the proposal distributions are similar. In fact, the aerosol number concentration distribution is narrow when the used proposal distribution is the same as the original distribution ($\alpha = 0$) (Fig. 2a), and it becomes

broader as $\alpha$ increases. Thus, the uniform sampling method introduces significant statistical fluctuations (or confidence interval) of aerosol number concentration. In contrast, the aerosol mass concentration distribution is narrow when $\alpha = 1.0$, and it broadens as $\alpha$ decreases (Fig. 2b). Thus, the uniform sampling method results in smaller statistical fluctuations of the aerosol mass concentration. That is, as $\alpha$ decreases, the importance sampling for the aerosol size distribution gradually changes its effect from the reduction of the variance of mass concentration to the reduction of the variance of number concentration. We





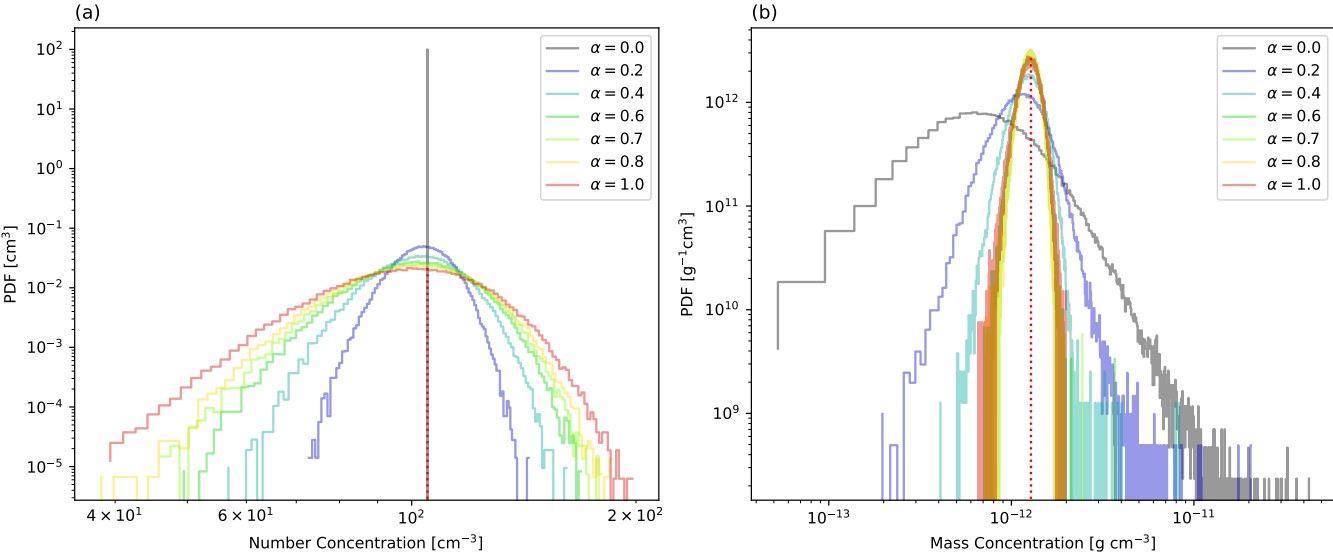

**Figure 2.** Probability distributions of (a) aerosol number concentration and (b) aerosol mass concentration, obtained by sampling from various proposal distributions. The red dotted lines show the exact expected values.

note that the results are almost identical when we store multiplicity as a real-type floating point number (not shown in the figures).

Based on the above considerations, the proposal distributions for $\alpha = 0.7$ were used for the numerical experiments described in Sect. 5. Although we focused on the statistical fluctuations of the aerosol, $\alpha$ may also be a sensitive parameter influencing the cloud dynamical and statistical fluctuations. Since this aspect is nontrivial because of the effect of cloud dynamics, we will describe the results of the sensitivity experiments for $\alpha$ in Sect. 4.3.

### 3.3 Model optimization

### 3.3.1 Strategy for acceleration

Based on the computers described in Sect. 2.3, we devised a strategy for optimizing the SDM. All algorithms used in the SDM have computational complexity of the order of SD numbers. In general, the PIC applications tend to have small B/F due to the large computations involved. This also holds for the SDM except for the collision–coalescence process because of the velocity interpolation to the position of SDs in movement, and the Newton iterations in activation/deactivation involve many calculations. Then, one may expect that a high computational efficiency can be achieved if the information of the grids and SDs are both on the cache as this can prevent the memory throughput being a bottleneck for the time-to-solution. However, since the calculation pattern in the cloud microphysics scheme changes depending on the presence of clouds and particle types, the codes in a loop body are complicated and often include conditional branches. Hence, high efficiency is difficult to achieve because of the difficulty of using SIMD vectorization and software pipelining. In the following paragraphs, we describe optimization





based on two strategies: First, we developed cache-efficient codes by cache blocking and reduction of information of the SDs. Second, we simplified the on-cache loop bodies to the maximum extent possible by excluding conditional branches.

We first considered applying cache-blocking techniques to the SDM. Since the L1 cache on A64FX is 64 KiB per core, 32 data arrays, which consist of $8^3$ grids of four-byte elements (each array consumes 2 KiB), can be stored on the L1 cache simultaneously. Similarly, since the user-available L2 cache is 7 MiB (of 8 MiB) / 12 = 597 KiB per core, two data arrays which consist of 128 SDs per cell $\times 8^3$ can be stored on the cache if an attribute of SDs consumes four bytes. Therefore, we divide the grids into groups of less than $8^3$ (hereafter called "blocks") for cache blocking. For each cloud microphysics process, we integrated all SDs by one time-step forward and then moved on to the next process. In the original SDM, a single loop is used for all SDs in the MPI domain. In this study, we decomposed this single loop for all SDs into loops for all blocks and all SDs in each block; subsequently, we parallelized the loop for all blocks using OpenMP by static scheduling with a chunk size of 1. Although applying dynamic scheduling to the loop for all blocks may improve load balancing among blocks, it is difficult to validate the reproducibility of the stochastic processes, such as collision–coalescence, because random seeds may change with every execution.

To simplify the loop body for the SDs in a block, it is essential that the gridded values in a block are a collection of similar values. The effective resolution in atmospheric simulations (Skamarock, 2004) imparts such numerical effects on the grid fields. The volume, which consists of $8^3$ grids, is comparable with the volume of effective resolution, which is the smallest spatial scale at which the energy spectrum is not distorted numerically by the spatial discretization. For example, since the energy spectrum obeys the $-5/3$ law roughly in the inertial range for LES, we regard the effective resolution as the smallest spatial scale at which the energy spectrum follows the $-5/3$ law. The typical effective resolution is $6\Delta$–$10\Delta$ for planetary boundary layer turbulence. The physical interpretation of effective resolution is that the flow is well resolved if the spatial scale is larger than $6\Delta$–$10\Delta$, and the variability decreases exponentially for scales smaller than this range. We used this prior knowledge to simplify the loop body, as described later.

### 3.3.2 Super-droplet movement

To guarantee consistency between the number density of SDs and air density during SD movement, we developed a second-order spatial accuracy conservative velocity interpolation (CVI) on 3D Arakawa-C grid. While the CVIs of the second-order spatial accuracy on 2D grids have been used in various studies such as Jenny et al. (2001), few studies have explored such CVIs on 3D grids. Recently, a CVI for divergence-free velocity field on a 3D A-grid was developed by Wang et al. (2015). We extend the method used in their study for the nondivergence-free velocity field on the C-grid. The accuracy of the interpolation is of the second order only within the cell, and we allowed discontinuous velocity across the cell. The derivation of our CVI using symbolic manipulation (Python SymPy) is available in Matsushima et al. (2023b). We only provide the specific form of the CVI in Appendix B.

The number of grid fields necessary to compute Eqs. (B7)–(B12) is important for computational optimization. While 24 elements (3 components $\times$ 8 vertices in a cell) are necessary to calculate the velocity at an SD position for trilinear interpolation (and the same applies for 2nd order CVI on the A-grid), only 18 elements are necessary for the second-order CVI on the C-grid





(B7)–(B12). That is, we can reduce $25\%$ of the velocity field data that occupies the L1 cache and can use the remaining cache for SDs, etc. The change in the spatial distribution of SDs through SD movement considering the spatial accuracy of CVI will be discussed in Sect. 4.2.

For warm clouds, since the information of the SD position accounts for half of all attributes, reduction of these data without

loss of representation and prediction accuracy contributes greatly to saving the overall memory capacity in the SDM. However, using FP32 instead of FP64 may cause critical problems due to the relative inaccuracy and nonuniform representation in the domain in the former case. In the following paragraphs, we describe these problems and a solution.

In the original SDM, the SD position is represented by its absolute coordinate over the entire domain, but this method requires many bits. However, since we already decomposed the domain into blocks, using the relative position of SDs in a

block is numerically more efficient. For this case, we can reduce the information per SD by subtracting information of partition by the MPI process and a block from the global position.

If we represent the position of SDs as a relative position in a block, additional calculations are necessary when an SD crosses a block. Such calculations introduce rounding errors for the SD position, and the cell position where the SD resides may not be conserved before and after its calculations. Let us consider an example. Consider a block that consists of a grid. Let us define

the relative position $x$ of SDs belonging $x \in [0,1)$ and the machine epsilon for the precision of floating point numbers as $\epsilon$. If SD crosses to the left boundary and reaches $-\epsilon/4 \notin [0,1)$, the relative position of the SD is calculated by adding the values of right boundary 1 in a new block to the SD position: $-\epsilon/4 + 1 \in [0,1)$. However, rounding to the nearest of its new position makes $\mathrm{round}(-\epsilon/4 + 1) = 1 \notin [0,1)$. For FP32, since $\epsilon \sim 1.2 \times 10^{-7} = 0.12\,\mu\mathrm{m}$ if we adopt meters as units, we expect this does not happen frequently. However, if such a case occurs even with only one SD of the vast number of SDs in the domain, the

computations may be terminated by an out-of-array index. Although a simple solution is exception handling using min/max or floor/ceiling, this solution may deteriorate the computational performance by making the loop bodies more complex, and the correction bias introduced by exception handling may be non-negligible when low-precision arithmetic is used. To ensure safe computing, the suitable approach is to calculate the relative position without introducing numerical errors.

In this study, we represented the relative position using fixed-point numbers. This format allows us to define the representable

position of SDs so that they are uniformly distributed in the domain, and integer–arithmetic-only calculations are used. Then, the same problem as in the case of simply using floating point numbers does not arise in principle. Let us denote the range for which the SD is in cell $k$ as $Z_k = [k, k+1)$ and the number of grids along an axis as $b$. Then the range of position in a block is represented as $Z = \bigcup_{k=0}^{b-1} Z_k$. We defined the conversion from $z \in Z$ to its fixed-point number representation $q$ as the following affine mapping:

$$q = 2^s \left( z - \left\lfloor \frac{b}{2} \right\rfloor \right). \tag{6}$$

When $b \leq 8$, $s = 21$ and when FP32 is used instead of INT32, the range of $-2^{23} \leq q \leq 2^{23} - 1$ is accurately represented by the mantissa of the floating-point numbers, and the representation does not exceed the representable range if it is only a few grids outside a block. With regard to the velocity, the amount of movement per step is represented using a fixed-point number. We used FP32 instead of INT32 for the actual representation because the representable range of fixed-point numbers is small





and could easily exceed its range by multiplication. Note that this step is avoidable if the architecture has instructions for fixed-point numbers such that multiplication and bit shift with rounding can be executed simultaneously (Jacob et al., 2018) as in ARM NEON and SVE2 (however, this is not the case in A64FX SVE).

By using relative coordinates for the SD positions within a block, the precision of their locations varies when $\Delta z$ is changed. This is because the change of position in real space is $2^{-s}\Delta z$ from Eq. (6) when the grid length is $\Delta z$, and the variation in
$q$ is 1; this value reduces for smaller $\Delta z$. In addition, the change in the relative position per time step is $2^{s}\boldsymbol{v}_i \Delta t \Delta z^{-1}$ when the time step is $\Delta t$; hence, it increases as $\Delta z$ decreases, thus providing a better representation of the relative position. $\Delta t$ is set sufficiently small to ensure there is no large deviation from the time step of tracer advection. Then, the change in the relative position does not change if the ratio of $\Delta t$ to $\Delta z$ is kept constant. In real space, the numerical representation accuracy of position and the arithmetic operations accuracy of the numerical integration vary with the spatial resolution and time step.
Therefore, we can maintain numerical precision for high-resolution experiments.

In terms of I/O, fixed-point numbers facilitate easy compression. For example, the interval of representable positions $q$ in real space with a block size of 8 is $0.95\,\mu\mathrm{m}$; this yields higher accuracy than the Kolmogorov length of $1\,\mathrm{mm}$ and thus is always excessive as a representation for DNS and LES. We can discard unnecessary bits when saving data on a disk.

### 3.3.3 Activation / Condensation

The time scale of activation/deactivation of the cloud condensation nuclei (CCN) is short if the aerosol mass dissolved in a droplet is small (Hoffmann, 2016; Arabas and Shima, 2017). Hence, the numerical integration of activation/deactivation is classified as a stiff problem. To solve Eq. (2), Hoffmann (2016) used the fourth-order Ronsenblock method with adaptive time stepping. SCALE-SDM employs the one-step backward differentiation formula (BDF1) with Newton iterations. Although BDF1 has first-order accuracy, it has good stability because it is an L-stable and implicit method, and we can change time
intervals easily because it is a single step method. However, with the implicit method, Newton iterations must be performed per SD, and the number of iterations required for convergence of the solution differs for each SD, thereby making vectorization a complicated task. To overcome this difficulty, the original SDM uses excessive Newton iterations (20–25) that are sufficient for all SDs to converge, assuming that numerical experiments are performed on a vector computer such as the Earth Simulator. However, we cannot tune codes for both vector computers and short-length vector computation by using SIMD instructions in
the same way. In the original SDM code, the loop body of time evolution by Eq. (2) is very complex because of the presence of conditional branches, grid fields at the SD position, and iterations; hence, it cannot issue SIMD instructions. Therefore, we devised a method to allow SIMD vectorization based on the previously described strategy.

Equation (2) is discretized by BDF1 as

$$f(R^2) = R^2 - p^2 - 2\Delta t A\left[S - 1 - \frac{a}{(R^2)^{1/2}} + \frac{b}{(R^2)^{3/2}}\right]$$

$$= 0,\qquad(7)$$

where $p$ is the current droplet radius, and $R$ is the updated droplet radius. Equation (7) has at most three solutions; in other words, one or two of them may be spurious solutions. However, the uniqueness of the solution is guaranteed analytically in the





following two cases (see Appendix C for derivation). Case 1, which depends on $\Delta t$, is

$$\Delta t \leq \frac{25b}{2Aa^2}\sqrt{\frac{5b}{a}}, \tag{8}$$

and Case 2, which depends on the environment and initial condition, is

$$S - 1 \leq 0, \quad p^2 < \frac{3b}{a}. \tag{9}$$

Case 1 implies that an activation time scale restricts the stable time step for each SD. Based on the estimation of temperature $T = 294.5\,\mathrm{K}$ at $z \sim 600\,\mathrm{m}$ in the BOMEX profile, when $\alpha = 0.7$, $87.7\%$ of SDs satisfy the condition for Case 1 if $\Delta t = 0.0736\,\mathrm{s}$, $91.0\%$ if $\Delta t/2$, and $100\%$ if $\Delta t/2^6$. Similarly, when $\alpha = 0.0$, $91.4\%$ of SDs satisfy the condition if $\Delta t = 0.0736$,
$97.6\%$ if $\Delta t/2$, and $100\%$ if $\Delta t/2^6$. The smaller the value of $\alpha$, the smaller is the frequency of sampling small droplets and the greater is the number of SDs that satisfy the condition.

On the other hand, Case 2 is a condition for the initial size of droplets $p$ in an unsaturated environment. In the BOMEX setup, since cloud fraction converges at a grid length of $12.5\,\mathrm{m}$ (Sato et al., 2018), we can estimate the ratio of SDs that satisfy Case 2 for higher resolutions by analyzing the results of similar numerical experiments using new SCALE-SDM. We define
droplets of the size $R \leq \sqrt{3b/a}$ as aerosol particles (or haze droplets), and droplets that are larger than $\sqrt{3b/a}$ and smaller than $40\,\mu\mathrm{m}$ as cloud droplets. We do not provide the detailed results, but the ratio of air density weighted volume (i.e., mass) where cloud water exists in a cell to the total volume in the BOMEX case is approximately $1.5\%$ in a quasisteady state based on the numerical experiments of our developed model. Therefore, we estimate that $98.5\%$ of SDs satisfy the condition of Case 2 in the BOMEX setup.

Therefore, if we ensure the uniqueness of the solution by Case 1 for a cloudy cell and Case 2 for a cell with no clouds, the frequency of exception handling during Newton iterations can be largely reduced. We first check whether we need a conditional branch of the unsaturated environment (of Case 2). Since the block has small volume that is comparable with the effective resolution, we can convert the conditional branches of the unsaturated condition for an SD to that for all SDs in a block with little or no decreasing ratio of SDs that satisfy the condition. This conversion of the conditional branch allows a
loop body of time evolution by Eq. (7) to be simple and specific to Case 2. Exception, when the initial size of droplets is larger, it is handled individually only if such droplets exist in a block. If the environment is saturated, we ensure the uniqueness of the solution by Case 1. In this case, we list the SDs that satisfy Case 1 and perform Newton iterations according to the list. Other SDs are calculated individually and using adaptive time stepping for unstable cases.

By using this method, we find that almost all SDs satisfy the uniqueness condition of the solution, and we should only focus
on optimizing these SDs. For tuning, the SDs in a block are classified into groups of 1024 SDs (which fit in the L1 cache), and each division calls the process of activation/condensation. In each call, the time evolution of each SD is calculated. A single loop for the updates of droplet radius calculates two iterations because this is the maximum number of Newton iterations that can allow SIMD vectorization and software pipelining without register spill of 32 registers with the current compiler we used for A64FX. The loop is repeated for all SDs in a division and breaks if the squares of all droplet radii of SDs fall below
the tolerance relative error of $10^{-2}$. Since the loop is vectorized by SIMD instructions and the number of iterations is often





limited to two if we use the previous droplet radius for the initial value for the Newton iterations, the computational time for activation/condensation is drastically less than that of the original SDM, as shown later.

### 3.3.4 Collision–Coalescence

The computational cost of the collision–coalescence process is already low for the algorithm developed by Shima et al. (2009).

We reduced the computational cost and data movement further rather than achieving a higher efficiency against theoretical peak performance of floating-point number operations. Since we used only the Hall kernel for coalescence, the coalescence probability was small for two droplets of small and similar sizes. Therefore, it is reasonable to ignore the collision–coalescence process in cells with no clouds. Note also that the no cloud condition precisely matches the Case 2 (9). If even a single cloud droplet exists in a block, it becomes necessary to sort the cell indices of all SDs in the block. However, we can remove sorting

if cloud droplets do not exist in a block. We do not sort the attributes of the SDs with cell indices as a key since they are already sorted with a block as a key, as will be described in Sect. 3.3.5. Further, some attributes are on the L2 cache during the collision–coalescence process due to cache-blocking. By not sorting the attributes of the SDs, the write memory access of SDs that do not coalesce is avoided. In the BOMEX setup, $98.5\%$ of the SDs satisfy Case 2 and we do not calculate the collision–coalescence of these SDs. Therefore, we expect a drastic reduction in the computational cost and data movement in some

cases in which cloudy cells occupy only a small fraction of the total domain volume. This method to reduce the computational cost potentially leads to a large imbalance as the Twomey-SDM by Grabowski et al. (2018). However, we also expect that the imbalance is mitigated better as cache blocking improves the worst-case elapsed time among the MPI processes.

### 3.3.5 Sorting for super-droplets

To effectively utilize cache-blocking during the simulation, the SDs in a block should be contiguous on memory. This is

possible if we sort the attributes of the SDs using the block ID as the sorting key when SDs move out from one block to another. This sorting is different from the usual sorting in which each block can send SDs to any other block; in the present sorting, the direction of SD movement is limited to adjacent blocks along $x, y$, and $z$ axes. Such sorting is commonly used in the field of high-performance computing. Although we did not make any novel improvement, we summarize this process because it is essential to our study, and some readers may not be familiar with on-cache parallel sorting for the PIC method

used during computation.

Since memory bandwidth generally limits sorting performance, it is essential to reduce data movement. In our case, the directions of data movement are limited, and most of the SDs in a block are already sorted. We should adopt a design such that these data are not moved and any unnecessary processes are not performed. We should also reduce the buffer size for sorting because of the low memory capacity of A64FX, perform parallelization, and reduce computational costs. However, ready-made

sorting, such as the counting sort, may not meet these requirements. Moreover, in the worst case, such sorting may be slower than the main computation in the SDM because of random access in the memory.

In this study, we sorted the attributes of SDs in three steps along the $x, y$, and $z$ axes. Each step requires at least two loops: copying in the SDs moving to adjacent blocks and copying back the SDs moving into the block. Since the SDs in a block either





stay in the same block or only move one block forward or backward, we did not sort the attributes of SDs with combinations
as a key. Instead, we made a list of SDs to move to reduce the computational costs and unnecessary data movement. Copying
in and back of the SDs to the working array should be divided into small groups so that size of the working array for SDs is
reduced by divisions. A loop for a block in each step can be parallelized naturally by using OpenMP. Although few invalid
SDs (buffer) may be included in the arrays, this study does not attempt to defragment them explicitly, expecting that the SD
movement and sorting with blocks as a key per microphysical time step may cause defragmenting.

This sorting can avoid the problems of using a ready-made algorithm. The drawback of the current implementation is that
a larger buffer space is necessary for SD attribute arrays because a block has few grids and the statistical fluctuation of the
number of SDs within a block is large. However, this can be improved if we adaptively adjust the size of SD attribute arrays in
a block according to air density and statistical fluctuations of SDs number.

This method is specialized for use during computation. If more flexible sorting is required, such as when the attributes of
SDs are sorted using the ID as a key for analysis, parallel sample sorting with larger working arrays should be employed.

## 4   Comparison of model performance

### 4.1   Methodology of performance evaluation

We evaluated the computational and physical performances by comparing the results of the new SCALE-SDM with those
obtained with the same model but using the conventional cloud microphysics schemes as well as with the results obtained
with the original SCALE-SDM. First, we describe the methodology of performance evaluation. Our optimization goal was to
enable ultrahigh resolution experiments of shallow clouds to reduce uncertainty and to contribute to solving future societal and
scientific problems. Therefore, we adopted a goal-oriented evaluation method instead of estimating the contributions of various
innovations for improving the time-to-solution. Here, we describe the evaluation of the time-to-solution and data processing
speed (throughput) to ensure the usefulness of our work for solving real problems. The throughput for the microphysics scheme,
including the tracer advection of the water and ice substances, is defined as follows:

$$\text{Throughput} = (\text{total \# of tracers, bins or SDs})$$
$$\times (\text{total steps})/(\text{elapsed time}), \tag{10}$$

where the number of steps and elapsed time correspond to the microphysics scheme. To compare the cloud microphysics
scheme that is based on different concepts, we defined the throughput for a bulk and a bin method by total tracers, including all
categories (e.g., water and ice) and statistics (e.g., number and mass). In contrast, we defined the throughput for the SDM by
sampling sizes in the data space $(\boldsymbol{x}, R, \xi, M)$. This is because we can add any attributes with less computational cost and data
movements, and the effective number of attributes may change during time integration; hence, considering many attributes for
defining throughput is inappropriate. For example, because we give an initial value of $R$ as a stationary solution of the Eq.
(2), $R$, $\xi$, and $M$ are initially correlated. We note that the number of tracers does not account for the water-vapor mass mixing





ratio. An increasing number of tracers or SDs improves the representation power for microphysics. Such an increase in the
representation power can be achieved easily for a bin method and the SDM, but is difficult for a bulk method.

To evaluate physical performance, we should confirm that we obtained qualitatively comparable results faster with the SDM
than with the original SDM. In terms of throughput, we should also confirm that we obtained qualitatively improved numerical
solutions if the elapsed time is approximately the same.

Next, we briefly describe the original SCALE-SDM and other cloud microphysics scheme used for performance evaluation.
We refer to the latest version of SCALE-SDM (retrieved 2022/6/6 from bitbucket private repository, contrib/SDM_develop) as
the original SCALE-SDM. The public version of the SCALE-SDM was used in Shima et al. (2020) (see code availability in
their paper). The base SCALE version of the original SCALE-SDM is 5.2.6. Meanwhile, the version developed in this study
is a developmental version based on SCALE 5.4.5. This version contains many improvements, such as untangling module
dependency and flexible module combinations for a model developer, in addition to our innovations. However, it does not have
critical changes to the physical process from version 5.2.6, except for our innovations, orders of calculations, and calculations
of the coefficient $A$ in Eq. (7) in the activation/condensation process. The original SCALE-SDM considers the dependency of
the diffusion coefficient and thermal conductivity on the environmental temperature, pressure, and water-vapor mass mixing
ratio used to calculate $A$. The original SCALE-SDM was used only for numerical experiments with the "original" SDM,
as labelled hereinafter. When focusing on some differences among cloud microphysics schemes, we will refer to the SDM
schemes associated with new SCALE-SDM or original SCALE-SDM as SDM-new or SDM-orig, respectively.

For the microphysics scheme, we used the Seiki and Nakajima (2014) scheme as a two-moment bulk method and Suzuki
et al. (2010) scheme as a (1-moment) bin method, both implemented in the SCALE. Seiki and Nakajima (2014) scheme solves
the number and mass mixing ratio of two categories of water substances and three categories of ice substances, and Suzuki
et al. (2010) scheme solves the mass-mixing ratio of each bin in discretized DSD for water and ice substances. In this study, we
considered only warm rain processes in the bin method. We used the latest versions of these schemes as is because performance
is not poor for solving real problems though these schemes may not be sufficiently optimized for A64FX. Some readers may
wonder why the SDM and the bin method solve only the warm process while the two-moment bulk method solves the mixed-
phase process. The validity of the comparisons of the computational performance of the SDM without the ice-phase process
with the mixed-phased two-moment bulk method and the future issues for optimization of the mixed-phase SDM is discussed in
Sect. 6.1. SCALE adopts terrain-following coordinates and contains features of map projection as a regional numerical model.
However, since any additional computational cost and data movement for these mappings cannot be ignored for high-resolution
simulations, we excluded these features in the new SCALE-SDM for the dynamical core, turbulence scheme, and microphysics
scheme.

**4.2  Warm bubble experiment**

We first evaluated the computational and physical performances via simple, idealized warm bubble experiments. The computa-
tional domain was $0.3 \, \text{km} \times 8 \, \text{km} \times 5 \, \text{km}$ for $x, y,$ and $z$ directions. For the lateral boundaries, doubly periodic conditions were
imposed on the atmospheric variables and positions of the SDs. The grid length was $100 \, \text{m}$. The initial potential temperature $\theta$,





relative humidity RH, and surface pressure $P_{\text{sfc}}$ were as follows:

$$\theta = \max\left(300, 300 + 4.0 \times 10^{-3}(z - 1{,}000)\right) \, \text{K,} \tag{11}$$

$$\text{RH} = 70\%, \tag{12}$$

$$P_{\text{sfc}} = 1{,}013.25 \, \text{Pa.} \tag{13}$$

The air density was given to be in hydrostatic balance. We provided a cosine-bell type perturbation of the potential temperature $\theta'$ to the initial field to induce a thermal convection:

$$\theta' = 2\cos\left(\frac{\pi}{2}\sqrt{\min(d_x + d_u + d_z, 1)}\right)^2 \, \text{K,} \tag{14}$$

$$d_x = (x - 50)^2/1{,}200^2, \quad d_y = (y - 2{,}500)^2/1{,}200^2,$$

$$d_z = (z - 500)^2/400^2.$$

For the SDM, the initial aerosol distribution was the same as that in VanZanten et al. (2011). For the two-moment bulk and the bin method, we used Twomey's activation formula and activated CCNs to cloud droplets according to the supersaturation as follows: $N = 100S^{0.462} \, \text{cm}^{-3}$, where $N$ is the cloud droplet number concentration, and $S$ is the supersaturation. The uniform sampling method was used to initialize the aerosol mass dissolved in a droplet and multiplicity. In the SDM-orig, SDs were initialized so that they were randomly distributed in the domain. In contrast, for SDM-new, SDs were initialized such that the SD number density was proportional to air density. In addition, to reduce the statistical fluctuations, instead of using pseudorandom numbers, we used the Sobol sequence (a low-discrepancy sequence) for the four-dimensional space of positions and aerosol dry radius in each block.

For the computational setup, the domain was decomposed to four MPI processes of one node in the $y$ direction using FX1000 (A64FX, 2.2 GHz). Local domains in each MPI process were further decomposed into blocks of size $3 \times 2 \times 5$ for $x, y$, and $z$ directions to apply cache-blocking for SDM-new. For the numerical precision of floating-point numbers, FP64 was used for the dynamics, two-moment bulk method, bin method, and SDM-orig. In contrast, SDM-new uses mixed precision, but calculations for SDs were primarily performed by FP32. For time measurement, we inserted MPI_Wtime and barrier synchronization at the start and end of the measurement interval. In this experimental setting, there were no background shear flows, and the simulated convective precipitation systems were localized and stationary in some MPI processes, thereby imposing a huge load imbalance of computational costs. However, the execution time was almost the same with and without barrier synchronization owing to stationarity of convective precipitation systems. In addition, if the measurement interval was nested, the times measured in its lowest level of nests did not include the wait time between MPI processes. To this end, we evaluated the performance of each microphysics subprocess without additional time. Time integrations were performed for $1800\,\text{s}$ by $\Delta t_{\text{dyn}} = 0.2\,\text{s}$ for dynamics, and $\Delta t = 1.0\,\text{s}$ for other physics processes.

Figure 3 shows the elapsed times of the warm bubble experiments for various cloud microphysics and different number of tracers or SDs per cell. Here, we show only the elapsed times of those numerical simulations that were completed in less than 3 h and that required less than $28\,\text{GB}$ of memory. The elapsed times obtained by the bin method (BIN) behave as $O(N^2)$, while

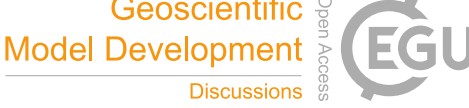

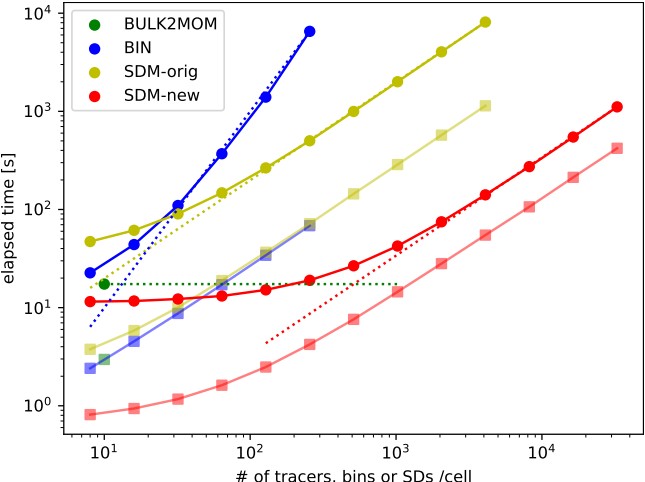

**Figure 3.** Elapsed times of the total (circles) and that of tracer advection and SD tracking (squares) using the two-moment bulk method (green), bin method (blue), SDM-orig (yellow), and SDM-new (red) with different numbers of tracers or mean SDs per cell. Here, SD tracking included SD movement and sorting with a block as a key. The blue dotted line is the line proportional to $N^2$. The red and yellow dotted lines are lines proportional to $N$. The green dotted line indicates a constant determined by $N$.

those of the SDM-orig behave as $O(N)$, indicating that the collision–coalescence calculation developed by Shima et al. (2009) contributes to a reduction in elapsed times. The SDM-new drastically reduced the elapsed time compared to the bin method and SDM-orig for the same number of bins or SDs. Moreover, the elapsed time obtained using the SDM-new with 128 SDs per cell was about the same as that obtained using the two-moment bulk method (BULK2MOM).

The results seem to contradict the intuition that computations using sophisticated cloud microphysics schemes take more time than simpler schemes because of the high computational costs of the former. The main reason for the present results is related to the tracer advection and SD tracking, which is a bottleneck for the elapsed times, as is described below, rather than to other cloud microphysics subprocesses. The elapsed times of tracer advection and SD tracking are shown in Fig. 3. The elapsed time of tracer advection and SD tracking obtained using the bin method and SDM-orig are comparable and increase as

$O(N)$. For small $N$, the elapsed time of tracer advection and SD tracking for the SDM-new up to 128 SDs per cell are shorter than that for the two-moment bulk method, which is advantageous in terms of the elapsed time of simulations.

The advantages of SDM-new against the two-moment bulk for calculating tracer and SD dynamics are fewer calculations, higher compactness, and more reasonable use of low-precision arithmetic for SD tracking than for tracer advection. While tracer advection requires a high-order difference scheme to reduce the effect of numerical viscosity, SD tracking does not

require a high-order scheme. We used Fujitsu's performance analysis tool (fapp) to measure the number of floating-point operations (FLOPs). We found 303.915 FLOPS per grid and tracer for tracer advection (UD5) excluding FCT and 164.3 FLOPS per SD for SD movement using CVI of second-order spatial accuracy. Since the calculation of UD5 requires values at five grids and halo regions of width 3 in each direction, the calculations are not localized, and a relatively larger amount of communication



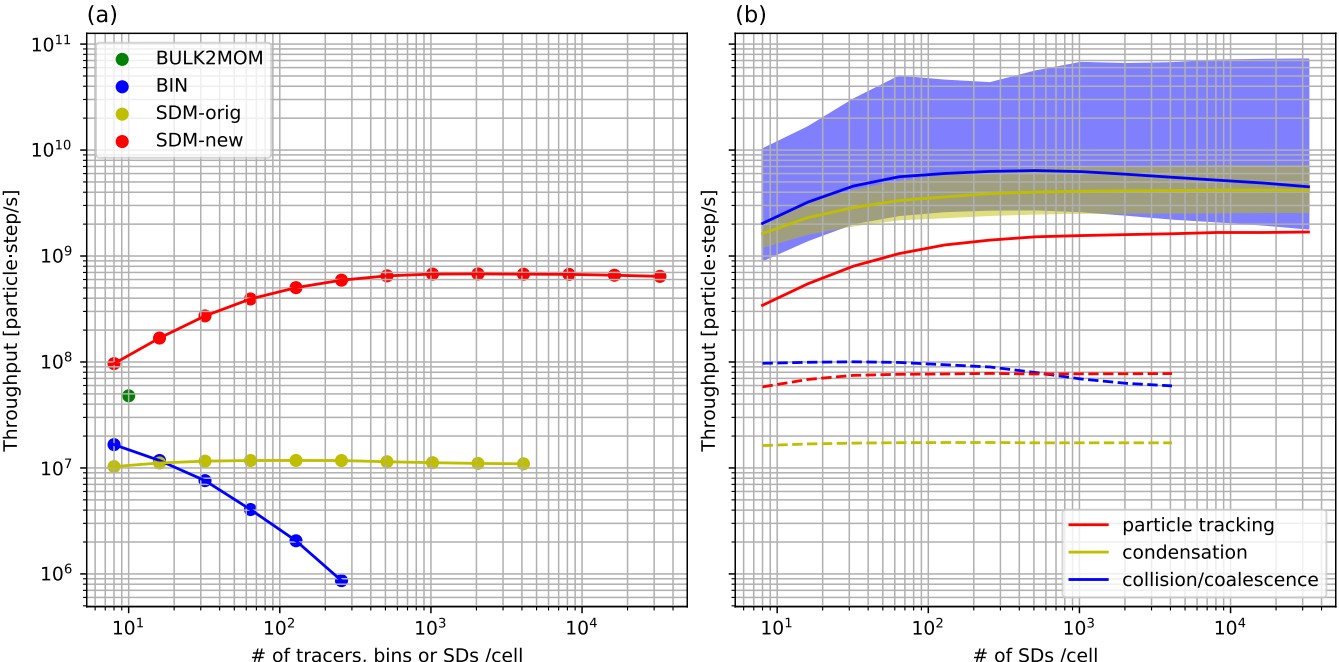

**Figure 4.** (a) Data throughput of microphysics for using the two-moment bulk method (green), bin method (blue), SDM-orig (yellow), and SDM-new (red) with different numbers of tracers or mean SDs per cell. (b) The mean data throughput of SD tracking (SD movement and sorting with a block as a key), condensation process, and collision–coalescence using SDM-orig and SDM-new with different numbers of mean SDs per cell. The dotted lines and solid lines show the mean data throughput for SDM-orig and SDM-new, respectively. The range between the minimum and maximum throughputs of condensation and collision–coalescence for SDM-new is indicated by the colors because the load imbalance is significant for only SDM-new.

is necessary. For SD tracking, the calculations for a single SD require only grids that contain the SD, and communication is
necessary only when the SD moves out of the MPI process. If FP32 is used for tracer advection, one of the advantages of the
SDM-new over the two-moment bulk method is lost. However, the calculations of tracer advection require differential oper-
ations, which may cause cancellation of the significant digits. This likely cannot be ignored for high-resolution simulations
where the amplitude of small-scale perturbations from the mean state decreases, especially for variables that have stratified
structures (e.g., water-vapor mass-mixing ratio). On the other hand, for the proposed SD tracking, numerical representation
precision of the SD positions in physical space becomes more accurate as the grid length and time interval decrease simulta-
neously. Therefore, the use of FP32 for high-resolution simulations is reasonable. Of course, another important factor behind
these results is the fact that the calculations of other SDM subprocesses are no longer bottlenecks in SDM-new.

Now, we compare computational performance among different cloud microphysics schemes in terms of data throughput. The
throughput of the microphysics scheme (tracer advection, SD tracking, and microphysics subprocesses) for a different number
of mean SDs per cell is shown in Fig. 4(a). The throughput of the bin method decreases as the number of bins increases,





while that of the SDM-orig remains almost constant but shows a slightly decreasing trend as the number of mean SDs per cell increases. The throughput of both methods is smaller than that of the two-moment bulk method; hence, the elapsed time does not become smaller than that obtained using the two-moment bulk method. In contrast to SDM-orig, the throughput of SDM-new is similar to that of the two-moment bulk method for eight SDs per cell, and it increases as the number of SDs

increases. Because of the increase in the throughput, which is related to the increase in computational performance and the grid calculations, the elapsed time obtained using the SDM-new resists linear increase with the number of SDs. Hence, the elapsed time becomes comparable with that obtained using the two-moment bulk method even for larger SDs ($\sim 256$). However, as with the SDM-orig, the throughput of the SDM-new shows a decreasing trend when the number of mean SDs per cell exceeds 1024. The maximum throughput of the SDM-new is 57.6 and 14.13 times that of SDM-orig and two-moment bulk method,

respectively.

The throughputs of subprocesses obtained by SDM-orig and SDM-new are shown in Fig. 4(b). The throughputs obtained by SDM-orig are almost constant with respect to the number of SDs per cell. As the number of SDs increases, the throughput of SD tracking converges to a constant, and the throughput of collision–coalescence decreases from approximately 256 SDs per cell. The throughput obtained by SDM-new is larger than that obtained by SDM-orig for all subprocesses. As the number

of mean SDs per cell increases, the throughput of SD tracking and condensation increase and converge to constants. The throughput of collision–coalescence increases to about 256 SDs per cell but then decreases as in the case of SDM-orig. The minimum throughput of collision–coalescence behaves as the mean throughput, while the maximum throughput increases as the number of SDs per cell increases. This finding reflects the fact that the throughput decreases only in MPI processes that contain clouds in the domain because the L1 and L2 cache miss ratio increases because the random access in the cache and

memory during collision–coalescence calculations increases for a large number of SDs. The maximum throughputs of SD tracking, condensation, and collision–coalescence obtained by the SDM-new are 21.6, 241, and 64.8 times that obtained by SDM-orig, respectively. In this study, we did not examine the contributed innovations for the acceleration of the throughput in detail. However, the acceleration rate of the throughput is roughly explained by SIMD vectorization ($\times 16$) for SD tracking and also reduced computational cost by terminating Newton iterations faster ($\times 16 \times 10$) for condensation. Before optimization, the

condensation calculations were the bottleneck of SDM-orig. After optimization, SD tracking calculations were the bottleneck of SDM-new.

Although we report only the computational performance on FX1000 (A64FX), our innovations are also effective on Intel Xeon. For example, using FUJITSU Server PRIMERGY GX2570 M6 (CPU part: a theoretical peak performance of 5.53 TFLOPS and memory bandwidth of 409.6 GB/s) equipped with Intel Xeon Platinum 8360Y, the elapsed time obtained

using the two-moment bulk method was 18.439 s, and that obtained using the SDM-new with 128 SDs per cell on average is 14.486 s. The maximum throughput of the SDM-new is 25.1 times that of the two-moment bulk method. The large ratio of the throughput against FX1000 indicates that using FX1000 instead of a more commercial computer with low memory bandwidth (GX2570 M6) is more advantageous for the two-moment bulk method.

We evaluated the physical performance of SCALE-SDM. First, we show the differences between the first- and second-order

CVI for SD tracking. In the SDM, we can add any new attribute, such as ID, to each SD. By using the ID for analysis, we

**Figure 5.** Distributions of SD positions at (left) $t = 600\,\mathrm{s}$ and (right) $t = 1{,}200\,\mathrm{s}$ colored by the initial $y$ coordinate ($Y$) when CVI of the first order (CVI-1) and second order (CVI-2) spatial accuracy are used for SD movement. The range of $0 \le y \le 5{,}000$ and $1{,}000 \le Y \le 4{,}000$ are shown in each panel.



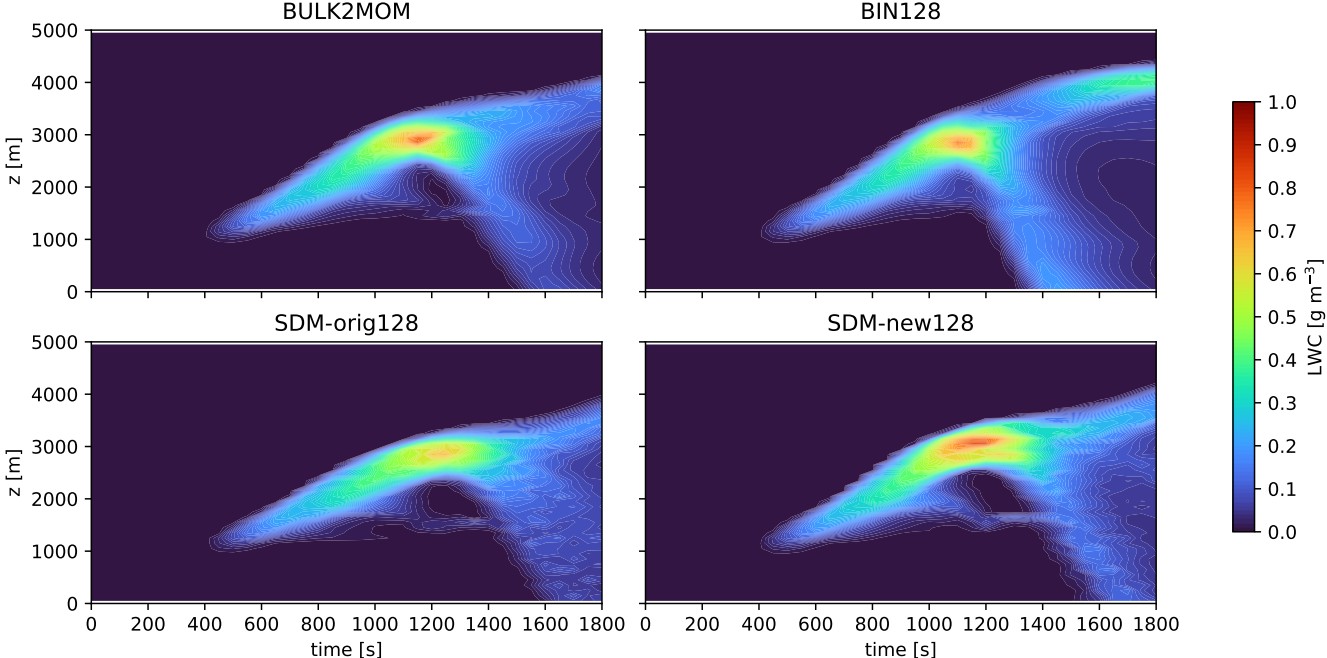

**Figure 6.** Horizontally averaged time–height cross section of the liquid water content (LWC) for different cloud microphysics schemes.

calculated the initial position of the SD to investigate SD mixing. The distributions of SD positions colored by the initial $y$ coordinate for warm bubble experiments (SDM-new with 128 SDs per cell on average) are shown in Fig. 5. Buoyancy torque induced by the initial bubble generates vorticity, and the results are different for the case when the first- and second-order CVI are used. At $t = 600.0\,\mathrm{s}$, a staircase-like pattern with width approximately the grid length appears in the CVI-1 because

it does not consider the variation in the velocity component relative to its orthogonal direction within the cell. In contrast, such a pattern does not appear in CVI-2. The motion of the particle in the fluid can be chaotic even for simple flow fields. Particles experience stretching and folding in flows, and fine and complex structures are generated even from large-scale flows. These features are called chaotic mixing (Aref, 1984) from the Lagrangian viewpoint, and they are distinct from turbulence mixing. At $t = 1,200.0\,\mathrm{s}$, fine structure ($x = 1,500\,\mathrm{m}, z = 1,200\,\mathrm{m}$) and filament ($x = 1,800\,\mathrm{m}, z = 2,800\,\mathrm{m}$) appear in CVI-2,

whereas such structures are noisy and obscure in CVI-1. This result indicates that such structures in CVI-1 can be nonphysical assuming that structures in CVI-2 are more correct. The accuracy of the CVI may affect the entrainment mixing induced by thermal, a coherent vortex ring in clouds. However, because our innovation works for the variations in velocity within a cell, it is difficult to discuss the effect on larger scales such as the cell-averaged variables, rather than the distributions of SDs. In addition, because in-cloud flows are generally well-developed turbulent flows, it is difficult to separate the effect of chaotic

and turbulent mixing. Second, we compared the results of warm bubble experiments among different cloud microphysics schemes. The horizontally averaged time–height sections of the liquid water content (LWC) are shown in Fig. 6. Here, we denote the names of the experiments, followed by the number of bins and SDs per cell on average, such as SDM-new128,





for the results obtained using SDM-new with 128 SDs per cell on average. Here, the elapsed times for the selected cases are SDM-new128 < BULK2MOM < SDM-orig128 < BIN128. In all cases, the qualitative characteristics of time evolution, such

as bubble-induced cloud generation and precipitation pattern, are the same. In addition, the LWC patterns of BIN128, SDM-new128, and SDM-orig128 are in good agreement with those of BIN256, SDM-new32768, and SDM-orig4096 (not shown in figures), respectively, which were obtained using the bin method and using the SDM with the maximum possible number of bins or SDs per cell. Thus, the LWC solutions attain convergence with the given number of bins and SDs per cell. The time evolution of SDM-new128, SDM-orig128, BULK2MOM, and BIN128 are very similar until $t = 1200\,\mathrm{s}$. For precipitation

onset at the surface, SDM-new128 ($t = 1,600\,\mathrm{s}$) is slower than BULK2MOM ($t = 1,500\,\mathrm{s}$) and BIN128 ($t = 1,400\,\mathrm{s}$). For the LWC remaining after precipitation in the upper layers ($z \sim 3,500\,\mathrm{m}$), SDM-new128 is larger than BULK2MOM and smaller than BIN128. The results of SDM-new128 and SDM-orig128 deviated slightly after $t = 1200\,\mathrm{s}$, partly because of the different SCALE versions. However, precipitation onset at the surface and LWC remaining in the upper layers are close to the results of SDM-new. We conclude that differences between SDM-orig128 and SDM-new128 in terms of the LWC are small as per the

warm bubble experiments.

### 4.3 BOMEX and SCMS cases

In Sect. 4.2, we discussed the evaluation of the computational performance using mainly data throughput by increasing the number of mean SDs per cell. This approach is appropriate for comparing SDM-orig and SDM-new as the contributions of the stencil calculations that are not relevant to the innovations in this study become small. However, the comparison of SDM-new

with the two-moment bulk and the bin methods may not be fair. In general, the computational efficiency improves in actual use cases when the number of grids per MPI process is increased. The number of grids in each MPI process used in Sect. 4.2 was relatively small. In addition, the numerical settings of warm bubble experiments were too simple to be regarded as representative of real-world problems. Therefore, we also evaluated computational and physical performances for the BOMEX case and a case study of isolated cumulus congestus observed during the Small Cumulus Microphysics Study field campaign

(Lasher-Trapp et al., 2005)—this case is referred to as the SCMS case—, as they present more practical problems.

The experimental settings for the BOMEX case were based on Siebesma et al. (2003). The computational domain was $7.2\,\mathrm{km} \times 7.2\,\mathrm{km} \times 3.0\,\mathrm{km}$ for $x$, $y$, and $z$ directions, and the horizontal and vertical grid lengths were $50\,\mathrm{m}$ and $40\,\mathrm{m}$, respectively. The experimental settings for the SCMS case were based on the model intercomparison project for the bin methods and particle-based methods conducted in International Cloud Modeling Workshop 2021 (see Xue et al. (2022) and reference therein). The

computational domain was $10.0\,\mathrm{km} \times 10.0\,\mathrm{km} \times 8.0\,\mathrm{km}$, and the grid length was $50\,\mathrm{m}$. For both cases, the time interval was $\Delta t_{\mathrm{dyn}} = 0.1\,\mathrm{s}$, $\Delta t_{\mathrm{adv}} = 2\Delta t_{\mathrm{dyn}} = 0.2\,\mathrm{s}$, $\Delta t_{\mathrm{phy}} = 0.2\,\mathrm{s}$. The Rayleigh damping imposed was $500\,\mathrm{m}$ and $1,000\,\mathrm{m}$ from the top of the domains for the BOMEX and SCMS cases, respectively. In the SDM, SDs were not initially placed in the Rayleigh damping layers, and we did not generate or remove SDs in the regions during simulations. For initialization, the uniform sampling method (i.e., the proposed method using $\alpha = 1.0$) was adopted for both BOMEX and SCMS cases. For the SCMS

case, we also used the proposed method using $\alpha = 0.5$ and $0.0$ for SDM128 to investigate the sensitivity of cloud microphysical variability to the initialization method.



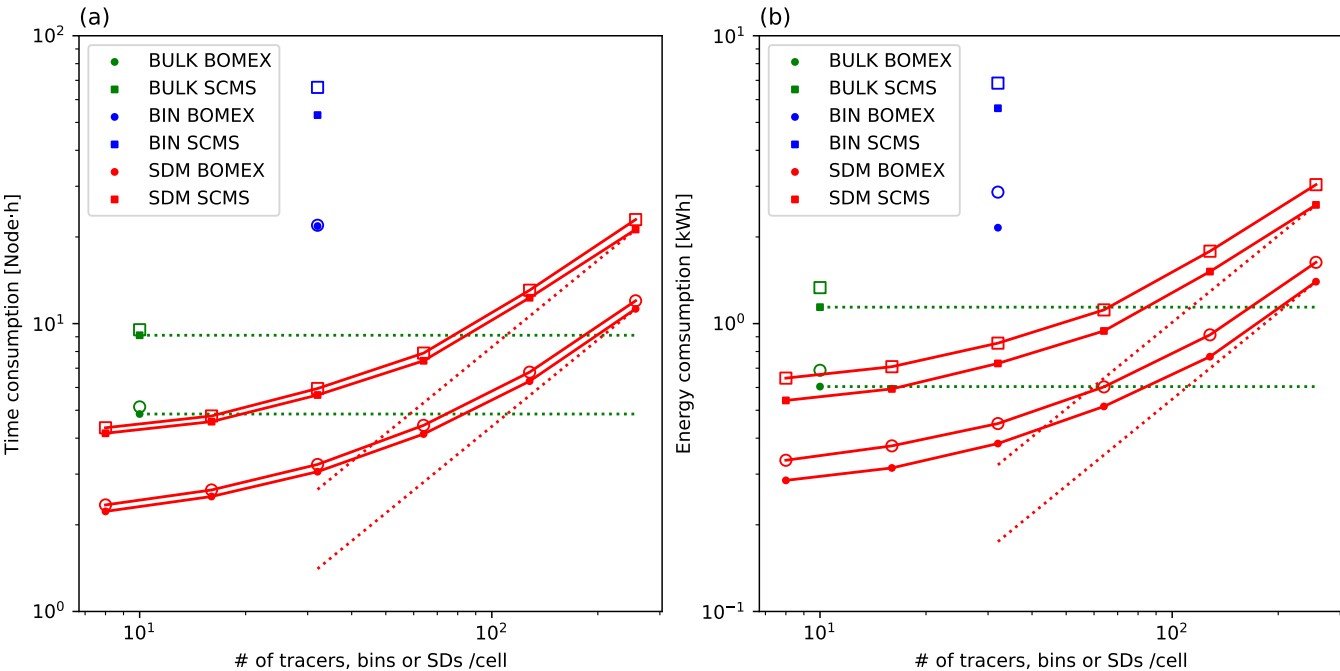

**Figure 7.** Computational resources of BOMEX and SCMS experiments for various cloud microphysics schemes and different numbers of tracers or mean SDs per cell: (a) node-hours using normal and boost mode, (b) energy consumption using boost and boost eco mode. Here, for (a), the results for the boost and normal modes are shown by filled and open markers, respectively. For (b), the results for boost eco mode and boost mode are shown by filled and open markers, respectively. The red dotted lines show the lines proportional to $N$.

Ensemble experiments with three members using different initial perturbations controlled by other random seeds were conducted for each experiment. The number of nodes used for simulations was determined as the minimum values so that the memory usage was within the system memory per node. For example, one node of FX1000 was used in both cases for the two-moment bulk method, and one node and two nodes of FX1000 were used in BOMEX and SCMS cases for SDM128, respectively. For time measurement, we used MPI_Wtime but did not use barrier synchronization. At the same time of simulating three ensemble experiments, we operated Power API to switch among the normal mode, boost mode, and boost-eco mode for each ensemble member. We measured the energy consumption per node between measurement intervals by operating Power API. The measured energy accounted for the energy consumed by all computing and assistant cores, L2 cache, memory, Tofu interconnect, optical modules, and PCI Express.

The computational resources for various cloud microphysics schemes using the normal, boost, and boost eco modes for each numerical setting are shown in Fig. 7. We first focused on the node-hours when the normal mode is used. Here, node-hours is a measure of the amount of time for which computing nodes are used, and it is calculated as the product of occupied nodes and the hours. Comparing the results between the SDM with 32 SDs per cell on average (SDM32) and the bin method with 32 bins (BIN32), the node-hours of BIN32 are 6.8 times and 11.1 times that of SDM32 for BOMEX and SCMS cases, respectively.





The node-hours consumed using the SDM with 64 to 128 SDs per cell are comparable to those consumed when using the two-moment bulk method, and they do not increase linearly with increasing number of mean SDs. The results are important because we should use larger than 128 SDs per cell to obtain converging solutions such as the cloud droplet number concentration with respect to the number of SDs (Shima et al., 2020; Matsushima et al., 2021). In terms of memory usage, the simulations using the two-moment bulk method consumed about $28.5\,\mathrm{GB}$ of system memory, whereas those using the SDM with 128 SDs per cell consumed about twice that memory. When the number of available nodes is limited, the simulations using the two-moment bulk method still have the advantage of increasing the problem scales.

In Fig. 7, we see that the difference in the patterns among the modes and between (a) and (b) is qualitatively small, and the advantage of SDM over the two-moment bulk method and bin method is apparent. For example, the energy consumption of BIN32 is 8.0 times and 6.4 times that of SDM32 for BOMEX and SCMS cases, respectively, when the boost eco mode is used. In terms of node-hours (Fig. 7 (a)), the following relations are observed: boost mode < normal mode. Further, node-hours for the boost eco mode is closer to that for the normal mode (not shown in the figure). For energy consumption (Fig. 7 (b)), the boost eco mode < boost mode, and the energy consumption by the normal mode is higher than that by the boost eco mode (not shown in the figure). The results obtained for the boost eco mode have the best power performance from the viewpoint of computational resources among different modes. Although the boost eco mode offers an option to improve power performance when floating-point operations per time are not large, the power performances when using not only two-moment bulk and the bin method but also SDM are improved.

We evaluated the physical performance of microphysical spatial variability obtained by the SCMS case experiments. This case is suited for investigating the effect of entrainment-mixing which may lead to different results among microphysics schemes. In this study, we focused on analyzing the results obtained using the SDM with 128 SDs per cell on average; the computational resources in this case are comparable to those for the two-moment bulk method but smaller than those for the bin method with 32 bins. The top panel of Fig. 8 shows contoured frequency by altitude diagrams (CFADs) of the cloud droplet number concentration (CDNC), LWC, mean radius, and standard deviation of radius for one member ($\alpha = 1.0$) of SDM128 at $t = 6,600\,\mathrm{s}$. The selected time of the snapshot was when the cloud top height almost reached its (local) maximum first (the movie of the CFADs from $t = 3,600\,\mathrm{s}$ to $t = 10,800\,\mathrm{s}$ is available in the supplements: SCMS-R50SD128-CFAD-m1.mp4). Once the clouds evolved to have depths larger than approximately $\sim 3\,\mathrm{km}$, the CFAD patterns did not change much with time qualitatively even for the other ensemble members (in supplement: SCMS-R50SD128-CFAD-m2.mp4, SCMS-R50SD128-CFAD-m3.mp4). To enable intercomparison of models for the readers, each variable was calculated from only the SDs in one cell. However, the spatial scales of the variables were shorter than the scales of effective resolution, which may introduce a numerical influence on the statistics (Matsushima et al., 2021). The adiabatic liquid water content (ALWC) was calculated using Eq. (6) in Eytan et al. (2021), which is recommended for the most accurate comparison with the passive tracer test as a reference solution. In addition, we calculated the adiabatic CDNC. The activated CDNC depends on the updraft of the parcel when crossing the cloud base, and hence, on the supersaturation of the parcel. However, we simply assign an adiabatic CDNC at the cloud base $1,155\,\mathrm{cm}^{-3}$ as the maximum value assuming large supersaturation, and all haze droplets activate to the cloud






**Figure 8.** Contoured frequency by altitude diagrams (CFADs) of cloud droplet number concentration (CDNC), LWC, and mean and standard deviation of the radius for SCMS experiments. Snapshots of (top row) SDM128 and (middle row) SDM128 obtained using FP64 as floating-point number operations and those with collision–coalescence calculations in all grids and (bottom row) SN14 are shown. Units of each variable are $m^{-1} \cdot cm^3$, $m^{-1} \cdot kg \cdot g^{-1}$, $m^{-1}\mu m^{-1}$, $m^{-1}\mu m^{-1}$, respectively. In each panel, the quartiles of variables at each height are indicated by white lines. The adiabatic predictions of CDNC and LWC are indicated by black lines in the panels of CDNC and LWC.





droplets. Then, we define CDNC including the height dependency as $N_a = 1{,}155\rho_a(z)/\rho_a(z_{\mathrm{cbase}})\mathrm{cm}^{-3}$, where $\rho_a(z)$ is the
air density of the most undiluted cells in $z$-section, and $z_{\mathrm{cbase}}$ is the cloud base height.

One of the drawbacks of the SDM is the statistical fluctuations caused by finite samples. Indeed, CDNC varies largely
centered around $500\,\mathrm{cm}^{-3}$, and some samples exceed simple adiabatic prediction, and some samples of LWC also exceed
ALWC. However, the frequencies, for which CDNC and LWC are larger than their adiabatic limits, are about one order of
magnitude smaller than frequencies within adiabatic limits. Near the cloud base, the most frequent values of LWC are close
to ALWC. At $z = 2{,}500\,\mathrm{m}$, the simulated congestus have a kink formed by detrainment indicating that cloud elements are
left behind from the upward flow or moved followed by a downward flow (not shown in figures). The frequency for which
LWC$\sim 0$ is large here. Above the middle layer of the clouds ($z > 2{,}500\,\mathrm{m}$), the LWC is strongly diluted. The mean radius
narrowly varies in the lower layers of the clouds, but the variation becomes large above $z = 2{,}500\,\mathrm{m}$ for small droplets because
of entrainment and activation. The most frequent values of the standard deviation of radius decrease as the height increases
below $z = 2{,}500\,\mathrm{m}$ in the adiabatic cores of the clouds. Above the middle layers of the clouds, the most frequent values of the
standard deviation of radius remain almost constant or increase with height, and the medians of the frequencies at each height
reach $3\,\mu\mathrm{m}$ at the upper layers of the clouds. These features are consistent with typical observations (Arabas et al., 2009). To
compare the obtained solution with a reference solution, we also adopted the same experimental setup as that of SDM128
but used mainly FP64 (50 bits per grid for SD tracking). Meanwhile, we changed the tolerance relative error for Newton
iterations in condensation calculations to $10^{-6}$ and computed the collision–coalescence process in all grids. The middle panel
of Fig. 8 shows the CFAD analyzed by the reference experiment. Although our innovations include the use of FP32 for the
numerical representation of droplet radius, the differences in the patterns of the mean radius between the top and middle
panels of Fig. 8 are minor. As we will show in Sect. 6.4.1, simply using FP16 may cause stagnation of the droplet radius and
numerical broadening of the DSD for condensational growth, but the use of FP32 does not cause these problems. Therefore,
our innovations do not worsen the physical performance compared with the reference solution and typical observation.

The CFAD for the two-moment bulk method (BULK2MOM) is shown in the bottom panel of Fig. 8. The variability of the
CDNC and LWC for BULK2MOM is smaller than those for SDM128. As in the SDM128, the mean radius increases with
height but exhibits a strange mode at $6\,\mu\mathrm{m}$. The relative standard deviation of the cloud droplet radius for the two-moment
bulk method was analytically calculated to be $0.248$ because it prescribes the shape parameters of the generalized gamma
distribution. Thus, the mean and standard deviation of the radius have identical patterns except for scaling. The standard
deviation of radius for BULK2MOM is smaller than that for SDM128 and does not decrease as height decreases in the adiabatic
core, as seen in the case of SDM128. To understand the origin of this strange pattern of the mean and standard deviation of
the radius, we calculated the mean and standard deviation of SDM128 from the CDNC and LWC, assuming that the DSD
shape follows the empirical DSD of the two-moment bulk method. In this case, the strange pattern did not appear (not shown).
Therefore, to investigate intracloud microphysical variability, it is not appropriate to use the two-moment bulk method because
the CDNC and LWC thus obtained are restricted by the effect of empirical assumptions. Our numerical simulations using new
SCALE-SDM provide a qualitatively better solution than that obtained using the two-moment bulk method with comparable
computational resources.



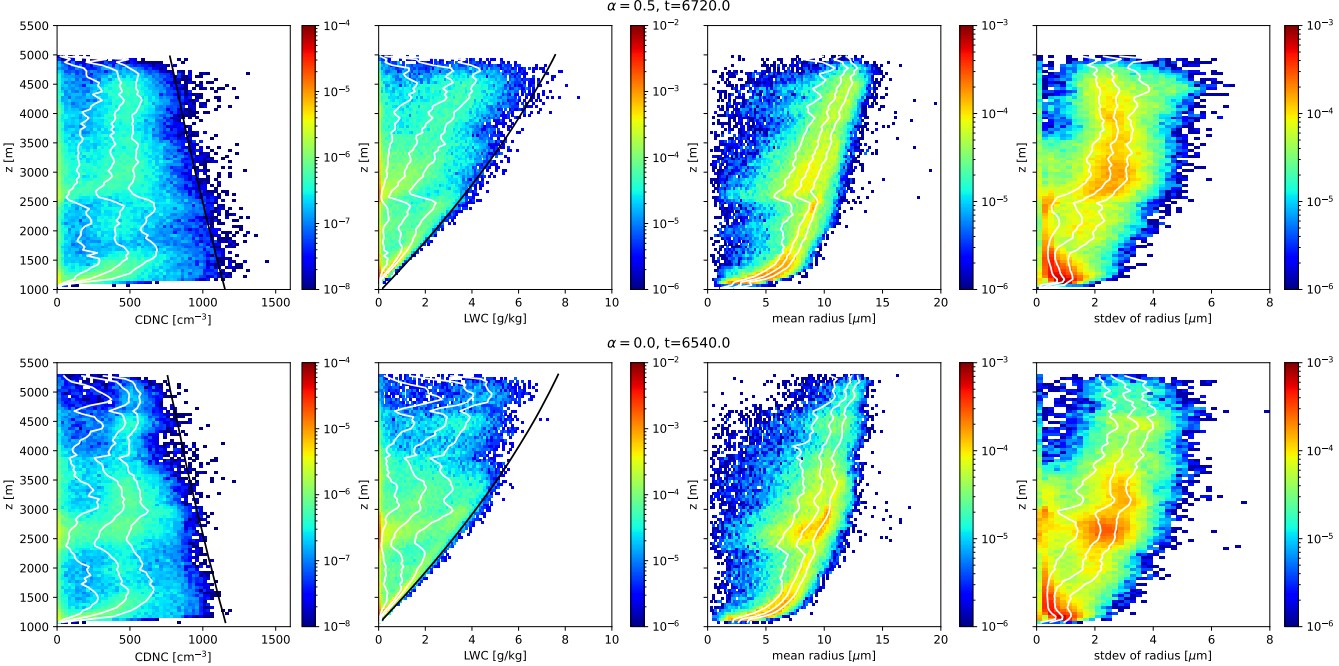

**Figure 9.** CFADs of CDNC, LWC, and mean and standard deviation of radius for SCMS experiments using different initialization parameter ($\alpha = 0.5, 0.0$). Units of each variable are $\mathrm{m}^{-1} \cdot \mathrm{cm}^3$, $\mathrm{m}^{-1} \cdot \mathrm{kg} \cdot \mathrm{g}^{-1}$, $\mathrm{m}^{-1}\mu\mathrm{m}^{-1}$, $\mathrm{m}^{-1}\mu\mathrm{m}^{-1}$, respectively. In each panel, the quartiles of the variables at each height are shown by white lines. The adiabatic predictions of CDNC and LWC are shown by black lines in the corresponding panels.

In Sect. 3.2, we proposed a new initialization method for ultrahigh resolution simulations. Because the aerosol number concentration of the SCMS case is high (11 times that in VanZanten et al. (2011)), the importance of collision–coalescence is relatively low. Then, it may be reasonable to use another initialization parameter instead of $\alpha = 1.0$, which is favorable for faster convergence of collision–coalescence with a number of SDs per cell. Despite the original motivation to develop an initialization method for high-resolution simulations, we investigated the sensitivity of microphysical variability to $\alpha$ for the SCMS case by

$50\,\mathrm{m}$ resolution simulations. The CFADs for the initialization parameters $\alpha = 0.5$ and $0.0$ are shown in Fig. 9. The selected times of the snapshots are $t = 6,720\,\mathrm{s}$ and $t = 6,540\,\mathrm{s}$ respectively, which are determined for the same reason as in the case of $\alpha = 1.0$. If we assume no spatial variability of aerosol number concentrations and that all aerosols (haze droplets) are activated to cloud droplets, the maximum CDNC for the SCMS case is $1,155\,\mathrm{cm}^{-3}$. Nevertheless, the maximum values of CDNC reach $1,500\,\mathrm{cm}^{-3}$ for $\alpha = 1.0$. As $\alpha$ decreases, the variation in CDNC decreases, and the maximum values of CDNC are almost

limited within $1,155\,\mathrm{cm}^{-3}$ for $\alpha = 0.0$. These results show that the statistical fluctuation of aerosol number concentration for large $\alpha$ affects that of the CDNC. We can interpret the cause of the statistical fluctuation of the CDNC as follows. Suppose that for a given supersaturation, the haze droplets that have an aerosol dry radius larger than the specific threshold activate to form cloud droplets, as assumed in the Twomey activation model. Then, the CDNC in each grid cell is determined by the SDs that





have an aerosol dry radius larger than the threshold size. If the proposal distribution with a limited area of the support (domain

of the random variable) for aerosol dry radius is not similar to the aerosol size distribution, the distribution of the CDNC also has a statistical fluctuation due to the property of importance sampling. Of course, the actual 3D simulations exhibit other effects, such as spatially varying supersaturation, considering a more detailed activation process and the dynamical fluctuation induced by varying the numbers of SDs per cell. On the other hand, the statistical fluctuation of aerosol mass concentration for small $\alpha$ does not affect that of LWC. Instead, the fluctuations of the LWC decrease as $\alpha$ decreases, and LWC is almost

within the ALWC. This finding can be physically interpreted as follows. As $\alpha$ decreases, the samples of small droplets that have a small contribution to the aerosol mass concentration increase, leading to more significant statistical fluctuations of aerosol mass. Similarly, the statistical fluctuation of the LWC for only haze droplets is larger as $\alpha$ decreases (not shown in figures). However, without the turbulence effect, droplet growth by condensation causes the droplet radius of the samples to be more similar with time, thereby damping the statistical fluctuations. In terms of microphysical variability without collision–

coalescence, the obtained results for small $\alpha$ are considered to be more accurate because the prediction of the microphysical variable for each grid is less variable. The sensitivity of variability for the mean and standard deviation of radius to $\alpha$ is unclear. However, the largest values of the mean radius become larger as $\alpha$ increases. This is consistent with the fact that such initialization that leads to a larger dynamic range of multiplicity (larger $\alpha$ in this study) creates more large droplet samples, and triggers precipitation, as observed in the study using a box model (Unterstrasser et al., 2017). The results suggest that

for nonprecipitating clouds, small $\alpha$ may be allowed even for low-resolution simulations, and optimization of $\alpha$ or proposal distribution by constraints from observations can be explored. For ultrahigh resolution simulations, when using small $\alpha$ such that the multiplicity of SDs is not smaller than 1, the microphysical variability induced by condensation/evaporation (majority of the droplets) and precipitation (triggered by rare, lucky droplets), and turbulent fluctuations interacting with clouds through phase relaxation can simultaneously better represent the natural variability of clouds.

## 5 Applicability for large-scale problems

### 5.1 Scalability

In Sect. 4, we evaluated the computational and physical performances of SCALE-SDM by relatively low-resolution experiments using at most four nodes. Here, we show the feasibility of using our model for large-scale problems using more computing nodes. First, we show the scaling performance of the new SCALE-SDM for the BOMEX case. Although our numerical

model adopts a hybrid type of 3D and 2D domain decompositions using the MPI, we investigated only weak scaling performance in horizontal directions with the vertical domain fixed. This is because almost all clouds localize in the troposphere, and hence, extending the vertical domain does not provide any benefit.

For all directions, the grid length is set to $2\,\mathrm{m}$. The number of grids without halo grids per MPI process is $72 \times 72 \times 96$ and $18 \times 18 \times 1,536$ for the 3D and 2D domain decompositions, respectively. The shape of network topologies is a 3D torus.

In one direction of the 3D torus, the number of 16 MPI processes/nodes are used for vertical domain decomposition. In each node, $2 \times 2$ MPI processes per node are used for horizontal domain decomposition. For grid conversions between 3D and 2D





domain decompositions, $N_z = 16$ is decomposed by $(N_{xl}, N_{yl}) = (4, 4)$. For the grid system in 2D domain decompositions, grids are divided into groups of $6 \times 6 \times 6$ for cache blocking. For arithmetic precision, FP64 is used for the dynamical process, and mixed-precision is used for the SDM. Here, most of the representations and operations for the SDM use FP32/INT32. In
contrast, reduction operations such as calculation of SDs within a cell to liquid water in the cell use FP64, and calculations of SD cell positions use INT16. The scales of problems per node are mainly limited by memory capacity because the usable system memory of HBM2 is $28\,\mathrm{GB}$, and SD information consumes $8.32\,\mathrm{GB}$ memory capacity per node for the above setting if extra $36\,\%$ of the buffer arrays for the SDs is reserved.

The node shapes are $4 \times 4 \times 16$, $24 \times 16 \times 24$, $48 \times 16 \times 48$ with horizontal domains of $1,152\,\mathrm{m}$, $6,912\,\mathrm{m}$, $13,824\,\mathrm{m}$, respectively.
For the BOMEX case, streaks and roll convection with about $1\,\mathrm{km}$ wavelength are apparent for high-resolution simulations, and they restrict cloud patterns (Sato et al., 2018). To exclude the effect of domain size, we evaluated the weak scaling performance from the horizontal domain of $1,152\,\mathrm{m}$.

Time integrations were performed for $3,680\,\mathrm{s}$. The time interval was $\Delta t_{\mathrm{dyn}} = 0.0046\,\mathrm{s}$, $\Delta t_{\mathrm{adv}} = 4\Delta t_{\mathrm{dyn}} = 0.0184\,\mathrm{s}$, $\Delta t_{\mathrm{phy}} = 0.0736\,\mathrm{s}$ for dynamical process, tracer advection, and physical process, respectively. The short integration time compared with
the standard numerical settings for the BOMEX case is because some challenges remain in outputting large restart files (see Sect. 6.3) and mitigating load imbalance due to clouds. Further, it takes longer to obtain profiles of the computational performance. However, since the integration time is sufficiently long for clouds to be generated in the domain and to approach a quasisteady state, the obtained performance is a good approximation of the actual sustained performance. Note that we set $\Delta t_{\mathrm{adv}}$ smaller than the constraint of CFL condition for tracer advection (typical wind velocity of shear flows is about $10\,\mathrm{ms}^{-1}$
for the BOMEX case). Because the time-splitting method was applied for compressible equations, the noise induced by the acoustic wave is dominant on the tracer fields if $\Delta t_{\mathrm{adv}}$ is larger than several times $\Delta t_{\mathrm{dyn}}$. If an instantaneous value for dynamical variables is used for the time integration of physical processes, and $\Delta t_{\mathrm{phy}}$ is several times $\Delta t_{\mathrm{dyn}}$, a compressional pattern may arise for the SD density because the instantaneous dynamic variables have a specific phase of the acoustic wave pattern. To reduce these effects, we used dynamic variables averaged over $\Delta t_{\mathrm{adv}}$ for physical process calculations.

For measuring the computational performance, we used both timer (MPI_Wtime) and fapp. We used the results obtained by the timer only for obtaining a quick view of the elapsed time and the results obtained by fapp for other detailed analysis, such as the number of floating point number operations, number of instructions, and amount of memory transfer. We note that the measured results have an overhead through the use of fapp. The I/O time is included in the total elapsed time of the time integration loops, but it is quite small. We did not use explicit barrier synchronization before and after the time measurement
intervals. All-to-all communications with blocking in the local communicator, which consists of $N_z$ MPI processes, were used for converting the grid systems. Since barrier synchronization is not performed for all MPI processes, the wait time of communication can affect across dynamics and microphysics processes. However, even if the variations in the presence of clouds in each MPI process is large, these effects become small when the variation of the clouds in each group of $N_z$ MPI processes is small. Since no large-scale cloud organization occurs in this case, we evaluated the computational performance of
individual components separately, such as the components of dynamics and microphysics.

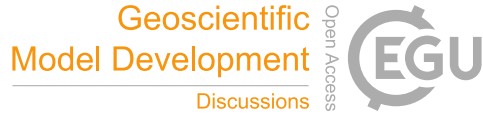

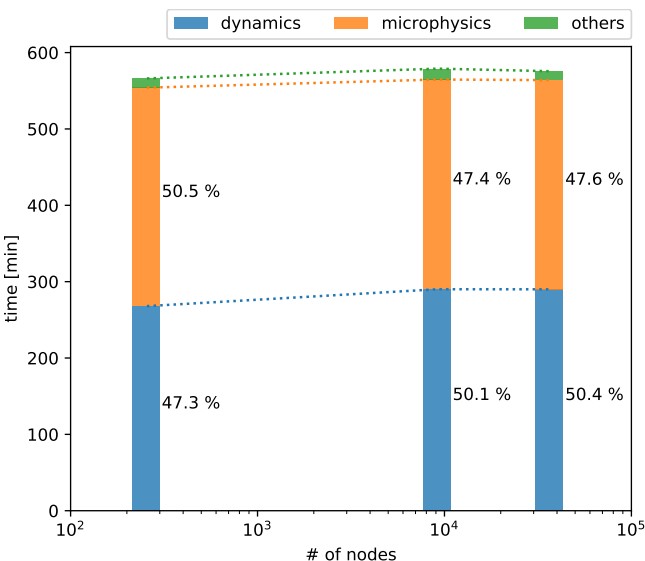

**Figure 10.** Weak scaling performance of 2 m resolution experiments for the BOMEX case.

The weak scaling performance of the new SCALE-SDM obtained for the above settings is shown in Fig. 10. We adopted the grid system for 3D domain decomposition as the default grid system. The elapsed time for the grid system conversion from 3D to 2D or from 2D to 3D domain decomposition is included in a required physics process. In this case, it is only included in the elapsed time for the microphysics process during the time integration loop. The total elapsed time of the experiments was 566 min for 256 nodes and exhibits 98 % weak scaling for 36,864 nodes. In addition, the elapsed time for dynamics and microphysics was 268 min and 286 min for 256 nodes and exhibit 92 % and 104 % weak scaling for 36,864 nodes, respectively. All-to-all communications during the conversion of grid systems do not degrade the weak-scaling performance for microphysics because the hop counts of communications are small, and the number of MPI processes involved is small. Other physics processes, such as the turbulence scheme, surface flux, and idealized radiation, consume only about 2% of the total elapsed times.

## 5.2 Largest-scale problem

The detailed profile of the largest problems among our experiments for the weak scaling test is summarized in Table 3. The peak ratio is obtained against the theoretical peak performance of FP64 operations. The overall time integration loop (excluding the initialization and finalization of the simulation) achieves 7.97 peta floating-point operations per second (PFLOPS), which is 7.04% of the theoretical peak performance, and 13.7 PB/s which is 37.2% of the peak performance. The achieved peak ratio of the FLOPs is comparable to that of 6.6% by NICAM-LETKF (Yashiro et al., 2020), which was nominated for the 2020 Gordon Bell Prize. In addition, because the effective peak ratio of memory throughput performance is approximately > 80% for the STREAM Triad benchmark, the obtained peak ratio achieves about half of it, implying that the overall calculations



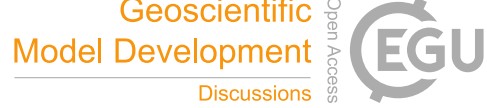

**Table 3.** Elapsed time, FLOPS (peak ratio of the FLOPS [%]), Peta instructions per second, memory throughput (peak ratio of the memory throughput [%]), and particle throughput (# of floating point operations per SD)

|  | Time [min] | Speed [PFLOPS] | PIPS | Memory Throughput [PB/s] | Part. Throughput [particle·step/s] |
|---|---|---|---|---|---|
| Time integration loop | 576 | 7.97 (7.04) | 1.86 | 13.7 (37.2) | |
| Dynamics | 290 | 8.55 (7.55) | 2.03 | 20.5 (55.7) | |
| Microphysics | 274 | 7.50 (6.62) | 1.69 | 6.25 (16.9) | $2.86 \times 10^{13}$ |
| Short time step | 238 | 9.50 (8.39) | 2.19 | 21.3 (57.9) | |
| Tracer time step | 15.0 | 5.85 (5.17) | 1.78 | 21.6 (58.7) | |
| Tracking | 87.9 | 15.3 (13.5) | 2.14 | 2.89 (10.5) | $8.91 \times 10^{13}$ (171) |
| Condensation | 32.6 | 18.2 (16.1) | 5.35 | 5.28 (14.3) | $2.40 \times 10^{14}$ (75.9) |
| Coalescence | 5.75 | 7.58 (6.69) | 2.96 | 17.5 (47.3) | $1.36 \times 10^{15}$ (5.57) |

utilize HBM2 well. At the subprocess level, the short time step (for acoustic waves), which consumes most of the elapsed
time in dynamics, achieves 9.5 PFLOPS (8.39% of the peak) and 21.3 PB/s (57.9%). SD tracking and condensation achieve
15.3 PFLOPS (13.5% of the peak) and 18.2 PFLOPS (16.1% of the peak), respectively. These relatively high performances are
partly attributed to the use of FP32 for most operations. For these cases, the effective peak ratios for the calculations should be
the ratio against peak performance for FP32, which are half of the ratio against FP64 and hence not high. The bottleneck of
these processes is a large L1 cache latency of A64FX due to the random access of the grid fields. For collision–coalescence,
the peak ratio of the floating-point operations is very low. However, in terms of instructions per second (IPS), which includes
integer operations, store and load operations, and computation for conditional branches, the performance is not low compared
with those of the other processes.

In the SDM, SD tracking, condensation, and collision–coalescence computations consume 44% of the elapsed time. Others
contributors are the mainly the times for data movement, such as the conversion of grid systems, sorting SDs with a block as
a key, and load imbalance for the presence of the clouds, which still needs improvement. In this experiment, because of the
limited memory capacity, we divided loops with a block into small groups to reduce the memory usage for sorting. This affects
the increase in the latency and wait time because of synchronization by increasing the communication counts and inefficient
OpenMP parallelization by decreasing the loop counts—this is one reason for the long time required for data movement.

The data throughput of the SDM, which we define as shown in Eq. (10) in Sect. 4, as well as the elapsed time, is a fundamental
measure that includes not just the number of floating-point operations but also all the information about a numerical model, a
scheme, an implementation, and a computer. In terms of data throughput, we attempted to compare our results with those of a
tokamak plasma PIC simulation, a study that shares similarities in computational algorithms but has an entirely different target.
The tokamak plasma PIC simulation performed by Xiao et al. (2021) was nominated for the 2020 Gordon Bell Prize. It used
the total system of the Sunway OceanLight, which has a higher theoretical peak performance than the Fugaku. For the largest-
scale problems, the throughput of the SDM reaches $2.86 \times 10^{13}$, which is comparable to $3.73 \times 10^{13}$ particle·steps/s of their
study. In addition, the throughput of each subprocess is larger than the simulated throughputs. The major difference between





our and their results in terms of data throughput is the number of operations per particle—it is $\sim 5,000$ in their simulations, which is much larger than that achieved in our study. In research focusing on FLOPS as a measure for better computational performance, it is common to reduce the application B/F by increasing the number of FLOPs per particle to fit a computer that has a small B/F, which may result in small data throughput. However, we achieved data throughput comparable to that of their study; this is a more practical measure for application than merely considering the FLOPS even if the throughputs are comparable.

Finally, we roughly estimated the elapsed time considering that the two-moment bulk and the bin method were used for the same numerical experiments of the same problem size, and we compared our results with the previous work. From Table 3, the elapsed time for tracer advection (only water vapor mass mixing ratio) is $15\,\mathrm{min}$, and the peak ratio of memory throughput is $58.7\%$, which indicates good performance from the viewpoint of the effective peak performance. In the current implementation, since the time evolution of the tracers was solved by each tracer separately, the total elapsed time for tracer advection was easily estimated as the product of $15\,\mathrm{min}$ with the number of tracers. If water-vapor mass-mixing ratio plus 10 or 32 tracers are used for the bulk of the bin method, respectively, the elapsed time for tracer advection is estimated as $165\,\mathrm{min}$ and $495\,\mathrm{min}$ respectively; these values are larger than the elapsed time of the sum of the SD movement and tracer advection ($103\,\mathrm{min}$). For the bin method, the estimated elapsed time of tracer advection is larger than the total elapsed time of the SDM. Here, we explain that this relationship is robust with respect to the optimization of the bin method. The bottleneck of the tracer advection is memory throughput for B/F$= 3.69 > 0.3$. We computed the tracer flux in each direction from the mass flux and tracer variables of the previous step to update the tracer variable based on the finite volume method. If the arrays are large, memory access occurs in nine arrays (one component mass flux, tracer variable, and one component tracer flux for each direction). Since the mass flux is common for different tracer variables, and memory access for the tracer variable occurs thrice for computing the tracer flux, our implementation is not optimal for minimizing memory access. Thus, in principle, there is room for optimization. However, since there are no known successful examples of such optimization in the Fugaku (and in the other general-purpose CPUs), tracer advection is a memory-bound application in practice. Then, a possible optimization may be to simply refactor the codes, and we may be able to improve the memory throughput performance of tracer advection to achieve up to $80\%$. However, even with such optimization, the elapsed time of tracer advection with 33 tracers is estimated to be $363\,\mathrm{min}$. Therefore, our simulations with the SDM still have an advantage against the bin method.

## 6 Discussion

### 6.1 Mixed-phased cloud

In this study, we optimized and sophisticated the SDM for only warm microphysics processes and compared the computational performance using the warm SDM with the two-moment bulk method with tracers for ice categories. However, unless a similar method can be applied for cold processes (Shima et al., 2020), the efficacy of our method for practical problems may be small. Here, we discuss possible extensions to such cases.





Shima et al. (2020) extends the SDM approach to consider the morphology of ice particles. Ice processes considered in
Shima et al. (2020) include immersion/condensation and homogeneous freezing; melting; deposition and sublimation; and
coalescence, riming, and aggregation. To solve these processes, new attributes, such as freezing temperature, equatorial radius,
polar radius, and apparent density, are introduced. A critical aspect of the approach using the SDM is that despite many
attributes for water and ice particles, the effective number of attributes decreases if particles are in either the water or ice state.
For example, when warm and cold processes are considered, the apparent density is necessary for ice particles but it is not so
for liquid droplets. Indeed, the memory space used for the attributes for ice particles can be reused to represent the attributes
of droplets when they change to liquids, and vice versa. Thus, if well implemented, the increase of the memory requirement
for considering both warm and cold processes can be mitigated. In addition, if we can easily discriminate the particle state as
water or ice, the computational cost of the mixed phase SDM when used for warm clouds will be almost identical to that of the
warm SDM.

Thus, this is relatively easy. The initial freezing temperature takes values from $-38\,^{\circ}\mathrm{C}$ to $-12\,^{\circ}\mathrm{C}$, and its low-precision
representation is reasonable because of its small dynamic range. For the mixed-phase SDM, a water droplet is converted to an
ice particle when the environmental temperature falls below the freezing temperature. An ice particle is converted to a water
droplet if the environmental temperature exceeds $0\,^{\circ}\mathrm{C}$. All SDs in the cell are ice particles if the environmental temperature
is below $-38\,^{\circ}\mathrm{C}$. In contrast, all SDs in the cell are water droplets if the environmental temperature is above $0\,^{\circ}\mathrm{C}$. For such
cases, we can use specialized codes for water droplets or ice particles, thereby reducing the cost of conditional branches for
each particle. This justifies our comparison between the SDM and the two-moment bulk method.

On the other hand, if water droplets and ice particles are mixed in the cell, the computational performance will decreased
because of some challenges such as the different formats of information of water droplets and ice particles and SIMD vector-
ization. In addition, they make assumptions such as particles are in either the water or ice states, and instantaneous melting
occurs above $0\,^{\circ}\mathrm{C}$. These problems should be addressed in future works by making fewer assumptions.

## 6.2 Terrain

An extension of this study to the case with terrain is also essential. For terrain-following coordinates with the map factor used
in the regional model, our SD tracking using a fixed-point representation of the SD's position can be applied when we map
from the terrain-following coordinate to the Cartesian coordinates. However, if coordinate mapping is introduced, the CVI
scheme may not guarantee consistency between air density and SD density. In addition, there is an additional computational
cost for SD tracking. If computational cost is critical, we can include the effect of terrain in the SCALE-SDM by combining it
with the immersed boundary or cut-cell methods. Then, the computational performance will not deteriorate because additional
cost arises only in the block with the terrain. When realistic terrain is considered, another additional cost will be incurred at the
top/bottom/side boundaries to impose inflow/outflow conditions. Moreover, it will be more complex to sample SDs by ensuring
consistency between air density and SD density because the probability for sampling will be a 3D distribution. However, the
cache-blocking algorithm introduced in this study also helps improve the computational efficiency for such complex processes.
By examining if and how we can construct a CVI for terrain-following coordinates and spherical coordinates is a future task.





## 6.3 Long-time run

In Sect. 5.2, we focused on the feasibility of large-scale problems and performed only about 1 h time integration. To investigate
the statistical behavior of clouds, longer time integration is required. However, if we create a checkpoint/restart file for the
largest-scale problems in this study, it will require approximately $225\,\mathrm{TB}$ without compression for the total number of SDs of
$9.39 \times 10^{12}$ SDs, and each SD consists of six attributes with four bytes for each attribute. It is possible to output such large-size
files on the Fugaku because of its system design and to utilize most of the computing nodes in Fugaku for a short period.
However, it is usually difficult to use such a large amount of storage only for one project. One way to address this problem is
further optimize to improve the strong scaling performance for longer integration in a single run. Another way is to use lossy
compression by giving up the exact reproducibility of the simulations. For example, if we do not store haze droplets on a disk,
and resample SDs at the restart, the amount of data can be reduced to less than $10\%$ for BOMEX. On the other hand, this
method will eliminate the effect of hysteresis on the SDs. A better resampling method from compressed data is a challenge for
future studies.

## 6.4 Can we achieve higher performance?

### 6.4.1 Lower precision arithmetic

Since A64FX is a general-purpose CPU with FP16/INT16, it may be possible to reduce memory usage and data movement
and achieve higher performance if low-precision arithmetic is utilized. Unfortunately, we could not use it simply for this study.
However, since using lower-precision arithmetic may be essential for future high-performance computing, we briefly discuss
the obstacles for the same.

Grabowski and Abade (2017) showed that supersaturation fluctuation can broaden the DSD even in the adiabatic parcel.
Their method and Abade et al. (2018) serve as a type of parameterization of the turbulence effect for the SDM. Instead of
using the 3D numerical model, we discuss the sensitivity of the DSD to numerical precision based on Grabowski and Abade
(2017). The numerical settings of the adiabatic parcel model are the same as theirs. The box size of a parcel is $50\,\mathrm{m}$. Time
integration was performed for $1,000\,\mathrm{s}$ by the time interval of $\Delta t = 0.2\,\mathrm{s}$. In contrast, we used different numerical precisions
(FP64, FP32, and FP16) and different rounding modes (round to the nearest and two modes of stochastic roundings) for time
integration of droplet radius. The detailed mathematical property of stochastic rounding is described in Connolly et al. (2021).
Mode 1 rounds to an up/down direction considering the precise position (calculated by other methods such as higher precision
arithmetic) in the interval between the upward and downward rounded values. Mode 2 rounds to an up/down direction with a
probability of $1/2$. For basic operations such as the inner product, the expected values calculated using the stochastic rounding
of mode 1 are identical to true values.

The DSDs at $500\,\mathrm{s}, 1,000\,\mathrm{s}$ are shown in Fig. 11. Without the effect of supersaturation fluctuations, the results obtained using
FP64 and FP32 are in good agreement. In contrast, the DSD obtained using FP16 is stagnant in time because the tendency of
condensational growth is too small to add to the droplet radius (i.e., loss of trailing digits). However, the DSD obtained using
FP16 with mode 1 rounding is similar to that obtained using FP64 or FP32 because the tendencies can be added to droplet





radius stochastically. If we focus on individual SDs, some SD may experience more rounding down, and some may experience more rounding up. That is, the DSD is slightly diffusive compared with that obtained using FP64 and FP32. If we use FP16 with mode 2, the obtained DSD shifts toward a larger droplet radius, indicating that the probability for rounding direction is essential to ensure accuracy. With supersaturation fluctuations, the DSD obtained using FP16 is less stagnant because the magnitude of tendencies does not reach 0 because of the fluctuations. The DSD obtained using FP16 with mode 1 is similar to that obtained using FP64 or FP32 except for a slight diffusional trend.

These results indicate that we cannot simply use FP16, but we can use FP16 with mode 1 rounding for some problems. For example, suppose $\Delta t$ is used in low-resolution simulations. In that case, the DSD becomes less diffusive because the effect of rounding becomes small, and the magnitude of supersaturation fluctuations becomes large. On the other hand, if $\Delta t$ is small in high-resolution simulations, the effect of rounding error on the DSD becomes large. In such cases, the use of FP16 is not suitable even if stochastic rounding is used. For SD movement, because of the variable precision for SD position, it may be feasible to use fixed-point number representation such as INT16 using mode 1 in high-resolution simulations. For collision–coalescence, FP16/INT16 may be troublesome. For example, since the mass of aerosol dissolved in droplets has a wide dynamic range (at least $10^9$ from Fig. 1), it is difficult to represent it by FP16 even if scaling is performed by adopting an appropriate unit.

The design of the A64FX architecture also makes the use of FP16 difficult. Since the terminal velocity is calculated based on polynomial fittings from laboratory experiments, and these fittings include measurement errors, the calculations using higher-precision floating-point operations may not improve the accuracy of the results. For such cases, the use of FP16 can be considered. The formula proposed by Beard (1976) for terminal velocity can be divided into three intervals depending on the droplet radius. To apply SIMD vectorization for such loops, we must group particles of similar droplet radii. However, due to a lack of suitable load/store instructions to deal with the jumped data for FP16, a loop cannot be fully vectorized on A64FX. Similarly, the grid fields referenced by each SD are randomly accessed and cannot be vectorized by SIMD. Therefore, we cannot expect faster computation because of the wider SIMD vectorization. We also cannot expect faster computation because of faster data transfer, as the SD movement and condensation/evaporation are not memory-bound computations. These points should be considered when designing computer architecture in the future.

### 6.4.2 Reduction of data movement

For the largest-scale problems, the time for data movement (i.e., other than SD tacking, condensation, and collision–coalescence) in the SDM accounts for 53.9% of the time in the SDM, which accounts for 25.7% of the total elapsed time. To further reduce the time-to-solution, it is necessary to optimize data movement.

One possible optimization is to not to sort SDs with a block as a key for every time step of the SDM. Although such an approach is adopted in the tokamak plasma PIC application (Xiao et al., 2021), it requires some consideration for application to the SDM. For collision–coalescence, all SDs in a block must be in the same MPI process to calculate the interaction between SDs in a cell; however, this is not necessary for SD movement and condensation processes. That is, if the $\Delta t_{\mathrm{coll/coalse}}$ for collision–coalescence process can be taken larger than $\Delta t_{\mathrm{move}}$ for SD movement and $\Delta t_{\mathrm{cond}}$ for condensation, the sorting

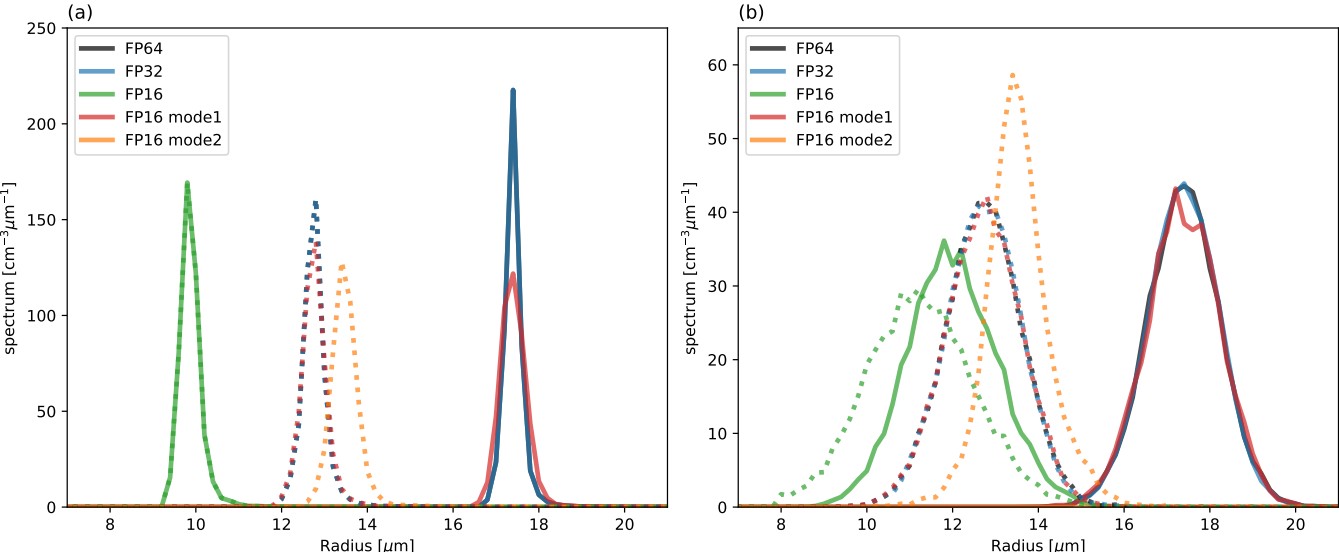

**Figure 11.** DSD obtained by using different numerical precisions of floating-point number operations and rounding modes for (a) without and (b) with the effect of supersaturation fluctuations. The DSD for $t = 500\,\mathrm{s}$ and $t = 1,000\,\mathrm{s}$ are shown by dotted and solid lines, respectively. The DSDs obtained using FP16 mode 2 for $t = 1,000\,\mathrm{s}$ are flat in the range shown in figures. At that time, the mean radius of the DSD are approximately $36\,\mu\mathrm{m}$ for both cases—that is, without and with supersaturation fluctuations. The standard deviation of the DSD are approximately $1.7\,\mu\mathrm{m}$ and $3.0\,\mu\mathrm{m}$ for the cases without and with supersaturation fluctuations, respectively.

1080     frequency can be reduced by $\Delta t_{\mathrm{sort}}$ for sorting equals to $\Delta t_{\mathrm{coll/coalse}}$. In addition, when cloud or rain droplets are not included in a block, collision–coalescence process is not calculated. Then, it is possible to set $\Delta t_{\mathrm{sort}}$ larger than $\Delta t_{\mathrm{move}}$ and $\Delta t_{\mathrm{cond}}$ to reduce the sorting frequency.

      The second possible optimization is to merge the loops divided by subprocesses in microphysics to lower the required B/F of the SDM. However, this approach may be less effective on computers with high B/F, such as A64FX, and it requires a large

1085     amount of L2 cache to store all SDs information in a block.

      From the operations in each subprocess listed in Table 3, the minimum B/F for the SDM is estimated as $\mathrm{BF} = (6 \times 4 \times 2)/(171 + 75.9 + 5.57) = 0.190 < 0.3$ where we assume read/write for six attributes (positions, radius, multiplicity, and aerosol mass) that consist of each four-byte information. On the other hand, if we separate each subprocess and create a working array for two-byte SD cell positions instead of using SD positions, the minimum B/F for SD movement and condensation are

1090     $\mathrm{BF} = (4 \times 3 \times 2 + 4)/171 = 0.164 < 0.3$ (assuming read/write for four byte 3 positions and read for 4 bytes multiplicity), and $\mathrm{BF} = (2 + 4 \times 2 + 4 \times 2)/75.9 = 0.237 < 0.3$ (assuming read for 2 bytes cell position, read for 4 bytes multiplicity and mass of aerosol, and read/write for 4 bytes droplet radius), respectively. These results are consistent with the measured B/F (from speed and memory throughput in Table 3) ($0.189$ and $0.290$, respectively). The minimum B/F for SD movement and condensation are smaller than the B/F of the A64FX. However, since the measured B/F for the SDM and collision–coalescence is $0.833$ and





2.31, respectively, collision–coalescence is a memory-bound computation, and it causes an increase in the level of the total B/F for the SDM.

As is the case with many computers, the B/F values are expected to be smaller in future. For future high-performance computing, merging loops will be necessary, assuming that the high-capacity and high B/F cache or local memory may be achieved by developing new technologies such as 3D stacking.

## 6.5 Possible research directions

Our study focused on the optimization and sophistication of the numerical model. The developed model can be applied to many fields of research from technical and scientific viewpoints.

For model development, in addition to the discussion in Sect. 6.1 to 6.3, the sensitivity of the microphysical variability and precipitation to initialization parameter $\alpha$ should be further explored by high-resolution simulations. The reduction in the variance of prediction for the SDM, such as when using low-discrepancy sequences, should also be explored. We did not examine this impact in this study. Moreover, the continuation of proposal distributions between the DNS and LES may help in realizing more sophisticated model components. The computational performance of our numerical model may be further improved by applying Grabowski et al. (2018). Their method can reduce computational cost only when the cloud volume occupies a small fraction of the total volumes and cannot reduce memory usage unless dynamic load balancing is employed. In contrast, our optimization can improve the performance and reduce memory usage even when the cloud volume occupies a large fraction. Suppose we could further reduce the computational cost and data movement for SD tracking by applying Grabowski et al. (2018). In this case, our model may be more practical than a bulk method in terms of the costs for complex real-world problems because we have already achieved performance comparable to that of a bulk method.

For scientific research, the study enables us to address the problems described in Introduction as $\sim 1\,\mathrm{m}$ resolution numerical experiments are now possible. For example, we can investigate the cloud turbulence structure in shallow cumuli (Hoffmann et al., 2014) and its interaction with boundary layer turbulence (Sato et al., 2017) in detail. We can also confidently compare the simulation results with observational studies (Matsushima et al., 2021) because the effective resolution of simulations is now comparable to the observational scale ($\sim 10\,\mathrm{m}$). We also improved the initialization method. For stratocumulus, we can investigate the statistical quasisteady state DSD, which is affected by cloud-top entrainment and a realistic radiation process.

## 7 Conclusions

In the present study, we developed a particle-based cloud model to perform ultrahigh-resolution simulations to reduce the uncertainty in weather and climate simulations. The SDM is promising for complex microphysical process modeling. The main contributions of our SDM-based work are as follows: (1) the development of an initialization method for SDs that can be used for simulating spatial resolutions between the centimeter and meter scales, (2) improvement of the algorithms of the SDM, and computational and physical performance evaluations, and (3) demonstration of the feasibility for large-scale problems using supercomputer Fugaku.





(1) The uniform sampling method, which has good convergence property for the mass of SDs, results in many invalid samples when the number of SDs is large because the number of SDs becomes larger than the number of real droplets, and multiplicity falls below 1 for rare but important SDs. In contrast, the constant multiplicity method is a natural choice for DNS. We developed a new initialization method that is suitable for scales between the centimeter and meter scales by connecting the uniform sampling method and constant multiplicity method. The developed initialization method requires a proposal distribution apart from the aerosol distribution. The proposal distribution is formulated as an $\alpha$-weighted Fréchet mean of proposal distributions between the uniform sampling method and the constant multiplicity method. To calculate the Fréchet mean, we require a measure of the distance between elements. For this metric, instead of using the $L^2$ norm, we suggest using the Wasserstein distance, which is the natural distance between probability distributions, or the Sinkhorn distance, which is a regularization of Wasserstein distance. The developed method gives a larger minimum and reduces the dynamic range of SD multiplicity. As $\alpha$ decreases, importance sampling for the aerosol size distribution gradually changes from a variance reduction effect for mass concentration to a variance reduction effect for number concentration.

(2) We improved the algorithms of the SDM to achieve high performance on Fujitsu A64FX processor, which is used in supercomputer Fugaku. The developed model employs a hybrid type of 3D and 2D domain decompositions using MPI to reduce communication cost and load imbalance of calculations for the SDM. The SDM, or more generally the PIC method, has a limitation in high-performance computing because such codes include many complex calculation patterns and conditional branches. We further divided the decomposed domain for the cache block into blocks and set the block size with a spatial scale equivalent to the effective resolution of the LES so that the variables within the block were nearly uniform. We converted the conditional branches for each SD, which depends on supersaturation or the presence of clouds, into conditional branches for each block. This conversion improved the ratio of identical instructions for each SD and resulted in parallelization by SIMD vectorization even for Newton iterations and reducing the costs of calculations and data movement for the collision–coalescence process. For SD movement, the 3D CVI of second-order spatial accuracy on the C-grid was derived to guarantee consistency of the SD number density and air density. We subtracted partition information using MPI processes and blocks from the information of SD global positions to reduce information per SD. Then, we stored the relative position of the SD in a block by a fixed-point number using FP32 to guarantee uniform representation precision in the domain.

Next, we evaluated the computational and physical performances of the model on A64FX by comparing the results obtained using SDM-new, two-moment bulk method, bin method, and SDM-orig. The simple warm bubble experiments showed that the time-to-solution obtained by using SDM-new is smaller than that for the bin method with the same number of tracers or SDs per cell, and is comparable to that of the two-moment bulk method when an average of 128 SDs per cell is used. The factors contributing to the enhancements are fewer calculations, higher compactness, and more reasonable use of low-precision arithmetic for SD tracking than for the conventional tracer advection used with the bulk and the bin methods. The data throughput of SDM-new is 57.6 times that of SDM-orig. For the BOMEX and SCMS cases, the computational resources consumed in terms of node hours and energy consumption using the SDM with about 100 SDs per cell are comparable to those consumed using the two-moment bulk method; this is an important result because previous studies showed that the SDM requires about 128 SDs per cell for the convergence of statistics such as the CDNC. For the SCMS, new SCALE-SDM





yielded realistic microphysical variability comparable with that typically observed in nature, including features that cannot be simulated by the two-moment bulk method. As the initialization parameter $\alpha$ decreased, the in-cloud variabilities of CDNC and LWC gradually improved, and they were distributed within their simple adiabatic limits. We confirmed that new SCALE-SDM yields qualitatively better solutions than the two-moment bulk method for a comparable time-to-solution.

(3) Finally, we demonstrated the feasibility of using our approach for simulating large-scale problems using supercomputer Fugaku. The target problem was based on the BOMEX case but with a wider domain and higher spatial resolutions. The new SCALE-SDM exhibited 98 % weak scaling from 256 to 36,864 nodes (23 % of the total system) on Fugaku. For the largest-scale experiment, the horizontal and vertical extents were 13,824 m and 3,072 m covered with 2 m grids, respectively, and 128 SDs per cell were initialized on average. The time integration was performed for about 1 h. This experiment required about 104 and 442 times the number of grids and SDs compared to the current state-of-the-art experiment (Sato et al., 2018). The overall calculations achieved 7.97 PFLOPS (7.04 % of the peak), and the maximum performance was 18.2 PFLOPS (16.1 % of the peak) for the condensation process in the SDM. The overall throughput in the SDM was $2.86 \times 10^{13}$ particle·step/s. These results are comparable to those reported by the recent Gordon Bell prize finalists, such as the peak ratio of the simulation part of the NICAM-LETKF and the particle throughput of the tokamak plasma PIC simulation. We did not examine the largest-scale problem by using the bin model or the two-moment bulk model; instead, we used a simple extrapolation to estimate that for the largest problem, the time-to-simulation of the SDM is shorter than that of the bin method and is comparable to that of the two-moment bulk method.

Several challenges remain—for example, optimization for mixed-phase clouds, inclusion of terrain, and long-time integration. However, our approach can handle such further sophistication. The simplification of a loop body innovated in this study can contribute to optimizing the mixed-phase SDM. We also discussed the possibility of reducing attributes, which increases when using mixed-phase SDM, to obtain effective attributes. However, our approach cannot simply be applied to improve the computational performance when the water and ice states are both present in a cell. Thus, further sophistication is necessary. The developed CVI scheme can be applied to cases with terrain if we combine our algorithm with the immersed boundary or cut-cell methods. The computational performance of our model will not be degraded in such cases. However, SD tracking over a larger area and in spherical coordinates remains a challenge. The long-time integration of SCALE-SDM is still difficult because of the large data volume. Additional study on reducing data volume by using lossy compression and resampling to restore the data is necessary. For future supercomputers, reducing data movement will be the key to achieving high computational performance. This can be achieved, for example, by reducing information on SD positions, reducing the SD sorting frequency, lowering the application B/F by merging the loops for physics subprocesses, and developing computers that will make this possible.

Our study is still in the stage of demonstrating the feasibility of large-scale problems for ultrahigh-resolution simulations. However, suppose the ultrahigh-resolution cloud simulations demonstrated in this study can be performed routinely. In this case, these results can be compared with DNS, laboratory experiments, and field studies to study turbulence and microphysics processes over a vast range of scales. Therefore, we strongly believe that our approach is a critical building block of the future





cloud microphysics models and advances the scientific understanding of clouds and contributes to reducing the uncertainties of weather simulation and climate projection.

*Code and data availability.* The numerical model codes and configuration files used in this study are available at Matsushima et al. (2023a). The supplemental codes, figures, movies, and datasets are available at Matsushima et al. (2023b).

## Appendix A: Wasserstein distance

As described in Sect. 3.2, the Wasserstein distance was used to develop a new initialization method for the SDM. The Wasserstein distance and its strongly related optimal transport theory are powerful mathematical tools for tackling problems dealing with a probability distribution, such as machine learning. Here, we briefly introduce the Wasserstein distance, its regularization, and displacement interpolation (McCann, 1997) for readers who are unfamiliar with them.

Let two probability distributions as $\boldsymbol{a}$ and $\boldsymbol{b}$. If we allow mass split during transportation, the amount of transportation from $i$-th bin $a_i$ to $j$-th bin $b_j$ is represented using a coupling matrix $P_{ij}$. Let a set of coupling matrix $\boldsymbol{U}$ as

$$\boldsymbol{U}(\boldsymbol{a}, \boldsymbol{b}) =$$

$$\left\{ \boldsymbol{P} \in R^{n \times n} : P_{ij} \geq 0, \sum_j P_{ij} = a_i, \sum_i P_{ij} = b_j \right\}. \tag{A1}$$

The $p$th $(p \geq 1)$ Wasserstein distance $W_p$ for two probability density distributions $(\boldsymbol{a}, \boldsymbol{b})$ is defined as

$$W_p(\boldsymbol{a}, \boldsymbol{b}) = \left( \min_{P \in \boldsymbol{U}(\boldsymbol{a}, \boldsymbol{b})} \sum_{i,j} |i - j|^p P_{ij} \right)^{\frac{1}{p}}. \tag{A2}$$

That is, $W_p^p$ is the minimum total cost of transportation from $\boldsymbol{a}$ to $\boldsymbol{b}$ when transport cost from $i$ to $j$ is $|i - j|^p$. On the other hand, the difference between two distributions are often measured using $L^p$ norm:

$$L^p(\boldsymbol{a}, \boldsymbol{b}) = \sum_i (a_i - b_i)^p. \tag{A3}$$

The significant difference between the Wasserstein distance and $L^p$ norm is that distance between two distributions is measured in terms of horizontal or vertical differences. Therefore, the Wasserstein distance is a useful measure if the location of the random variable is essential. A coupling matrix $\boldsymbol{P}$ can be obtained by solving a linear programming problem, which is computationally expensive for large-scale problems because its computational complexity is of the order of $O(N^3)$ for $N$ dimension. If the computational cost is important, the Sinkhorn distance (Cuturi, 2013), which is a regularization of the Wasserstein distance, can be used instead:

$$S_\gamma(\boldsymbol{a}, \boldsymbol{b}) = \min_{P \in \boldsymbol{U}(\boldsymbol{a}, \boldsymbol{b})} \left[ \sum_{i,j} |i - j|^p P_{ij} + \gamma \sum_{i,j} P_{ij} (\log P_{i,j} - 1) \right]. \tag{A4}$$





The negative sign of the second term on the right-hand side is the entropy of the probability distribution, which is non-negative and increases with the uncertainty.

For the one-dimensional case, the Wasserstein distance has a simple alternative form:

$$W_p(\boldsymbol{a}, \boldsymbol{b}) = \left( \int\limits_0^1 |F_a^{-1}(y) - F_b^{-1}|^p dy \right)^{\frac{1}{p}}, \tag{A5}$$

where $F_a^{-1}$, and $F_b^{-1}$ are quantile functions (inverse functions of the cumulative function) for $\boldsymbol{a}$ and $\boldsymbol{b}$, respectively. In this case, the displacement interpolation (a solution of continuous case of Eq. (5)) is represented as

$$F_a^{-1}(y) = (1 - \alpha) F_{b_1}^{-1}(y) + \alpha F_{b_2}^{-1}(y). \tag{A6}$$

When we denote right-hand side of Eq. (A6) as

$$x = (1 - \alpha) F_{b_1}^{-1}(y) + \alpha F_{b_2}^{-1}(y), \tag{A7}$$

then Eq. (A6) is rewritten as

$$F_a^{-1}(y) = x. \tag{A8}$$

Here, we describe a method to obtain $y = F_a(x)$, assuming we already know the specific forms of $F_{b_1}, F_{b_2}$, and $F_{b_2}^{-1}$. We change the variable from $y$ to $x'$ in Eq. (A7) as $y = F_{b_1}(x')$ and we get

$$x = (1 - \alpha) x' + \alpha F_{b_2}^{-1} F_{b_1}(x'). \tag{A9}$$

This means that if we assign a value to $x'$, we can obtain a function of $x$ as $y = F_a(x)$. The simple discretization of these calculations yields practical numerical algorithms to obtain $y = F_a(x)$. For example, in this study, we assign $\boldsymbol{b}_1$ as the normalized aerosol distribution and $\boldsymbol{b}_2$ as the uniform distribution. Because $\boldsymbol{b}_1$ is close to $0$ near the edge of the support for the distribution and because the quantile function of $\boldsymbol{b}_1$ changes sharply, it is difficult to construct discrete points in $y$ directly. However, if we discretize $x'$ using equidistant points, the points in $y$ are automatically ensured to resolve the sharp changes in the quantile function.

## Appendix B: Second-order conservative velocity interpolation on Arakawa C-grid

For simplicity, we considered interpolation within a cell, and let the coordinates be $(x, y, z)$ and let the regions be $0 \le x \le \Delta x$, $0 \le y \le \Delta y$, $0 \le z \le \Delta z$. Coordinates and velocities are nondimensionalized as follows.

$$x' = \frac{x}{\Delta x},\, y' = \frac{y}{\Delta y},\, z' = \frac{z}{\Delta z}, \tag{B1}$$

$$u' = \frac{u \Delta t}{\Delta x},\, v' = \frac{v \Delta t}{\Delta y},\, w' = \frac{w \Delta t}{\Delta z}. \tag{B2}$$





In the following discussion, $'$ is omitted, and only the results are shown (the proof is available at Matsushima et al. (2023b)).

Let $u(x,y,z),\ v(x,y,z),\ w(x,y,z)$ be the nondimensional velocities, and let its values on the C-grid be represented as follows:

$$u_0 = u(0,1/2,1/2), u_1 = u(1,1/2,1/2), \tag{B3}$$

$$v_0 = v(1/2,0,1/2), v_1 = v(1/2,1,1/2), \tag{B4}$$

$$w_0 = v(1/2,1/2,0), w_1 = w(1/2,1/2,0). \tag{B5}$$

Further, let the partial differential coefficient for the nondimensional velocities be represented as

$$\delta_y u_0 = \frac{\partial u}{\partial y}(0,1/2,1/2). \tag{B6}$$

Then, the velocity at the SD position $\boldsymbol{U} = (u_p, v_p, w_p)$ obtained using the second-order conservative velocity interpolation is represented as follows:

$$u_{f\{0,1\}} = u_{\{0,1\}} + \delta_y u_{\{0,1\}}\left(y - \frac{1}{2}\right) + \delta_z u_{\{0,1\}}\left(z - \frac{1}{2}\right), \tag{B7}$$

$$v_{f\{0,1\}} = v_{\{0,1\}} + \delta_x v_{\{0,1\}}\left(x - \frac{1}{2}\right) + \delta_z v_{\{0,1\}}\left(z - \frac{1}{2}\right), \tag{B8}$$

$$w_{f\{0,1\}} = w_{\{0,1\}} + \delta_x w_{\{0,1\}}\left(x - \frac{1}{2}\right) + \delta_y w_{\{0,1\}}\left(y - \frac{1}{2}\right), \tag{B9}$$

$$u_p = (1-x)u_{f0} + xu_{f1}$$
$$+ x(1-x)\left\{\frac{1}{2}(\delta_x w_1 - \delta_x w_0) + \frac{1}{2}(\delta_x v_1 - \delta_x v_0)\right\}, \tag{B10}$$

$$v_p = (1-y)v_{f0} + yv_{f1}$$
$$+ y(1-y)\left\{\frac{1}{2}(\delta_y w_1 - \delta_y w_0) + \frac{1}{2}(\delta_y u_1 - \delta_y u_0)\right\}, \tag{B11}$$

$$w_p = (1-z)w_{f0} + zw_{f1}$$
$$+ z(1-z)\left\{\frac{1}{2}(\delta_z v_1 - \delta_z v_0) + \frac{1}{2}(\delta_z u_1 - \delta_z u_0)\right\}. \tag{B12}$$

If all partial differential coefficients in Eq. (B7)–(B12) are set as $0$, the interpolated velocity becomes identical to the results obtained using the first-order conservative velocity interpolation. The coefficients are evaluated simply by calculating the second-order central difference from the velocities at the cell boundaries.

## Appendix C: Conditions for existence and uniqueness of the solutions of discretized activation/condensation equation

To solve Eq. (7) numerically, we consider two cases in which the uniqueness of the solution can be easily determined. Here, $f$ is continuous function of $R^2$ in the interval $R^2 \in (0, \infty)$, and it behaves as $f(+0) = -\infty$ and $f(+\infty) = \infty$. The intermediate value theorem states that Eq. (7) has at least one solution in the interval $(0, \infty)$.


To derive the Case 1 condition, we first differentiate $f$ with respect to $R^2$:

$$f'(R^2) = 1 - \frac{\Delta t A}{(R^2)^{3/2}} \left[ a - \frac{3b}{R^2} \right], \tag{C1}$$

$$f''(R^2) = \frac{3\Delta t A}{2} \frac{1}{(R^2)^{5/2}} \left[ a - \frac{5b}{R^2} \right]. \tag{C2}$$

Since $f'$ has a minimum value at $\alpha^2 = 5b/a$ where $(f')' = 0$, $f'$ is always positive in $R^2 \in (0, \infty)$ if $f'(\alpha^2) > 0$. In this case, there is one unique solution in the interval. From $f'(\alpha^2) \geq 0$, we obtain Case 1 condition of Eq. (8). On the other hand, the solution for $f = 0$ has at most three solutions if $f'(\alpha^2) < 0$, and one or two of them may not be physical solutions. Our purpose is neither to find sufficient conditions for the uniqueness of solutions nor to discriminate physical solutions from at most three solutions. Although Eq. (8) is a more stringent condition than the condition for the uniqueness of solutions, it has the advantage that Newton's method becomes more stable because $f'$ is always positive.

Case 2 condition is obtained when we constrain the initial values and environmental conditions. We consider the interval $0 < R^2 \leq 3b/a$ where $f$ behaves as $f'(R^2) \geq 1$ and $f(+0) = -\infty$. The intermediate value theorem states that Eq. (7) has the unique solution in the interval if $f(3b/a) > 0$:

$$f\left(\frac{3b}{a}\right) = \frac{3b}{a} - p^2 - 2\Delta t A \left[ S - 1 - \frac{2a}{3}\sqrt{\frac{a}{3b}} \right]. \tag{C3}$$

If we give $S - 1 \leq 2a\sqrt{a}/(3\sqrt{3b})$, then $f(3b/a) \geq 3b/a - p^2$. Therefore, the condition $f(3b/a) > 0$ is met if $p^2 < 3b/a$. Since $b$ depends on an attribute of the droplets, we can make the condition more stringent to depend on only a variable at a cell. For an unsaturated environment, $S - 1 \leq 0$ and $p^2 < 3b/a$.

*Author contributions.* All authors contributed to conceiving the idea of this study. TM developed the numerical model with assistance from SN and SS, performed numerical experiments, analyzed the results, and drafted the original manuscript. All authors reviewed and revised the manuscript and approved the final version for submission.

*Competing interests.* The authors declare no conflicts of interest associated with this article.

*Acknowledgements.* TM acknowledges the members of the Center for Planetary Science at Kobe University for encouraging this study. This work was supported by JSPS KAKENHI, Grant Number 20K14559. SN was supported by JSPS KAKENHI, Grant Number 19H01974. SS was supported by JSPS KAKENHI, Grant Number 20H00225; and JST [Moonshot R & D][Grant Number JPMJMS2286]. The numerical experiments in this study were conducted using the supercomputer Fugaku (Project ID: ra000005) in RIKEN R-CCS, FUJITSU PRIMEHPC FX1000 and PRIMERGY GX2570 (Wisteria/BDEC-01) at the Information Technology Center, The University of Tokyo. The authors used facilities of the Center for Cooperative Work on Data science and Computational science, University of Hyogo, for data server and analysis. The authors would like to thank Enago (www.enago.jp) for the English language review.



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
