# Peer review of "Optimization and sophistication of the super-droplet method for ultrahigh resolution cloud simulations"

_Geoscientific Model Development, 2023_

## Referee Comment (RC1)

**Review on article gmd-2023-26 with title "Optimization and sophistication of the super-droplet method for ultrahigh resolution cloud simulations"**

This paper presents a new implementation of the superdroplet method (SDM) to reach centimeter-to-meter scale resolutions in cloud macrophyiscs simulations. The authors discuss general aspects of how to represent microphyiscal processes in the SDM framework and the corresponding algorithms. In particular, the authors discuss the initialization, the SD movement, and the use of mixed precision in some parts of the implementation. Besides this general aspects, the authors also present the particular implementation, validation and scaling analysis in the supercomputer Fugaku, in particular the use of cache. The test case is shallow cumuli cases, and the new SDM implementation is compared with bin and bulk methods. The new SDM model is comparable and even faster that these methods, and it shows a weak-scaling of 98% up to 36864 nodes, which demonstrates the potential of SDM to advance research and simulation of microphysics processes and its coupling to cloud macrophysics.

I think the paper is very interesting. This work addresses the important topic of what are the current and near-future capabilities to study the interactions between cloud microphysics and cloud macrophysics, and thereby advance the understanding of the role of clouds in climate and weather. The discussion is thoroughly presented, and I think this is a good contribution to the community and this journal. I only have a few minor comments that the authors could consider before publication:

1. Maybe the only general comment is whether the authors could reduce the length of the manuscript, or consider a more clear separation between the discussion of general aspects of the SDM and the particularities of the implementation in Fugaku, as I tried to do in the first paragraph of this letter. I wonder if section 6 could be shorten. As said, this is a minor point and I realize it is difficult, but somehow I think the reader might be better guided through the paper to concentrate on the aspects that might be more interesting for her or for him.

   I found the conclusions, however, very well and they helped me to end the paper with a good idea of the various parts.

2. Title, abstract, line 1, and through the paper, the authors use the term *"ultrahigh-resolution"* and I was wondering if they could substitute that term by another one that is more informative. It seems that, in line 9, the authors refer explicitly to centimeter to meter scale, so why not say "centimeter-to-meter scale resolution" or "submeter resolution"?

   Otherwise, what would come after "ultrahigh resolution" when we reach the following step towards higher resolution?

3. Title, abstract, line 5, and through the paper, the authors use the term *"sophisticated"*. I was not sure what it means. Does it mean that it considers more physical processes, or a better model of them, or does it refer to the technical implementation from a computer science point of view?

4. abstract, line 4, the authors say *"does not make any assumption for the droplet size distribution"*. Since some aspects of the dynamics within the superdroplet or the interaction between superdroplets are still modeled, as discusses in section 2 and 3, I wonder if it might be better to say "makes less assumptions about the droplet size distribution and it is more physically sounded", or something similar.

5. Abstract, line 18: instead of *"perfect weak scaling"*, it might be stronger and clearer to say 98% weak scaling up to the corresponding number of nodes or cores, as it is done in the conclusions.

6. Introduction, line 35, the authors refer to Schulz and Mellado, 2018, but the reference Schulz and Mellado [2019] might be stronger for their case. Schulz and Mellado [2019] studies one micro-physical effect, namely, sedimentation, and shows that sedimentation is more important than previously thought, which strongly supports the efforts presented in this SDM manuscript to better represent the DSD.

7. Introduction, line 34 say *"which sets the eddy viscosity constant"* referring to DNS. I wonder if this sentence is needed. This seems to suggest that DNS is one type of LES, which is not the way DNS is used in turbulence research, where the concept originates from, since there is no eddy viscosity in DNS [Orszag and Patterson, 1972, Moin and Mahesh, 1998, Pope, 2000, Mellado et al., 2018]. DNS rescales the original in terms of size or in terms of physical properties to study Reynolds number effects, and remains accurate in the smallest resolved scales, which LES does not.

8. In several places through the manuscript, the authors refer to particle-in-cell (PIC) methods. What are the differences between the superdroplet method (SDM) and particle-in-cell methods?

9. Section 2, line 118, the authors write *"the anelastic equations assume horizontally uniform mean fields and are not appropriate for computing wider domains"*. I wonder if this sentence is needed. The mean fields need not be the reference fields that are used in the anelastic formulation, and one can have anelastic formulations of statistically inhomogeneous flows (a cloud bubble).

10. Equation (1), what are the assumptions for using this equation? The same applies for instance to the description of collision-coalescence and relates to point 3 before, namely, that there are still some assumptions in the SDM and it might be convenient to explain them as clearly as possible to better interpret the results and the limits of applicability.

11. Section 3.3.1, *"The effective resolution is $6\Delta - 10\Delta$ for planetary boundary layer turbulence"* I assume that this is for the second-order methods that the authors use, but they might indicate it explicitly here because the effective resolution might be substantially smaller in pseudo-spectral schemes or similar, which are often used in boundary-layer meteorology [Sullivan and Patton, 2011, Pope, 2000].

12. In section 6.4.1, I was wondering how much computational time or memory save is gained by using FP32 instead of FP64 in that part of the microphysics. I was trying to have a sense of priorities in addressing the standing challenges. For instance, the disk space challenge indicated in section 6.3 seems most important, but I might be wrong.

13. Conclusions, line 1129 *"In contrast, the constant multiplicity method is a natural choice for DNS"*. I think I did not understand this sentenc.

14. I think the conclusions do not refer to section 3.3.2 "Super-droplet movement", which I thought was interesting in general, not only for the particular implementation described here.

**References**

J. P. Mellado, C. S. Bretherton, B. Stevens, and M. C. Wyant. DNS and LES of stratocumulus: Better together. *J. Adv. Model. Earth Syst.*, 10:1421–1438, 2018.

P. Moin and K. Mahesh. Direct numerical simulation: A tool in turbulence research. *Annu. Rev. Fluid Mech.*, 30:539–578, 1998.

S. A. Orszag and G. S. Patterson. Numerical simulation of three-dimensional homogeneous isotropic turbulence. *Phys. Rev. Lett.*, 28(2):76–79, 1972.

S. B. Pope. *Turbulent Flows.* Cambridge University Press, 2000.

B. Schulz and J. P. Mellado. Competing effects of wind shear and droplet sedimentation at stratocumulus tops. *J. Adv. Model. Earth Syst.*, 2019. doi: doi:10.1029/2019MS001617.

P. P. Sullivan and E. G. Patton. The effect of mesh resolution on convective boundary layer statistics and structures generated by large-eddy simulations. *J. Atmos. Sci.*, 68:2395–2415, 2011.

---

## Author Comment (AC1)

**Reply to the comments of Referee #1 (RC1) on article gmd-2023-26**

Toshiki Matsushima, Seiya Nishizawa, Shin-ichiro Shima

Thank you for taking the time to read through our manuscript and provide us with positive and helpful comments. Below are our point-by-point responses to your comments.

1. Maybe the only general comment is whether the authors could reduce the length of the manuscript, or consider a more clear separation between the discussion of general aspects of the SDM and the particularities of the implementation in Fugaku, as I tried to do in the first paragraph of this letter. I wonder if section 6 could be shorten. As said, this is a minor point and I realize it is difficult, but somehow I think the reader might be better guided through the paper to concentrate on the aspects that might be more interesting for her or for him.
I found the conclusions, however, very well and they helped me to end the paper with a good idea of the various parts.

    Thank you very much for your suggestion. We have simplified the Discussion (Section 6) and other sections as much as possible for clarity and to reduce redundancy.

2. Title, abstract, line 1, and through the paper, the authors use the term "ultrahigh-resolution" and I was wondering if they could substitute that term by another one that is more informative. It seems that, in line 9, the authors refer explicitly to centimeter to meter scale, so why not say "centimeter-to-meter scale resolution" or " submeter resolution" ?
Otherwise, what would come after *"ultrahigh resolution"* when we reach the following step towards higher resolution?

    As you suggested, we have revised the term to "meter-to-submeter scale."

3. Title, abstract, line 5, and through the paper, the authors use the term *"sophisticated"*. I was not sure what it means. Does it mean that it considers more physical processes, or a better model of them, or does it refer to the technical implementation from a computer science point of view?

    Thank you for your comment. In the revised manuscript, we have considered words that improve clarity. As other papers have also used the term "sophisticated" microphysics model to describe a model that introduces a more complex process, we have retained it in the revised manuscript. Alternatively, our study focused on improving many aspects, such as numerical precision, algorithms, and computational performance. Thus, we have made some revisions to clearly state that.

    In addition, based on your suggestion, we have revised the title of the manuscript to "Overcoming computational challenges to enable meter-to-submeter-scale resolution cloud simulations using the super-droplet method"

4. abstract, line 4, the authors say *"does not make any assumption for the droplet size distribution"*. Since some aspects of the dynamics within the superdroplet or the interaction between superdroplets are still modeled, as discusses in section 2 and 3, I wonder if it might be better to say "makes less assumptions about the droplet size distribution and it is more physically sounded", or something similar.

    As suggested, we have revised the statement to "makes fewer assumptions for the droplet size distribution."

5. Abstract, line 18: instead of *"perfect weak scaling"*, it might be stronger and clearer to say 98% weak scaling up to the corresponding number of nodes or cores, as it is done in the conclusions.

    Thank you for your suggestion. We have revised the phrase to "98 % weak scaling" in the revised manuscript.

6. Introduction, line 35, the authors refer to Schulz and Mellado, 2018, but the reference Schulz and Mellado [2019] might be stronger for their case. Schulz and Mellado [2019] studies one micro-physical effect, namely, sedimentation, and shows that sedimentation is more important than previously thought, which strongly supports the efforts presented in this SDM manuscript to better represent the DSD.

    Thank you for the suggested reference. We have cited Schulz and Mellado (2019) instead of Schulz and Mellado (2018). We have also added the following statement to L35: Schulz and Mellado (2019) investigated the joint effect

of droplet sedimentation and wind shear on cloud top entrainment and found that their effects are equally important.

7. Introduction, line 34 say *"which sets the eddy viscosity constant"* referring to DNS. I wonder if this sentence is needed. This seems to suggest that DNS is one type of LES, which is not the way DNS is used in turbulence research, where the concept originates from, since there is no eddy viscosity in DNS [Orszag and Patterson, 1972, Moin and Mahesh, 1998, Pope, 2000, Mellado et al., 2018]. DNS rescales the original in terms of size or in terms of physical properties to study Reynolds number effects, and remains accurate in the smallest resolved scales, which LES does not.

> Thank you for highlighting this. In the revised manuscript, we have explained DNS and LES, based on your comment and Mellado et al. (2018).
>
> DNS solves the original Navier–Stokes equation but changes only the kinematic viscosity (or Reynolds number) among the atmospheric parameters.
>
> LES solves low-pass filtered Navier–Stokes equation for unresolved flow below filter length.

8. In several places through the manuscript, the authors refer to particle-in-cell (PIC) methods. What are the differences between the superdroplet method (SDM) and particle-in-cell methods?

> Thank you for your comment. The PIC method is used to solve a specific type of partial differential equation, which describes a coupled system of particles and cell-averaged variables. SDM can be regarded as an application of the PIC method to solve cloud microphysics and macrophysics. In the original manuscript, we did not introduce an abbreviation for the super-droplet method, but in the revised manuscript, we have defined the abbreviation properly and have defined the PIC method and its relation with SDM more clearly.
>
> added to L42: In this study, we focus on the super-droplet method (SDM) as one of the particle-based Lagrangian cloud microphysics schemes.
>
> added to L72: In general, the method of solving partial differential equations, which describes a coupled system of particles and cell-averaged variables, is called the particle-in-cell (PIC) method. The SDM can be regarded as an application of the PIC method to solve cloud microphysics and macrophysics.
>
> simplified L72: Although such dynamic load balancing is adopted in some plasma simulations (Nakashima et al. 2009) when the PIC method is applied, dynamic load balancing is not a good option for weather and climate models.

9. Section 2, line 118, the authors write *"the anelastic equations assume horizontally uniform mean fields and are not appropriate for computing wider domains"*. I wonder if this sentence is needed. The mean fields need not be the reference fields that are used in the anelastic formulation, and one can have anelastic formulations of statistically inhomogeneous flows (a cloud bubble).

> Thank you for your comment. We agree that the description of anelastic equations in Section 2 is redundant and confusing. In the revised manuscript, we have clearly stated the differences between fully compressible and anelastic equations.

10. Equation (1), what are the assumptions for using this equation? The same applies for instance to the description of collision-coalescence and relates to point 3 before, namely, that there are still some assumptions in the SDM and it might be convenient to explain them as clearly as possible to better interpret the results and the limits of applicability.

> In the revised manuscript, we have stated the process ignored and assumptions made for microphysics to be solved explicitly. We added the phrases in blue, as follows:
>
> L132: In this study, only the following warm cloud processes were considered: movement, activation/deactivation and condensation/evaporation, and collision–coalescence. Spontaneous and collisional breakup process were not considered in this study.
>
> L136: The $i$th SD moves according to the wind and falls with terminal velocity assuming that the velocity of each SD reaches the terminal velocity instantaneously.
>
> L146: The terms $a/R$ and $b/R^3$ represent the curvature and solute effects, respectively. The ventilation effect is ignored in Eq. (2).

11. Section 3.3.1, *"The effective resolution is $6\Delta - 10\Delta$ for planetary boundary layer turbulence"* I assume that this is for the second-order methods that the authors use, but they might indicate it explicitly here because the effective resolution might be substantially smaller in pseudo-spectral schemes or similar, which are often used in boundary-layer meteorology

[Sullivan and Patton, 2011, Pope, 2000].

> Thank you for your comment. Second-order spatial discretization for the pressure gradient was employed in this study. Thus, the dynamical core used is (overall) second-order spatial accuracy. However, as discussed by Nishizawa et al. (2015), the pressure gradient term is mainly for fast acoustic waves and has less effect on slow modes, such as mixing, i.e., energy spectrum. Thus, we believe that our simulations have a certain degree of numerical accuracy. As you highlighted, large-eddy simulations using the (pseudo) spectral method are numerically more accurate and might have an advantage in effective resolution because explicit filtering can be used. This has been stated in the revised manuscript.

12. In section 6.4.1, I was wondering how much computational time or memory save is gained by using FP32 instead of FP64 in that part of the microphysics. I was trying to have a sense of priorities in addressing the standing challenges. For instance, the disk space challenge indicated in section 6.3 seems most important, but I might be wrong.

> In implementing SDM, most of the information about SDM is the arrays for SD attributes because other information, such as intermediate values of SD position, which is required for the second-order Runge-Kutta method (Heun's method) for SD tracking, is mainly stored in a local stack in each OpenMP thread, and because the size of the working arrays, such as for SD sorting is small, as discussed in Section 3.3.5. The amount of memory that SDM uses is estimated as
>
> meshes $\times$ SDs per cell $\times$ attributes $\times$ information of each attribute $\times$ (1+extbuf);
>
> here, extbuf is the ratio of the extra buffer arrays to the number of SDs. Thus, the amount of information would be halved if information on all attributes is halved. This applies if we reduce the numerical precision from double to single precision. Alternatively, the computational cost is a bit more complicated, for example, for condensation/evaporation and activation/deactivation. The number of iterations for Newton's method would likely change if the tolerance relative error is squared. We speculate that the number of iterations increases by 1 or 2 because Newton's method is quadratically convergent.
>
> In the case of SCMS, the elapsed time only for microphysics using four nodes of FX1000 and with FP64 was 248(349) min without (with) collision-coalescence calculations in non-cloudy volumes. The memory for only microphysics was 54.9 GB. On the other hand, the elapsed time with FP32 was 150(274) min, and the memory was 29.7 GB. The computational-cost reduction on collision–coalescence is insignificant because the amount of information of a key for SD sorting did not change when we used FP64 to FP32, and the multiplicity remained INT64 due to a small number of SDs. In the revised manuscript, we have added the above discussion to L811.
>
> The problem for checkpoint/restart files can be avoided by consulting the computer center, and we have modified this in the revised manuscript. However, we believe that it is an important subject, and a data-scientific approach may help mitigate it.

13. Conclusions, line 1129 *"In contrast, the constant multiplicity method is a natural choice for DNS"*. I think I did not understand this sentenc.

> Thank you for your suggestion. We have removed this sentence.

14. I think the conclusions do not refer to section 3.3.2 "Super-droplet movement", which I thought was interesting in general, not only for the particular implementation described here.

> We are glad to hear that you are interested in our manuscript. In the original manuscript, we summarized Section 3.3.2 at the end of the first paragraph in the second item. In the revised manuscript, we have added the following: For SD movement, to maintain many SDs within convective clouds and reduce the nonphysical variance of SDs, the 3D CVI of the second-order spatial accuracy on the C-grid was derived to ensure consistency between SD number and air densities. Then, we stored the relative position of SD in a block by a fixed-point number using FP32 to ensure a uniform representation precision in the domain, which keeps the numerical representation precision for meter-to-submeter resolution while using a low-precision format.

**References**

Schulz, B. and Mellado, J. P.: Competing effects of droplet sedimentation and wind shear on entrainment in stratocumulus, J. Adv. Model. Earth Syst., 11, 1830–1846, https://doi.org/10.1029/2019MS001617, 2019.

---

## Author Comment (AC2)

**Reply to the comments of Referee #2 (RC2) on article gmd-2023-26**

Toshiki Matsushima, Seiya Nishizawa, Shin-ichiro Shima

Thank you for taking the time to read through our manuscript and provide us with positive and helpful comments. Below are our point-by-point responses to your comments.

**general comments**

Most of the model improvements are of "technical" nature. I am not computer scientist, hence I cannot completely judge these achievements.

Still, the performance achievements seem great. However, I would have liked it even better, if the comparison with traditional bulk method would be made fairer. E.g. in the SDM and bin model there is no "cold" microphysics (yet), but the bulk method contains these tracers. Hence, it is likely that the advection cost could be reduced to 4/10. Or in the time to solution comparison there is no discussion that with the bulk methods larger microphysical timesteps are possible and might be even sufficient to reach convergence. Considering that the SDM-new128 might be more costly than the bulk method. However, I still would want to apply it, as it makes, as stated in the paper, far less assumptions.

> Thank you for the compliment on the performance improvement we have made. We agree with you that the comparison between the mixed-phase microphysics bulk model and SDM is not fair, but it is appropriate in the specific case discussed in this study. We did not use warm microphysics because the original codes need to be modified. Thus, we decided to exclude cold microphysics from Seiki and Nakajima (2014) scheme, performed additional experiments, and compared the results with those of SDM. Nonetheless, we would like to retain the results for mixed-phase microphysics for discussion. Before showing how it changed the results herein, we share our thoughts on comparing SDM and Seiki and Nakajima (2014) scheme. There are two aspects of convergence for the two-moment bulk method; spatial and temporal resolution. The two-moment bulk method imposes empirical assumptions on DSD, which may lead to less spatial variability or no dependency on spatial resolution, such as the spectral width of DSD. However, this does not mean it converges fast to increase spatial resolution; rather, it indicates that, in principle, fair comparison in terms of spatial resolution is difficult. For the temporal resolution, the time step for the two-moment bulk method could be fairly large (e.g., 5 min) in climate simulations. Based on this, one may assume that the two-moment bulk method can solve groups of droplets using large time steps by bundling the fast time evolution of individual droplets. However, as discussed in Santos et al. (2020), this is not the case. The authors performed eigenvalue analysis for a two-moment bulk scheme and found that a fast mode (<1 s) also exists in the bulk scheme. It does not considerably deviate from the time step determined by the evolution of individual droplet dynamics, such as condensation/evaporation. Thus, it is reasonable for convergence to use a sufficient time step that is determined by eigenvalues, instead of a stable time step. Alternatively, large time steps can be considered if it is stable and the mean climate does not change for earth and planetary climate simulations. For such a case, we agree that a bulk method is suitable. Based on the above discussion, we would like to maintain the same spatial and temporal resolution to compare SDM and the two-moment bulk method.
>
> As explained later, we modified the initial condition of the mass mixing ratio of water vapor (QV) for (only) warm bubble experiments using SDM, the two-moment bulk method, and the bin method. In addition, we have modified the evaporation calculations for the two-moment bulk method. We have also added the results using the bin method with a stochastic collision–coalescence algorithm developed by Sato et al. (2009). Figures 1 and 2 in this letter correspond to Figures 3 and 4 in the original manuscript. The differences are small for warm bubble cases. The elapsed time using the warm microphysics two-moment bulk method is still comparable to that of SDM-128. The description in the revised manuscript

is modified as follows (shown in blue):

L675: The maximum throughputs of SDM-new are 61.3 and 20.1 times that of SDM-orig and two-moment bulk method, respectively.

L684: The elapsed time obtained using the two-moment bulk method was 14.0 s, and that obtained using SDM-new with 128 SDs per cell on average is 13.9 s. The maximum throughput of SDM-new is 31.6 times that of the two-moment bulk method.

Figure 3 in this letter shows the results of BOMEX and SCMS case experiments, corresponding to Figure 7 in the original manuscript. In this case, the elapsed time and energy consumption obtained using warm microphysics two-moment bulk scheme are comparable to that of the results between SDM-new32 and SDM-new64. However, as discussed in the revised manuscript, as the elapsed time for SDM increased more gradually than linearly with the number of SDs when 8-128 SDs/cell were used, the main conclusion did not change considerably. Based on the above discussion, we have changed the conclusion that the elapsed time for SDM-new32 to SDM-new128 is comparable to that of the (warm microphysics) two-moment bulk method.

[Figure]

**Figure 1.** Elapsed times of the total (circles) and tracer advection and SD tracking (squares) using the two-moment bulk method (green), bin method (blue), SDM-orig (yellow), and SDM-new (red) with different numbers of tracers or mean SDs per cell. Elapsed times (triangles) of the total using the bin method with stochastic collision–coalescence algorithms are also shown. Here, SD tracking includes SD movement and sorting with a block as a key. The blue dotted line is proportional to $N^2$, the red- and yellow-dotted lines are proportional to $N$, and the green-dotted line indicates a constant determined by $N$.

It is quite a long paper. Consider whether it can be shortened. Personally, I tend to read and cite paper more often, if they are briefly and succinctly. To provide examples, I would not miss the following parts in main body of the paper:

> We thank you for kindly reading through our long paper. Below are our responses to the specific examples you provided for us. For other parts, we would like to simplify them as much as possible based on the comments from two referees.

- discussion about the flow solver, i.e. focus on the microphysics (e.g. microphysics - radiation coupling is also not included only mentioned in L914. It is an interesting question how to couple SDM with radiation but clearly out of the scope of this paper). Of course there are a few details that would need to kept, like that a C grid is used and the effective resolution.

> We agree with your comment. In the revised manuscript, we have simplified the description of anelastic equations in L111, flux-corrected transport in L122, model framework in L234, and other components in L914.

- the very details about Fugaku, e.g. L751ff (At that point in the paper I had forgotten what the eco-boost mode does and had to reread the paper to find it on L201. The portion of readers that have access to Fugaku and hence for which these information are relevant, are probably low).

> We agree that we provided extensive detail of Fugaku for application scientists. We have simplified Section 2.3 and deleted L431 in the revised manuscript. Alternatively, we would like to retain the discussion on power performance and its depen-

[Figure]

**Figure 2.** (a) Data throughput of microphysics for the two-moment bulk method (green), bin method (blue circles), bin method with stochastic collision–coalescence algorithms (blue triangles), SDM-orig (yellow), and SDM-new (red) with different numbers of tracers or mean SDs per cell. (b) The mean data throughput of SD tracking (SD movement and sorting with a block as a key), condensation process, and collision–coalescence using SDM-orig and SDM-new with different mean SDs per cell. The dotted and solid lines show the mean data throughputs for SDM-orig and SDM-new, respectively. The range between the minimum and maximum throughputs of condensation and collision–coalescence for SDM-new is indicated by the colors because load imbalance is crucial only for SDM-new.

[Figure]

**Figure 3.** Computational resources for BOMEX and SCMS experiments for various cloud microphysics schemes and different numbers of tracers or mean SDs per cell: (a) node-hours using normal and boost modes; (b) energy consumption using boost and boost eco modes. In (a), the results for the boost and normal modes are shown by the filled and open markers, respectively. In (b), the results for boost eco and boost modes are shown by the filled and open markers, respectively. The red-dotted lines show the lines proportional to $N$.

dence on modes while we provide a guide for those not interested in it because we believe it will become more important in the future.

I read section 3 as if the improvements of the SDM are of technical nature and should not deteriorate the physics. Then I was surprised that in only physical comparison between the SDM-orig and SDM-new in Fig. 6 the difference seem quite substantial around t=1200. You argue with different versions of the flow solver. What change prevents you from using the same version? Then also the discussion in L565ff could be removed. To strengthen the confidence in the SDM-new it would be worth to investigate the difference, their cause and relevance a bit more.

We investigated the difference in detail and found that the main difference is the contribution of the slightly different initial value for QV. The initial value of QV used in SDM-new is larger than expected from the given relative humidity compared with that of SDM-orig, resulting in stronger convection. After using the same initial condition for QV, the difference in the results decreased (Figure 4). The BOMEX and SCMS experiments are not affected by the modification.

Next, to investigate the contribution of the model version and improvement, we performed warm bubble experiments by taking as many SDs as possible (SD32768). Figure 5 shows the LWC for different model versions and settings. The upper right panel in Figure 5 shows the LWC obtained with SDM-new, using almost the same setup as for SDM-orig, except for some minor modifications (numerical representation precision, random number seeds, order of calculations, etc.). Then, we considered model improvements (lower left panel in Figure 5), including the exclusion of the monotone FCT, the use of time-averaged variables during each long time step for the calculations of microphysics, the use of second-order CVI, the use SDs sampling from PDF, which is proportional to total density, and the use of Sobol sequence for initialization. Lastly, we simplified the model setup by reducing collision–coalescence calculations in noncloudy cells and ignoring pressure dependence on coefficient $A$ for condensation calculations (the lower right panel in Figure 5). The difference in the upper left and right panels is very small, indicating consistency between SDM-orig and SDM-new. The model improvement in this study resulted in slightly different results for precipitation timing and behavior of clouds in the upper layer. The simplification of the calculations resulted in a small difference in LWC.

Based on the above discussion, we conclude that the model version is unimportant and the difference in the results between SDM-orig and SDM-new becomes smaller for many SDs, indicating that sampling and randomness caused the differences. In the revised manuscript, the above discussion is provided in L705. Note that we have not provided Figure 5 in the revised manuscript but have provided it in the supplemental Data sets. According to the correction to the initial value of QV, we also performed an additional experiment and moved the corresponding figures (Figure 6, markers for particles are slightly larger) to Figure 5 in the revised manuscript.

[Figure]

**Figure 4.** Horizontally averaged time–height cross-section of the liquid water content (LWC) for different cloud microphysics schemes.

**specific comments**

**2.1 Governing equations:**

L109-119: The discussion about anelastic vs fully-compressible equation is a bit confusing and maybe misleading. I understand that global communications can be the performance bottleneck. However, as the authors acknowledge, other groups have success-

[Figure]

**Figure 5.** Same as Figure 4 but for SDM-orig32768 and SDM-new32768 using different settings

fully managed to gain an overall benefit from the larger timestep possible with the anelatic approximation. The SDM timestep seems comparable to Mellado et al. 2018 dynamics time step, while in this paper the dynamics timestep is 16 times lower. L891ff states another disadvantage.

> We agree with you that the description of anelastic equations in Section 2 is redundant and confusing. In the revised manuscript, we have specified the differences between fully-compressible and anelastic equations.

L132: Mention, that this is not a complete list of the warm phase microphysical processes. For example spontaneous and collisional break-up are not implemented, but are relevant for other meteorological situations with high rain rates. Actually, collisional break-up might be most complicated warm phase process, even if can be more naturally implemented in the SDM framework.

> In the revised manuscript, we have stated the process ignored and assumptions made for microphysics to be solved explicitly. We added the phrases in blue, as follows:
>
> L132: In this study, only the following warm cloud processes were considered: movement, activation/deactivation and condensation/evaporation, and collision–coalescence. Spontaneous and collisional breakup process was not considered in this study.
>
> L136: The $i$th SD moves according to the wind and falls with terminal velocity assuming that the velocity of each SD reaches the terminal velocity instantaneously.
>
> L146: The terms $a/R$ and $b/R^3$ represent the curvature and solute effects, respectively. The ventilation effect is ignored in Eq. (2).

L150: Maybe I missed it, has the size of the collision grid ever been discussed in the context of SDM? If it is kept equal to size of the fluid cell, the interaction distance of SDs decreases with mesh resolution. I could even imagine that it could be a reason why the SD statistics in Sato et al. 2017 did not converge even at 5 m resolution. If you think that is a valid discussion point, it would fit in section 6 at L1077.

> We agree with your comment. Extending the collision grid to $2\Delta$–$8\Delta$, which may be a resolved-scale length for mixing, may be reasonable for determining the interaction between fluid dynamics and microphysics. Analyzing low-pass filtered variables below filter length may be valid for investigating grid convergence. Though it is still debatable, we performed a similar analysis in our previous report (Matsushima et al. 2021). Thus, in this study, we used the same grid for the

[Figure]

**Figure 6.** Distributions of SD positions at (left) $t = 600\,\mathrm{s}$ and (right) $t = 1{,}200\,\mathrm{s}$ colored by the initial $y$ coordinate $(Y)$ when CVI of the first-order (CVI-1) and second-order (CVI-2) spatial accuracy are used for SD movement. The range of $0 \leq y \leq 5{,}000$ and $1{,}000 \leq Y \leq 4{,}000$ are shown in each panel.

collision and model grids. In the revised manuscript, we have stated the collision grid explicitly. In the computational aspect, however, we are unsure if such a method is effective in optimization for SDM. Unfortunately, we cannot exclude sorting with cell index as a key, which is a bottleneck for collision–coalescence calculations. If we separate the collision grid from the model grid, we should subtract the noncloudy volume from the collision grid volume and avoid registering SDs within a noncloudy cell as a candidate for collision, which requires sorting with cell index anyway.

L155: A philosophical question: Do you view a super-particle as a sample from the real particles or as a statistical average over the multiplicity of real particles. I would prefer the former as the SDM could converge to the full problem for multiplicity -> 1 (even so that is not true for the O(N) collision algorithm). However, in my view turbulent (subgrid-scale) diffusion on the position space should then be included, otherwise initially very close SDs will not separate. This will prohibit the connection to DNS where two-point statistics are of interest. It seems even worse with low or fixed precision arithmetic (L441).

Thank you for the insightful question. We suppose that the philosophical difference becomes clearer if we consider how to take the limit to the real problem. We can assume two possible limits; $\xi \to 1$ with fixed $\Delta x$ and $\xi \to 1$ with $\Delta x \to 0$. The first limit does not solve the real problem unless subgrid diffusion is not considered. In this case, it might not be possible to distinguish SD's position for low-precision representation. Note that the representation precision is $\sim 1\,\mu\mathrm{m}$ for $2\,\mathrm{m}$ grid. This is sufficient to distinguish SD's position because the mean distance between SDs ignoring inertia is $0.92\,\mathrm{cm} \gg 1\,\mu\mathrm{m}$ for CDNC of $100\,\mathrm{cm}^{-3}$. We can also use stochastic rounding to distinguish SD particles so that initially very close SDs can separate for the uncertainty of SD position. Alternatively, the second limit leads to the real problem because subgrid motion becomes relatively small. In addition, the representation precision for SD position increases with the limit to the

real problem.

Mellado et al. (2018) considered the first limit (using the original Navier–Stokes equation but changing only the kinematic viscosity among the atmospheric parameters). We implicitly assumed the second limit in the original manuscript, although we were motivated by the Mellado et al. (2018) approach. We have provided the following discussion in the revised manuscript: In the SDM, we did not consider the effect of turbulent fluctuations on movement, activation/deactivation and condensation/evaporation, and collision–coalescence owing to the high additional computational cost and amount of memory space required. However, we should include the effect of subgrid motion (or Brownian motion by kinematic viscosity) to ensure the convergence to DNS with $\xi \to 1$ while fixing the spatial grid length (Mellado et al. 2018), which should be addressed in future work.

**3.1 Model framework of SCALE**

L238: 3D/2D decomposition leads to a lot of data transfer via network (argument against the anelastic approximation in section 2.1). SDM physics contributes only to 50% runtime attributed to SDM. How much this MPI Wait times compared to other technical work as SD sorting?

> The measured time for 3D/2D decomposition is listed in Table 1, which is provided at the bottom of this letter. The important point is that the number of MPI processes involved in communication in 3D/2D decomposition is small compared to that of all MPI processes. For example, in Section 5.2, the number of nodes involved in 3D/2D decomposition is only 16, which is much smaller than 36864. Therefore, we expect the cost to be considerably lower than that of the all-to-all communication among all nodes. We have stated this in the revised manuscript.

L245: Another argument for a 2D decomp for the SDM is that particle sedimentation in z direction does not lead to MPI communications.

> We have revised L247 as follows: In addition, the amount of computation and data movement varies, depending on whether clouds and precipitation shaft are within the domain.

**3.2 Initialization of super-droplets**

L281: Could you make it clearer, why constant multiplicity is more natural? I can only understand that in the limit of multiplicity -> 1. It is important as it serves as motivation to develop the new init scheme.

> When the number of SDs is smaller than the number of actual droplets but does not considerably deviate from it, the multiplicity of some droplets could become less than 1, depending on the PDF for sampling. In this case, it is reasonable to impose a constraint on the number so that the dynamic range of multiplicity is smaller (i.e., more similar to constant multiplicity) and multiplicity for all SDs is larger than 1.
>
> In the revised manuscript, we have stated the following: When we sample a vast number of SDs, and the number of samples becomes close to the actual number of droplets, it is reasonable to impose a constraint on the number so that the dynamic range of multiplicity is smaller (i.e., more similar to constant multiplicity) and multiplicity for all SDs are larger than 1.

L285: This question is not completely answered in L825 ff. Add in section 6?

> Please refer to L1103 (original manuscript): the sensitivity of the microphysical variability and precipitation to initialization parameter $\alpha$ should be further explored by high-resolution simulations.

L362-363: cache blocking and moving branches out of loops was already the way to go for vector computers, maybe citations would be appropriate.

> We have added the reference Lam et al. (1991) for cache blocking.

**3.3.2**

L385 Important point, could you motivate the consistency of the densities a bit further. L155 states a turbulence scheme is used for tracers but not for the SDM. see comment to L155

We assume that SD number density is initially proportional to the total density. Then, the relation always holds if the eddy diffusivity of the total density and SD number density are identical, and the sedimentation of water substances is ignored. Even if the SD number density decreases with an updraft in the Lagrangian view, it is consistent with the environmental SD number density. Thus, we retained the number of SDs in convective clouds. In addition, we have reduced the non-physical variance of SD number density caused by the inconsistency of interpolated divergence at the SD position and grid divergence. In this study, we considered only the changes in SD number density due to grid-scale motion and precipitation and ignored the effect of eddy viscosity (or subgrid Lagrangian motion) because it is relatively small. Changes in the SD number density due to subgrid turbulence will be considered in future studies. In the revised manuscript, we have briefly stated the above in Introduction and Section 3.3.2.

**3.3.5**

L520 citations seem to be appropriated

We have added the reference Decyk and Singh (2014) (L520) for PIC sorting, although their method is slightly different from ours.

L527-534 This is not obvious to me, maybe a sketch would help?

Data hierarchy within each MPI process and an example of one-dimensional SD sorting are shown in Figure 7. In the revised manuscript, we have added Figure 7 and cited it in L527-534.

[Figure]

**Figure 7.** Data hierarchy (particle & cell, block, group) in each MPI process and algorithm for SD sorting toward $x$-direction in each block. In the example, an MPI process has 4 groups, a group has 4 blocks, and a block has $4 \times 4$ cells. Using a list ({}) that stores the indices, SD sorting completes by copying in the SDs moving to adjacent blocks and copying back the SDs moving into the block. The number of total SDs within a block is monitored by counting only the moving SDs.

**4.1**

eq. 10 this metric seems to measure the usage of the available computer power, but ignores convergence, i.e. how many tracers/bins/SDs are actually needed.

Thank you for highlighting this. We agree that it is important to include some convergence aspects in the discussion. However, it would be difficult to incorporate such into the metric. First, convergence properties can depend on the variable to be checked, for example, liquid water content, cloud droplet number concentration, and precipitation. Second, they can depend on the cloud form, amount of CCN, PDF used for initialization, and random number properties (pseudorandom number or low-discrepancy sequence). Thus, in this study, we did not mention the convergence aspect, except that 128/cell SDs are necessary for SCMS case, which was confirmed in our previous study (Matsushima et al. 2021) and other studies (Xue et al. 2022). In the revised manuscript, we have stated the limitations of the metric and cautions to be taken.

L582 A short discussion on how much optimization might be possible might be appropriate and link to the discussion in section 5 L953ff.

General optimization has been applied to Seiki and Nakajima (2014) scheme. In this scheme, the innermost loop for the vertical grid index, which performs complex calculations on each water substance, is vectorized by SIMD instructions. The innermost calculations are divided by separate loops by hand and compiler to improve computational performance by using cache. However, there may still be room for optimization to search for optimal loop fission and reordering calculations to hide operation latency. Optimization in terms of computational cost is applied to Suzuki et al. (2010) scheme. However, the innermost loops for bins are not vectorized for a small number of iterations. We have added this to the revised manuscript.

L583 This comparison seems indeed unfair, especially as you emphasize in several places that SD-new128 has similar performance (see general comment above)

We have compared the physical performance of SDM with that of the two-moment bulk scheme for only warm microphysics in the revised manuscript and deleted L583. Nonetheless, we would like to retain the results of the two-moment bulk scheme with mixed-phase microphysics for discussion in Section 6.

**4.2**

L625 Why has BIN a O(N**2) scaling? That implies that advection not dominant. If the reason is collisions, a linear sampling might also be possible. Hence, degree of optimization is hard to judge.

Thank you for highlighting this. For the bin method, the computational complexity is $O(N^2)$ for $N$ bins when all possible combinations of collision are considered ($_N\mathrm{C}_2$). A collision method similar to SDM may be applied. Sato et al. (2009) proposed reducing collision combinations by applying Monte Carlo integration inspired by SDM. As the method is implemented in the bin method (Suzuki et al. 2010), which we have used in our study, we performed additional warm bubble experiments with the option, and the results are shown in Figures 1 and 2 (BIN stoch.). We used the number of combinations of collision ($M$), $M = 16, 16, 16, 32, 128, 1024, 4096$ for the number of bins $N = 8, 16, 32, 64, 128, 256, 512$, respectively. The elapsed time of the bin method was reduced by using this option. To reduce the order of the computational complexity to $O(N)$, $M \sim N/2$ can be used. However, when we set the number of bins $N \geq 128$ and used $M \propto N$, the computations were terminated by large negative values of liquid water that cannot be compensated by filters. This is consistent with the previous study (Sato et al. 2009), which stated that $M \geq 0.056_N\mathrm{C}_2$ should be used to avoid large negative values of liquid water. Thus, computational complexity has not been successfully reduced. When this option can be stably used in the future, the elapsed time of the bin method becomes comparable to or faster than that of SDM-orig. In such situations, however, SDM would have advantages for considering multidimensional attributes to study collision–coalescence, as discussed by Shima et al. (2020). The above discussion has been provided in the revised manuscript.

**4.3**

L761 It do not understand the last sentence.

For clarity, we have revised the sentence as follows: When the number of available nodes is limited, simulations using the two-moment bulk method with more grids can be performed.

L783 what do you mean by "only SDs in one cell"?

We have revised the sentence as follows: To enable the intercomparison of models for the readers, each microphysical variable at a cell was calculated by taking statistics for SDs (only) within the cell.

L814 it is unfortunate that the bulk scheme seems to have a bug or inappropriate threshold. That weakens the conclusion in L823 and L1165 considerably.

Thank you for allowing us to reconsider this problem. The strange pattern is attributed to the fact that the bulk method cannot effectively solve evaporation by entrainment.
In Seiki and Nakajima (2014) scheme, the change of cloud water number density by evaporation was modeled such that it does not decrease unless the mean mass of cloud water (xc $\equiv 4/3\pi\rho_w R^3$, here $\rho_w$ is water density, and $R$ is droplet radius)

falls below the threshold $xc_{ccn}$. If it occurs, the cloud droplet number density completely evaporates (although smoothing is introduced). Mixing by entrainment always occurs homogeneously in this model. In the default setting, the threshold was very large ($10^{-12}$). This corresponds to a droplet radius of approximately $6\,\mu m$ (CFADs are shown in the upper panel of Figure 8). If we reduce the threshold to a reasonably small value ($4.2 \times 10^{-15}$, approximately $1\,\mu m$), the strange pattern becomes less noticeable (middle panel in Figure 8).

However, as we can assume that the real mixing can be homogeneous and inhomogeneous, the assumption of homogeneous mixing may be too simple. To incorporate the mixing scenario, we introduced the subgrid evaporation model (Morrison and Grabowski 2008, Jarecka et al. 2013), which considers a decrease in the cloud droplet number and water content, depending on the entrainment-mixing scenario. Jarecka et al. (2013) predicted a local mixing scenario calculated from the characteristic scale of cloud filaments and the fraction of a grid box occupied by cloudy air, which requires two new tracers. In this study, for simplicity and to avoid additional computational costs, we further assumed that the mixing scenario is fixed everywhere by a parameter $\alpha$ ($0 \le \alpha \le 1$, $\alpha = 0$ for homogeneous and $\alpha = 1$ for extreme inhomogeneous mixing) and evaporation is not delayed by subgrid-scale mixing. Based on the report of Jarecka et al. (2013), the parameters were set as $\alpha = 0.5$ for the pristine case (warm bubble, BOMEX) and $\alpha = 0.75$ for the polluted case (SCMS). The lower panel of Figure 8 shows the CFAD obtained using the Seiki and Nakajima (2014) scheme with subgrid-scale evaporation. CFAD slightly decreased from the original setup. The strange pattern at $R \sim 1\,\mu m$ observed in the original setup completely disappeared and became consistent with the CFAD obtained using SDM. In our future studies, we will investigate CFAD when predicting local mixing scenarios with additional tracers and a more reasonable assumption for delayed subgrid-scale mixing before evaporation.

**5.2**

Table 3. tracking + condensation + coalescence only adds up to 141 min of 274 in microphysics. How much is due to the 3D/2D conversion + MPI wait steeming from the flow solver decomposition and how much from the technical workload needed for the microphyiscs itself like the SD sorting?

The measured times for 3D/2D decomposition and SD sorting are listed in Table 1. However, it should be noted that these times may not be the time needed to process 3D/2D decomposition and SD sorting as load imbalances from other processes will likely affect them. Also, as precise measurements are unimportant for practical cases, such as BOMEX and SCMS, we prefer to avoid measuring them.

**Table 1.** Elapsed time, FLOPS (peak ratio of the FLOPS [%]), Peta instructions per second, memory throughput (peak ratio of the memory throughput [%]), and particle throughput (# of floating point operations per SD)

|  | Time [min] | Speed [PFLOPS] | PIPS | Memory Throughput [PB/s] | Part. Throughput [particle·step/s] |
|---|---|---|---|---|---|
| Time integration loop | 576 | 7.97 (7.04) | 1.86 | 13.7 (37.2) | |
| Dynamics | 290 | 8.55 (7.55) | 2.03 | 20.5 (55.7) | |
| Microphysics | 274 | 7.50 (6.62) | 1.69 | 6.25 (16.9) | $2.86 \times 10^{13}$ |
| Short time step | 238 | 9.50 (8.39) | 2.19 | 21.3 (57.9) | |
| Tracer time step | 15.0 | 5.85 (5.17) | 1.78 | 21.6 (58.7) | |
| Tracking | 87.9 | 15.3 (13.5) | 2.14 | 2.89 (10.5) | $8.91 \times 10^{13}$ (171) |
| Condensation | 32.6 | 18.2 (16.1) | 5.35 | 5.28 (14.3) | $2.40 \times 10^{14}$ (75.9) |
| Coalescence | 5.75 | 7.58 (6.69) | 2.96 | 17.5 (47.3) | $1.36 \times 10^{15}$ (5.57) |
| SD sorting | 79.2 | | | 12.5 (33.9) | |
| 3D to/from 2D conversion | 53.47 | | | | |

L953ff bin method O(N**2), tracer advection is not even dominant (see comment to L625

The elapsed time depends on the computational algorithms and degree of optimization for the microphysics schemes. Here, we estimated the minimum elapsed time required by tracer advection and SD movement. In the revised manuscript, we have stated this assumption.

L972 0.8*15*33 /= 363

[Figure]

**Figure 8.** Contoured frequency by altitude diagrams (CFADs) of cloud droplet number concentration (CDNC), LWC, and mean and standard deviation of the radius for SCMS experiments. Snapshots of BULK2MOM with (top row) $xc_{ccn} = 1.0 \times 10^{-12}, \alpha = 0$ and (middle row) $xc_{ccn} = 4.2 \times 10^{-15}, \alpha = 0$ and (bottom row) $xc_{ccn} = 4.2 \times 10^{-15}, \alpha = 0.75$ are shown.

The estimation in L972 is based on when a memory throughput of 80% against theoretical peak performance is achieved. Thus, the elapsed time of tracer advection with 33 tracers was estimated to be $33*15*0.587/0.8 \sim 363$ min. We have modified L971 as follows: Then, a possible optimization could refactor the codes, and we may improve the memory throughput performance of tracer advection to achieve up to 80% of the theoretical peak performance.

**6.1**

L990ff are too optimistic. First, instantaneous freezing and melting assumptions are quite restrictive, e.g. leads to complete neglect of wet growth in deep convection. Mixed-phase regimes -38 to 0 are very important and frequent, e.g. riming of ice particle is a main driver for cold phase precip. Hence, I think the coexistence of water and ice particles will be needed often and extension to mixed-phase particle should be the next step after Shima 09 and Shima 20. In conclusion, I feel that the comparison to mixed-phase bulk is not justified.#f

Thank you for your comment. We have deleted the paragraph starting from L990. In L979, we refer only to the case for temperatures above $0\,°C$ everywhere. In this paragraph, we retain the discussion on special cases in which the computational cost of mixed-phase SDM should be comparable to that of warm SDM if well implemented, whereas the computational cost of the mixed-phase two-moment bulk method increases for additional tracers. In L997 of the revised manuscript, we

have provided an example of riming for a mixed-phase regime, based on your suggestion, and described the challenges of optimization.

**6.3**

L1014ff Online postprocessing seems needed, for statistical analyses DSD on the scale of blocks should be sufficient. Of course the checkpoint/restart problem remains if the simulation time is longer than allowed by the compute queue. However, for such large scale applications it might be possible to discuss such rules with the computing center.

Thank you for your suggestion. Currently, statistical information about aerosols and clouds (number and water content) and activated SDs are outputted in a low-precision format. Analyses that need all information about SDs are possible only online. As you observed, we can make checkpoint/restart files in Fugaku, whose size is less than 400 TB, if we are permitted by the center. In the revised manuscript, we have mentioned that an online analysis should be performed if all information on SDs is necessary, and we may consult with the center for storing big data files. We would like to retain the discussion on data compression but in a simpler form.

**technical corrections**

L12: Add something like "possible" or "probably" for GPUs, as the effectiveness is not demonstrated in this paper.

Thank you for your suggestion. We have revised the sentence accordingly in the revised manuscript.

Fig 1. end of caption, dotted lines also in (d)

We have revised the statement in the revised manuscript.

L355 reformulate the sentence to avoid misinterpretation. E.g. move "except for the collision-coalescence process" to a new sentence or put it in parenthesis.

Thank you for your suggestion. We have revised the sentence accordingly.

L482 link to L375?

We have added "As discussed in Section 3.3.1" in L482 to remind readers of L375.

L661 "resists" might not be the best word

We have revised the word to "increases more gradually than linearly."

Fig 7. larger symbols in the legend

We have revised the symbols accordingly (see Figure 3 in this letter).

**References**

Decyk, V. K. and Singh, T. V.: Particle-in-cell algorithms for emerging computer architectures, Comput. Phys. Commun, 185, 708–719, https://doi.org/10.1016/j.cpc.2013.10.013, 2014.

Jarecka, D., Grabowski, W. W., Morrison, H., and Pawlowska, H.: Homogeneity of the subgrid-scale turbulent mixing in large-eddy simulation of shallow convection, J. Atmos. Sci., 70, 2751–2767, https://doi.org/10.1175/JAS-D-13-042.1, 2013.

Lam, M. D., Rothberg, E. E., and Wolf, M. E.: The cache performance and optimizations of blocked algorithms, Oper. Syst. Rev. (ACM), 25, 63–74, https://doi.org/10.1145/106973.106981, 1991.

Morrison, H. and Grabowski, W. W.: Modeling supersaturation and subgrid-scale mixing with two-moment bulk warm microphysics, J. Atmos. Sci., 65, 792–812, https://doi.org/10.1175/2007JAS2374.1, 2008.

Santos, S. P., Caldwell, P. M., and Bretherton, C. S.: Numerically relevant timescales in the MG2 microphysics model, J. Adv.

Model. Earth Syst., 12, e2019MS001 972, https://doi.org/10.1029/2019MS001972, 2020.

Sato, Y., Nakajima, T., Suzuki, K., and Iguchi, T.: Application of a Monte Carlo integration method to collision and coagulation growth processes of hydrometeors in a bin-type model, J. Geophys. Res., 114, https://doi.org/10.1029/2008JD011247, 2009.

---

## Author Response (AR1)

**Reply to the comments of Referee #1 (RC1) on article gmd-2023-26**

Toshiki Matsushima, Seiya Nishizawa, Shin-ichiro Shima

Thank you for taking the time to read through our manuscript and provide us with positive and helpful comments. Below are our point-by-point responses to your comments.

1. Maybe the only general comment is whether the authors could reduce the length of the manuscript, or consider a more clear separation between the discussion of general aspects of the SDM and the particularities of the implementation in Fugaku, as I tried to do in the first paragraph of this letter. I wonder if section 6 could be shorten. As said, this is a minor point and I realize it is difficult, but somehow I think the reader might be better guided through the paper to concentrate on the aspects that might be more interesting for her or for him.

   I found the conclusions, however, very well and they helped me to end the paper with a good idea of the various parts.

   > Thank you very much for your suggestion. We have simplified the Discussion (Section 6) and other sections as much as possible for clarity and to reduce redundancy.

2. Title, abstract, line 1, and through the paper, the authors use the term "ultrahigh-resolution" and I was wondering if they could substitute that term by another one that is more informative. It seems that, in line 9, the authors refer explicitly to centimeter to meter scale, so why not say "centimeter-to-meter scale resolution" or " submeter resolution" ?

   Otherwise, what would come after *"ultrahigh resolution"* when we reach the following step towards higher resolution?

   > As you suggested, we have revised the term to meter-to-submeter scale.

3. Title, abstract, line 5, and through the paper, the authors use the term *"sophisticated"*. I was not sure what it means. Does it mean that it considers more physical processes, or a better model of them, or does it refer to the technical implementation from a computer science point of view?

   > Thank you for your comment. In the revised manuscript, we have considered words that improve clarity. As other papers have also used the term "sophisticated" microphysics model to describe a model that introduces a more complex process, we have retained it in the revised manuscript. Alternatively, our study focused on improving many aspects, such as numerical precision, algorithms, and computational performance. Thus, we have made some revisions to clearly state that.

   > In addition, based on your suggestion, we have revised the title of the manuscript to Overcoming computational challenges to realize meter-to-submeter-scale resolution in cloud simulations using super-droplet method

4. abstract, line 4, the authors say *"does not make any assumption for the droplet size distribution"*. Since some aspects of the dynamics within the superdroplet or the interaction between superdroplets are still modeled, as discusses in section 2 and 3, I wonder if it might be better to say "makes less assumptions about the droplet size distribution and it is more physically sounded", or something similar.

   > As suggested, we have revised the statement to making less assumption for the droplet size distributions.

5. Abstract, line 18: instead of *"perfect weak scaling"*, it might be stronger and clearer to say 98% weak scaling up to the corresponding number of nodes or cores, as it is done in the conclusions.

   > Thank you for your suggestion. We have revised the phrase to 98 % weak scaling in the revised manuscript.

6. Introduction, line 35, the authors refer to Schulz and Mellado, 2018, but the reference Schulz and Mellado [2019] might be stronger for their case. Schulz and Mellado [2019] studies one micro-physical effect, namely, sedimentation, and shows that sedimentation is more important than previously thought, which strongly supports the efforts presented in this SDM manuscript to better represent the DSD.

   > Thank you for the suggested reference. We have cited Schulz and Mellado (2019) instead of Schulz and Mellado (2018). We have also added the following statement to L35:

Following Mellado et al. (2018), Schulz and Mellado (2019) investigated the joint effect of droplet sedimentation and wind shear on cloud-top entrainment and found that their effects can be equally important for cloud-top entrainment, while

7. Introduction, line 34 say *"which sets the eddy viscosity constant"* referring to DNS. I wonder if this sentence is needed. This seems to suggest that DNS is one type of LES, which is not the way DNS is used in turbulence research, where the concept originates from, since there is no eddy viscosity in DNS [Orszag and Patterson, 1972, Moin and Mahesh, 1998, Pope, 2000, Mellado et al., 2018]. DNS rescales the original in terms of size or in terms of physical properties to study Reynolds number effects, and remains accurate in the smallest resolved scales, which LES does not.

Thank you for highlighting this. In the revised manuscript, we have explained DNS and LES, based on your comment and Mellado et al. (2018):

Mellado et al. (2018) suggests that combining the direct numerical simulation (DNS) approach, which solves the original Navier–Stokes equations while changing only the kinematic viscosity (or Reynolds number) among the atmospheric parameters, and large-eddy simulation (LES) approach, which solves low-pass filtered Navier–Stokes equations for unresolved flow below filter length, can accelerate research on related processes.

8. In several places through the manuscript, the authors refer to particle-in-cell (PIC) methods. What are the differences between the superdroplet method (SDM) and particle-in-cell methods?

Thank you for your comment. The PIC method is used to solve a specific type of partial differential equation, which describes a coupled system of particles and cell-averaged variables. SDM can be regarded as an application of the PIC method to solve cloud microphysics and macrophysics. In the original manuscript, we did not introduce an abbreviation for the super-droplet method, but in the revised manuscript, we have defined the abbreviation properly and have defined the PIC method and its relation with SDM more clearly.

added to L42: In particular, herein, we focus on the super-droplet method (SDM), which is one of the particle-based schemes developed by Shima et al. (2009).

added to L72: To the best of our knowledge, load balancing has not been applied to the SDM, even though some studies have applied it to other simulations, such as plasma simulations (Nakashima et al., 2009) The SDM and some other plasma simulations are based on solving partial differential equations that describe a coupled system of particles and cell-averaged variables, known as the particle-in-cell (PIC) method.

simplified L72: However, applying load balancing for weather and climate models is not a good option because such codes are complicated and changes in dynamic load balancing can affect the computational performance of other components.

9. Section 2, line 118, the authors write *"the anelastic equations assume horizontally uniform mean fields and are not appropriate for computing wider domains"*. I wonder if this sentence is needed. The mean fields need not be the reference fields that are used in the anelastic formulation, and one can have anelastic formulations of statistically inhomogeneous flows (a cloud bubble).

Thank you for your comment. We agree that the description of anelastic equations in Section 2 is redundant and confusing. In the revised manuscript, we have clearly stated the differences between fully compressible and anelastic equations:

The fully-compressible equations require a shorter time step $(20^{-1}$–$10^{-1})$ than that needed to solve anelastic equations (advection time step). However, they may have an advantage when using a large number of MPI nodes, as they do not require collective communications.

10. Equation (1), what are the assumptions for using this equation? The same applies for instance to the description of collision-coalescence and relates to point 3 before, namely, that there are still some assumptions in the SDM and it might be convenient to explain them as clearly as possible to better interpret the results and the limits of applicability.

In the revised manuscript, we have stated the process ignored and assumptions made for microphysics to be solved explicitly.

L132: Spontaneous and collisional breakup processes were not considered here.

L136: The $i$th SD moves according to the wind and fall with terminal velocity, assuming that the velocity of each SD

> reaches the terminal velocity instantaneously:
>
> L146: The ventilation effect is ignored in Eq. (2).

11. Section 3.3.1, *"The effective resolution is* $6\Delta - 10\Delta$ *for planetary boundary layer turbulence"* I assume that this is for the second-order methods that the authors use, but they might indicate it explicitly here because the effective resolution might be substantially smaller in pseudo-spectral schemes or similar, which are often used in boundary-layer meteorology [Sullivan and Patton, 2011, Pope, 2000].

> Thank you for your comment. Second-order spatial discretization for the pressure gradient was employed in this study. Thus, the dynamical core used is (overall) second-order spatial accuracy. However, as discussed by Nishizawa et al. (2015), the pressure gradient term is mainly for fast acoustic waves and has less effect on slow modes, such as mixing, i.e., energy spectrum. Thus, we believe that our simulations have a certain degree of numerical accuracy. As you highlighted, large-eddy simulations using the (pseudo) spectral method are numerically more accurate and might have an advantage in effective resolution because explicit filtering can be used. This has been stated in the revised manuscript:
>
> The typical effective resolution is $6\Delta$–$10\Delta$ for planetary boundary layer turbulence, which may depend on the numerical accuracy of the spatial discretization of basic equations and filtering length and shape of LES.

12. In section 6.4.1, I was wondering how much computational time or memory save is gained by using FP32 instead of FP64 in that part of the microphysics. I was trying to have a sense of priorities in addressing the standing challenges. For instance, the disk space challenge indicated in section 6.3 seems most important, but I might be wrong.

> In implementing SDM, most of the information about SDM is the arrays for SD attributes because other information, such as intermediate values of SD position, which is required for the second-order Runge-Kutta method (Heun's method) for SD tracking, is mainly stored in a local stack in each OpenMP thread, and because the size of the working arrays, such as for SD sorting is small, as discussed in Section 3.3.5. The amount of memory that SDM uses is estimated as
>
> meshes $\times$ SDs per cell $\times$ attributes $\times$ information of each attribute $\times$ (1+extbuf);
>
> here, extbuf is the ratio of the extra buffer arrays to the number of SDs. Thus, the amount of information would be halved if information on all attributes is halved. This applies if we reduce the numerical precision from double to single precision. Alternatively, the computational cost is a bit more complicated, for example, for condensation/evaporation and activation/deactivation. The number of iterations for Newton's method would likely change if the tolerance relative error is squared. We speculate that the number of iterations increases by 1 or 2 because Newton's method is quadratically convergent.
>
> In the case of SCMS, the elapsed time only for microphysics using four nodes of FX1000 and with FP64 was 248 (349) min without (with) collision-coalescence calculations in non-cloudy volumes. The memory for only microphysics was 54.9 GB. On the other hand, the elapsed time with FP32 was 150 (274) min, and the memory was 29.7 GB. The computational-cost reduction on collision–coalescence is insignificant because the amount of information of a key for SD sorting did not change when we used FP64 to FP32, and the multiplicity remained INT64 due to a small number of SDs. In the revised manuscript, we have added the above discussion to L811:
>
> Here, the elapsed time and memory usage for only microphysics with FP64 were 1.03(1.45) NH without (with) collision–coalescence calculations in non-cloudy volumes and 54.9 GB, which are 1.3–1.6 times and 1.8 times larger than the case with FP32 for 0.625(1.14) NH and 29.7 GB memory usage, respectively.
>
> The problem for checkpoint/restart files can be avoided by consulting the computer center, and we have modified this in the revised manuscript. However, we believe that it is an important subject, and a data-scientific approach may help mitigate it.

13. Conclusions, line 1129 *"In contrast, the constant multiplicity method is a natural choice for DNS"*. I think I did not understand this sentenc.

> Thank you for your suggestion. We have removed this sentence.

14. I think the conclusions do not refer to section 3.3.2 "Super-droplet movement", which I thought was interesting in general, not only for the particular implementation described here.

We are glad to hear that you are interested in our manuscript. In the original manuscript, we summarized Section 3.3.2 at the end of the first paragraph in the second item. In the revised manuscript, we have modified the paragraph as the following:

For SD movement, the 3D CVI of the second-order spatial accuracy on the C-grid was derived to ensure consistency between the SD number density and air density. The interpolated velocity can represent simple vortical and shear flows within a cell, and the divergence at the position of SDs that are calculated from the interpolated velocity is consistent with divergence at the cell. We subtracted partition information using MPI processes and blocks from the information of SD global positions to reduce information per SD. Then, we stored the relative position of the SD in a block with a fixed-point number using FP32. This approach guarantees uniform precision in representing the absolute position of SD across the computational domain and good numerical accuracy for meter-to-submeter resolution simulations, even when using a low-precision format.

**References**

Schulz, B. and Mellado, J. P.: Competing effects of droplet sedimentation and wind shear on entrainment in stratocumulus, J. Adv. Model. Earth Syst., 11, 1830–1846, https://doi.org/10.1029/2019MS001617, 2019.

**Reply to the comments of Referee #2 (RC2) on article gmd-2023-26**

Toshiki Matsushima, Seiya Nishizawa, Shin-ichiro Shima

Thank you for taking the time to read through our manuscript and provide us with positive and helpful comments. Below are our point-by-point responses to your comments.

**general comments**

Most of the model improvements are of "technical" nature. I am not computer scientist, hence I cannot completely judge these achievements.

Still, the performance achievements seem great. However, I would have liked it even better, if the comparison with traditional bulk method would be made fairer. E.g. in the SDM and bin model there is no "cold" microphysics (yet), but the bulk method contains these tracers. Hence, it is likely that the advection cost could be reduced to 4/10. Or in the time to solution comparison there is no discussion that with the bulk methods larger microphysical timesteps are possible and might be even sufficient to reach convergence. Considering that the SDM-new128 might be more costly than the bulk method. However, I still would want to apply it, as it makes, as stated in the paper, far less assumptions.

> Thank you for the compliment on the performance improvement we have made. We agree with you that the comparison between the mixed-phase microphysics bulk model and SDM is not fair, but it is appropriate in the specific case discussed in this study. We did not use warm microphysics because the original codes need to be modified. Thus, we decided to exclude cold microphysics from Seiki and Nakajima (2014) scheme, performed additional experiments, and compared the results with those of SDM. Nonetheless, we would like to retain the results for mixed-phase microphysics for discussion. Before showing how it changed the results herein, we share our thoughts on comparing SDM and Seiki and Nakajima (2014) scheme. There are two aspects of convergence for the two-moment bulk method; spatial and temporal resolution. The two-moment bulk method imposes empirical assumptions on DSD, which may lead to less spatial variability or no dependency on spatial resolution, such as the spectral width of DSD. However, this does not mean it converges fast to increase spatial resolution; rather, it indicates that, in principle, fair comparison in terms of spatial resolution is difficult. For the temporal resolution, the time step for the two-moment bulk method could be fairly large (e.g., 5 min) in climate simulations. Based on this, one may assume that the two-moment bulk method can solve groups of droplets using large time steps by bundling the fast time evolution of individual droplets. However, as discussed in Santos et al. (2020), this is not the case. The authors performed eigenvalue analysis for a two-moment bulk scheme and found that a fast mode (<1 s) also exists in the bulk scheme. It does not considerably deviate from the time step determined by the evolution of individual droplet dynamics, such as condensation/evaporation. Thus, it is reasonable for convergence to use a sufficient time step that is determined by eigenvalues, instead of a stable time step. Alternatively, large time steps can be considered if it is stable and the mean climate does not change for earth and planetary climate simulations. For such a case, we agree that a bulk method is suitable. Based on the above discussion, we would like to maintain the same spatial and temporal resolution to compare SDM and the two-moment bulk method. We have added the above discussion points to L590 in the marked-up manuscript.
>
> As explained later, we modified the initial condition of the mass mixing ratio of water vapor (QV) for (only) warm bubble experiments using SDM, the two-moment bulk method, and the bin method. In addition, we have modified the evaporation calculations for the two-moment bulk method. We have also added the results using the bin method with a stochastic collision–coalescence algorithm developed by Sato et al. (2009). Figures 1 and 2 in this letter correspond to Figures 3 and 4 in the original manuscript. The differences are small for warm bubble cases. The elapsed time using the warm

microphysics two-moment bulk method is still comparable to that of SDM-128. The description in the revised manuscript is modified as follows (shown in blue):

L675: The maximum throughputs of SDM-new are 61.3 and 20.1 times that of SDM-orig and two-moment bulk method, respectively.

L684: The elapsed time obtained using the two-moment bulk method was 14.0 s, and that obtained using SDM-new with 128 SDs per cell on average is 13.9 s. The maximum throughput of SDM-new is 31.6 times that of the two-moment bulk method.

Figure 3 in this letter shows the results of BOMEX and SCMS case experiments, corresponding to Figure 7 in the original manuscript. In this case, the elapsed time and energy consumption obtained using warm microphysics two-moment bulk scheme are comparable to that of the results between SDM-new32 and SDM-new64. However, as discussed in the revised manuscript, as the elapsed time for SDM increased more gradually than linearly with the number of SDs when 8-128 SDs/cell were used, the main conclusion did not change considerably. Based on the above discussion, we have changed the conclusion that the elapsed time for SDM-new32 to SDM-new64 is comparable to that of the (liquid phase) two-moment bulk method.

[Figure]

**Figure 1.** Elapsed times of the total (circles) and tracer advection and SD tracking (squares) using the two-moment bulk method (green), bin method (blue), SDM-orig (yellow), and SDM-new (red) with different numbers of tracers or mean SDs per cell. Elapsed times of the total (triangles) using the bin method with stochastic collision–coalescence algorithms are also shown. Here, SD tracking includes SD movement and sorting with a block as a key. The blue dotted line is proportional to $N^2$, the red- and yellow-dotted lines are proportional to $N$, and the green-dotted line indicates a constant determined by $N$.

It is quite a long paper. Consider whether it can be shortened. Personally, I tend to read and cite paper more often, if they are briefly and succinctly. To provide examples, I would not miss the following parts in main body of the paper:

> We thank you for kindly reading through our long paper. Below are our responses to the specific examples you provided for us. For other parts, we would like to simplify them as much as possible based on the comments from two referees.

- discussion about the flow solver, i.e. focus on the microphysics (e.g. microphysics - radiation coupling is also not included only mentioned in L914. It is an interesting question how to couple SDM with radiation but clearly out of the scope of this paper). Of course there are a few details that would need to kept, like that a C grid is used and the effective resolution.

> We agree with your comment. In the revised manuscript, we have simplified the description of anelastic equations in L111, flux-corrected transport in L122, model framework in L234, and other components in L914.

- the very details about Fugaku, e.g. L751ff (At that point in the paper I had forgotten what the eco-boost mode does and had to reread the paper to find it on L201. The portion of readers that have access to Fugaku and hence for which these information are relevant, are probably low).

> We agree that we provided extensive detail of Fugaku for application scientists. We have simplified Section 2.3 and deleted

[Figure]

**Figure 2.** (a) Data throughput of microphysics for the two-moment bulk method (green), bin method (blue circles), bin method with stochastic collision–coalescence algorithms (blue triangles), SDM-orig (yellow), and SDM-new (red) with different numbers of tracers or mean SDs per cell. (b) The mean data throughput of SD tracking (SD movement and sorting with a block as a key), condensation process, and collision–coalescence using SDM-orig and SDM-new with different mean SDs per cell. The dotted and solid lines show the mean data throughputs for SDM-orig and SDM-new, respectively. The range between the minimum and maximum throughputs of condensation and collision–coalescence for SDM-new is indicated by the colors because load imbalance is crucial only for SDM-new.

[Figure]

**Figure 3.** Computational resources of BOMEX and SCMS experiments for various cloud microphysics schemes and different numbers of tracers or mean SDs per cell: (a) node hours using normal and boost mode, (b) energy consumption using boost and boost-eco mode. Here, for (a), the results for the boost and normal modes are shown by filled and open markers, respectively. For (b), the results for boost-eco mode and boost mode are shown by filled and open markers, respectively. The red dotted lines show the lines proportional to $N$.

L431 in the revised manuscript. Alternatively, we would like to retain the discussion on power performance and its dependence on modes while we provide a guide for those not interested in it because we believe it will become more important in the future.

I read section 3 as if the improvements of the SDM are of technical nature and should not deteriorate the physics. Then I was surprised that in only physical comparison between the SDM-orig and SDM-new in Fig. 6 the difference seem quite substantial around t=1200. You argue with different versions of the flow solver. What change prevents you from using the same version? Then also the discussion in L565ff could be removed. To strengthen the confidence in the SDM-new it would be worth to investigate

the difference, their cause and relevance a bit more.

We investigated the difference in detail and found that the main difference is the contribution of the slightly different initial value for QV. The initial value of QV used in SDM-new is larger than expected from the given relative humidity compared with that of SDM-orig, resulting in stronger convection. After using the same initial condition for QV, the difference in the results decreased (Figure 4). The BOMEX and SCMS experiments are not affected by the modification.

Next, to investigate the contribution of the model version and improvement, we performed warm bubble experiments by taking as many SDs as possible (SD32768). Figure 5 shows the LWC for different model versions and settings. The upper right panel in Figure 5 shows the LWC obtained with SDM-new, using almost the same setup as for SDM-orig, except for some minor modifications (numerical representation precision, random number seeds, order of calculations, etc.). Then, we considered model improvements (lower left panel in Figure 5), including the exclusion of the monotone FCT, the use of time-averaged variables during each long time step for the calculations of microphysics, the use of second-order CVI, the use SDs sampling from PDF, which is proportional to total density, and the use of Sobol sequence for initialization. Lastly, we simplified the model setup by reducing collision–coalescence calculations in noncloudy cells and ignoring pressure dependence on coefficient $A$ for condensation calculations (the lower right panel in Figure 5). The difference in the upper left and right panels is very small, indicating consistency between SDM-orig and SDM-new. The model improvement in this study resulted in slightly different results for precipitation timing and behavior of clouds in the upper layer. The simplification of the calculations resulted in a small difference in LWC.

Based on the above discussion, we conclude that the model version is unimportant and the difference in the results between SDM-orig and SDM-new becomes smaller for many SDs, indicating that sampling and randomness caused the differences. Note that we have not provided Figure 5 in the revised manuscript but have provided it in the supplemental Data sets. According to the correction to the initial value of QV, we also performed an additional experiment and moved the corresponding figures (Figure 6, markers for particles are slightly larger) to Figure 5 in the revised manuscript.

For detailed changes we have made to the revised manuscript, see the sentences starting from L625 and L796 in the marked-up manuscript.

[Figure]

**Figure 4.** Horizontally averaged time–height cross-section of the liquid water content (LWC) for different cloud microphysics schemes.

**specific comments**

**2.1 Governing equations:**

[Figure]

**Figure 5.** Same as Figure 4 but for SDM-orig32768 and SDM-new32768 using different settings

L109-119: The discussion about anelastic vs fully-compressible equation is a bit confusing and maybe misleading. I understand that global communications can be the performance bottleneck. However, as the authors acknowledge, other groups have successfully managed to gain an overall benefit from the larger timestep possible with the anelatic approximation. The SDM timestep seems comparable to Mellado et al. 2018 dynamics time step, while in this paper the dynamics timestep is 16 times lower. L891ff states another disadvantage.

> We agree with you that the description of anelastic equations in Section 2 is redundant and confusing. In the revised manuscript, we have specified the differences between fully-compressible and anelastic equations:
> The fully-compressible equations require a shorter time step ($20^{-1}$–$10^{-1}$) than that needed to solve anelastic equations (advection time step). However, they may have an advantage when using a large number of MPI nodes, as they do not require collective communications.

L132: Mention, that this is not a complete list of the warm phase microphysical processes. For example spontaneous and collisional break-up are not implemented, but are relevant for other meteorological situations with high rain rates. Actually, collisional break-up might be most complicated warm phase process, even if can be more naturally implemented in the SDM framework.

> In the revised manuscript, we have stated the process ignored and assumptions made for microphysics to be solved explicitly.
> L132: Spontaneous and collisional breakup processes were not considered here.
> L136: The $i$th SD moves according to the wind and fall with terminal velocity, assuming that the velocity of each SD reaches the terminal velocity instantaneously:
> L146: The ventilation effect is ignored in Eq. (2).

L150: Maybe I missed it, has the size of the collision grid ever been discussed in the context of SDM? If it is kept equal to size of the fluid cell, the interaction distance of SDs decreases with mesh resolution. I could even imagine that it could be a reason why the SD statistics in Sato et al. 2017 did not converge even at 5 m resolution. If you think that is a valid discussion point, it would fit in section 6 at L1077.

> We agree with your comment. Extending the collision grid to $2\Delta$–$8\Delta$, which may be a resolved-scale length for mixing, may be reasonable for determining the interaction between fluid dynamics and microphysics. Analyzing low-pass filtered variables below filter length may be valid for investigating grid convergence. Though it is still debatable, we performed a

[Figure]

**Figure 6.** Distributions of SD positions at (left) $t = 600\,\text{s}$ and (right) $t = 1{,}200\,\text{s}$ colored by the initial $y$ coordinate $(Y)$ when CVI of the first-order (CVI-1) and second-order (CVI-2) spatial accuracy are used for SD movement. The range of $0 \le y \le 5{,}000$ and $1{,}000 \le Y \le 4{,}000$ are shown in each panel.

similar analysis in our previous report (Matsushima et al. 2021). Thus, in this study, we used the same grid for the collision and model grids. In the revised manuscript, we have stated the collision grid explicitly:

The volume in which SDs are well mixed and capable of colliding is set to have the same size as the control volume of the model grid.

In the computational aspect, however, we are unsure if such a method is effective in optimization for SDM. Unfortunately, we cannot exclude sorting with cell index as a key, which is a bottleneck for collision–coalescence calculations. If we separate the collision grid from the model grid, we should subtract the noncloudy volume from the collision grid volume and avoid registering SDs within a noncloudy cell as a candidate for collision, which requires sorting with cell index anyway.

L155: A philosophical question: Do you view a super-particle as a sample from the real particles or as a statistical average over the multiplicity of real particles. I would prefer the former as the SDM could converge to the full problem for multiplicity -> 1 (even so that is not true for the O(N) collision algorithm). However, in my view turbulent (subgrid-scale) diffusion on the position space should then be included, otherwise initially very close SDs will not separate. This will prohibit the connection to DNS where two-point statistics are of interest. It seems even worse with low or fixed precision arithmetic (L441).

Thank you for the insightful question. We suppose that the philosophical difference becomes clearer if we consider how to take the limit to the real problem. We can assume two possible limits; $\xi \to 1$ with fixed $\Delta x$ and $\xi \to 1$ with $\Delta x \to 0$. The first limit does not solve the real problem unless subgrid diffusion is not considered. In this case, it might not be possible to distinguish SD's position for low-precision representation. Note that the representation precision is $\sim 1\,\mu\text{m}$ for $2\,\text{m}$ grid.

This is sufficient to distinguish SD's position because the mean distance between SDs ignoring inertia is $0.92\,\mathrm{cm} \gg 1\,\mu\mathrm{m}$ for CDNC of $100\,\mathrm{cm}^{-3}$. We can also use stochastic rounding to distinguish SD particles so that initially very close SDs can separate for the uncertainty of SD position. Alternatively, the second limit leads to the real problem because subgrid motion becomes relatively small. In addition, the representation precision for SD position increases with the limit to the real problem.

Mellado et al. (2018) considered the first limit (using the original Navier–Stokes equation but changing only the kinematic viscosity among the atmospheric parameters). We implicitly assumed the second limit in the original manuscript, although we were motivated by the Mellado et al. (2018) approach. We have provided the following discussion in the revised manuscript:

In the SDM, we do not consider the effect of turbulent fluctuations on movement, activation/deactivation, condensation/evaporation, and collision–coalescence due to the high additional computational cost and memory space required to consider these effects. However, the effect of subgrid motion (or Brownian motion by kinematic viscosity) should be included to ensure the convergence to DNS with $\xi \rightarrow 1$ while fixing the spatial grid length (Mellado et al., 2018); this will be addressed in future work.

**3.1 Model framework of SCALE $\rightarrow$ 3.1 Domain Decomposition**

L238: 3D/2D decomposition leads to a lot of data transfer via network (argument against the anelastic approximation in section 2.1). SDM physics contributes only to 50% runtime attributed to SDM. How much this MPI Wait times compared to other technical work as SD sorting?

The measured time for 3D/2D decomposition is listed in Table 1, which is provided at the bottom of this letter. The important point is that the number of MPI processes involved in communication in 3D/2D decomposition is small compared to that of all MPI processes. For example, in Section 5.2, the number of nodes involved in 3D/2D decomposition is only 16, which is much smaller than 36864. Therefore, we expect the cost to be considerably lower than that of the all-to-all communication among all nodes. We have stated the following sentences in the revised manuscript:

The hybrid type of domain decomposition requires the conversion of grid systems containing every $N_z$ of MPI processes. Note that the cost should not be a significant issue compared to collective communication across the entire MPI processes when $N_z$ is relatively small ($N_z < O(100)$).

L245: Another argument for a 2D decomp for the SDM is that particle sedimentation in z direction does not lead to MPI communications.

We have revised L247 as follows:

In addition, variations in the computation amount and data movement depend on whether clouds and precipitation shaft are within the domain.

**3.2 Initialization of super-droplets**

L281: Could you make it clearer, why constant multiplicity is more natural? I can only understand that in the limit of multiplicity -> 1. It is important as it serves as motivation to develop the new init scheme.

When the number of SDs is smaller than the number of actual droplets but does not considerably deviate from it, the multiplicity of some droplets could become less than 1, depending on the PDF for sampling. In this case, it is reasonable to impose a constraint on the number so that the dynamic range of multiplicity is smaller (i.e., more similar to constant multiplicity) and multiplicity for all SDs is larger than 1.

In the revised manuscript, we have stated the following:

If we sample a vast number of SDs and if the number of samples becomes close to the actual number of droplets, imposing a constraint on the number is reasonable so that the dynamic range of multiplicity will be small (i.e., more similar to constant multiplicity) and the multiplicity for all SDs will be larger than 1.

L285: This question is not completely answered in L825 ff. Add in section 6?

Please refer to L1103 (original manuscript): the sensitivity of the microphysical variability and precipitation to initialization parameter $\alpha$ should be further explored by high-resolution simulations.

L362-363: cache blocking and moving branches out of loops was already the way to go for vector computers, maybe citations would be appropriate.

We have added the reference Lam et al. (1991) for cache blocking.

**3.3.2**

L385 Important point, could you motivate the consistency of the densities a bit further. L155 states a turbulence scheme is used for tracers but not for the SDM. see comment to L155

We have added the following sentences to the revised manuscript:

If consistency between the SD number density and air density is maintained, more SDs can be placed at location where clouds are more likely to occur. This not only requires placing increased number of SDs so that the SD number density is proportional to air density, but it also requires designing the SD movement scheme so that the time evolution of the SD number density follows the changes in air density. We focus on such schemes for grid-scale motion since the effect of subgrid motion should be relatively small. Because the air density decreases by the divergence of the velocity fields, the interpolation of the velocity should be developed to provide the divergence at the position of SDs that are calculated from interpolated velocity equaled to divergence at the cell. For such a scheme, a reduction in the variability of the SD number density is also expected since the divergence at the SDs does not differ within a cell. In addition, the numerical accuracy of interpolation should be increased to incorporate the effect of vortical and shear flows within a cell.

**3.3.5**

L520 citations seem to be appropriated

We have added the reference Decyk and Singh (2014) (L520) for PIC sorting, although their method is slightly different from ours.

L527-534 This is not obvious to me, maybe a sketch would help?

Data hierarchy within each MPI process and an example of one-dimensional SD sorting are shown in Figure 7. In the revised manuscript, we have added Figure 7 and cited it in L527-534.

[Figure]

**Figure 7.** Data hierarchy (particle & cell, block, group) in each MPI process and algorithm for SD sorting toward $x$-direction in each block. In the example, an MPI process has 4 groups, a group has 4 blocks, and a block has $4 \times 4$ cells. Using a list ($\{\}$) that stores the indices, SD sorting completes by copying in the SDs moving to adjacent blocks and copying back the SDs moving into the block. The number of total SDs within a block is monitored by counting only the moving SDs.

**4.1**

eq. 10 this metric seems to measure the usage of the available computer power, but ignores convergence, i.e. how many

tracers/bins/SDs are actually needed.

Thank you for highlighting this. We agree that it is important to include some convergence aspects in the discussion. However, it would be difficult to incorporate such into the metric. First, convergence properties can depend on the variable to be checked, for example, liquid water content, cloud droplet number concentration, and precipitation. Second, they can depend on the cloud form, amount of CCN, PDF used for initialization, and random number properties (pseudorandom number or low-discrepancy sequence). Thus, in this study, we did not mention the convergence aspect, except that 128/cell SDs are necessary for SCMS case, which was confirmed in our previous study (Matsushima et al. 2021) and other studies (Xue et al. 2022). In the revised manuscript, we have stated the limitations of the metric and cautions to be taken:

We do not incorporate the number of SDs that are actually needed to obtain converged solutions into the metric in Eq. (10) for avoiding loss of generality, and we will separately discuss the SDs number. For example, the convergence properties can depend on the variable to be checked, including liquid water content, cloud droplet number concentration, and precipitation. They can also depend on the setup and the results of simulations, such as cloud form, amount of CCN, PDF used for initialization, and random number properties.

L582 A short discussion on how much optimization might be possible might be appropriate and link to the discussion in section 5 L953ff.

We have added the following sentences to the revised manuscript:

General optimization has been applied to Seiki and Nakajima (2014) scheme. In this scheme, SIMD instructions vectorized the innermost loop for the vertical grid index, performing complex calculations on each water substance. The innermost calculations are divided by separate loops to improve computational performance using cache. However, there may still be room to find optimal loop fission and reordering calculations to reduce the latency of operations. In terms of computational cost, optimization is applied to Suzuki et al (2010) scheme. However, the innermost loops for bins are not vectorized for a small number of iterations.

L583 This comparison seems indeed unfair, especially as you emphasize in several places that SD-new128 has similar performance (see general comment above)

We have compared the physical performance of SDM with that of the two-moment bulk scheme for only warm microphysics in the revised manuscript and deleted L583. Nonetheless, we would like to retain the results of the two-moment bulk scheme with mixed-phase microphysics for discussion in Section 6.

**4.2**

L625 Why has BIN a O(N**2) scaling? That implies that advection not dominant. If the reason is collisions, a linear sampling might also be possible. Hence, degree of optimization is hard to judge.

Thank you for highlighting this. For the bin method, the computational complexity is $O(N^2)$ for $N$ bins when all possible combinations of collision are considered ($_N C_2$). A collision method similar to SDM may be applied. Sato et al. (2009) proposed reducing collision combinations by applying Monte Carlo integration inspired by SDM. As the method is implemented in the bin method (Suzuki et al. 2010), which we have used in our study, we performed additional warm bubble experiments with the option, and the results are shown in Figures 1 and 2 (BIN stoch.). We used the number of combinations of collision ($M$), $M = 16, 16, 16, 32, 128, 1024$ and $4096$ for the number of bins $N = 8, 16, 32, 64, 128, 256,$ and $512$, respectively. The elapsed time of the bin method was reduced by using this option. To reduce the order of the computational complexity to $O(N)$, $M \sim N/2$ can be used. However, when we set the number of bins $N \geq 128$ and used $M \propto N$, the computations were terminated by large negative values of liquid water that cannot be compensated by filters. This is consistent with the previous study (Sato et al. 2009), which stated that $M \geq 0.056_N C_2$ should be used to avoid large negative values of liquid water. Thus, computational complexity has not been successfully reduced. When this option can be stably used in the future, the elapsed time of the bin method becomes comparable to or faster than that of SDM-orig. In such situations, however, SDM would have advantages for considering multidimensional attributes to study collision–coalescence, as discussed by Shima et al. (2019). The above discussion has been provided in the revised

manuscript (see L701 in the marked-up manuscript).

**4.3**

L761 It do not understand the last sentence.

For clarity, we have revised the sentence as follows:

When the number of available nodes is limited, simulations using the two-moment bulk method can be performed with more grids.

L783 what do you mean by "only SDs in one cell"?

We have revised the sentence as follows:

To enable intercomparison of models for the readers, each microphysical variable of a cell was calculated by taking statistics for SDs (only) within the cell.

L814 it is unfortunate that the bulk scheme seems to have a bug or inappropriate threshold. That weakens the conclusion in L823 and L1165 considerably.

Thank you for allowing us to reconsider this problem. The strange pattern is attributed to the fact that the bulk method cannot effectively solve evaporation by entrainment.

In Seiki and Nakajima (2014) scheme, the change of cloud water number density by evaporation was modeled such that it does not decrease unless the mean mass of cloud water ($xc \equiv 4/3\pi\rho_w R^3$, here $\rho_w$ is water density, and $R$ is droplet radius) falls below the threshold $xc_{ccn}$. If it occurs, the cloud droplet number density completely evaporates (although smoothing is introduced). Mixing by entrainment always occurs homogeneously in this model. In the default setting, the threshold was very large ($10^{-12}$). This corresponds to a droplet radius of approximately $6\,\mu m$ (CFADs are shown in the upper panel of Figure 8). If we reduce the threshold to a reasonably small value ($4.2 \times 10^{-15}$, approximately $1\,\mu m$), the strange pattern becomes less noticeable (middle panel in Figure 8).

However, as we can assume that the real mixing can be homogeneous and inhomogeneous, the assumption of homogeneous mixing may be too simple. To incorporate the mixing scenario, we introduced the subgrid evaporation model (Morrison and Grabowski 2008, Jarecka et al. 2013), which considers a decrease in the cloud droplet number and water content, depending on the entrainment-mixing scenario. Jarecka et al. (2013) predicted a local mixing scenario calculated from the characteristic scale of cloud filaments and the fraction of a grid box occupied by cloudy air, which requires two new tracers. In this study, for simplicity and to avoid additional computational costs, we further assumed that the mixing scenario is fixed everywhere by a parameter $\alpha$ ($0 \leq \alpha \leq 1$, $\alpha = 0$ for homogeneous and $\alpha = 1$ for extreme inhomogeneous mixing) and evaporation is not delayed by subgrid-scale mixing. Based on the report of Jarecka et al. (2013), the parameters were set as $\alpha = 0.5$ for the pristine case (warm bubble, BOMEX) and $\alpha = 0.75$ for the polluted case (SCMS). The lower panel of Figure 8 shows the CFAD obtained using the Seiki and Nakajima (2014) scheme with subgrid-scale evaporation. CFAD slightly decreased from the original setup. The strange pattern at $R \sim 1\,\mu m$ observed in the original setup completely disappeared and became consistent with the CFAD obtained using SDM. In our future studies, we will investigate CFAD when predicting local mixing scenarios with additional tracers and a more reasonable assumption for delayed subgrid-scale mixing before evaporation.

For detailed changes we have made to the revised manuscript, see the sentences starting from L647, L678, L826, and L918 in the marked-up manuscript.

**5.2**

Table 3. tracking + condensation + coalescence only adds up to 141 min of 274 in microphysics. How much is due to the 3D/2D conversion + MPI wait steeming from the flow solver decomposition and how much from the technical workload needed for the microphyiscs itself like the SD sorting?

The measured times for 3D/2D decomposition and SD sorting are listed in Table 1. We have also added the following senteces to the revised manuscript:

[Figure]

**Figure 8.** Contoured frequency by altitude diagrams (CFADs) of cloud droplet number concentration (CDNC), LWC, and mean and standard deviation of the radius for SCMS experiments. Snapshots of BULK2MOM with (top row) $xc_{ccn} = 1.0 \times 10^{-12}, \alpha = 0$ and (middle row) $xc_{ccn} = 4.2 \times 10^{-15}, \alpha = 0$ and (bottom row) $xc_{ccn} = 4.2 \times 10^{-15}, \alpha = 0.75$ are shown.

However, it should be noted that the elapsed time may not be the time needed to process SD sorting and conversion of grid systems as load imbalances from other processes will likely affect it.

As precise measurements are unimportant for practical cases, such as BOMEX and SCMS, we prefer to avoid measuring them.

L953ff bin method O(N**2), tracer advection is not even dominant (see comment to L625

In the revised manuscript, we have stated the following sentences:

Finally, we compare the elapsed time using the SDM with the estimated elapsed time using the two-moment bulk and the bin methods, specifically focusing on the SD movement and tracer advection. These components are chosen because the elapsed time for the microphysics schemes depends on the computational algorithms and degree of optimization, as discussed in Sect. 4.1. However, the elapsed time for tracer advection is more robust in terms of optimization. Moreover, it can be one of the major computational bottlenecks and can be easily estimated.

L972 0.8*15*33 /= 363

The estimation in L972 is based on when a memory throughput of 80% against theoretical peak performance is achieved.

**Table 1.** Elapsed time, FLOPS (peak ratio of the FLOPS [%]), Peta instructions per second, memory throughput (peak ratio of the memory throughput [%]), and particle throughput (# of floating-point operations per SD)

| | Time [min] | Speed [PFLOPS] | PIPS | Memory Throughput [PB/s] | Part. Throughput [particle·step/s] |
|---|---|---|---|---|---|
| Time integration loop | 576 | 7.97 (7.04) | 1.86 | 13.7 (37.2) | |
| Dynamics | 290 | 8.55 (7.55) | 2.03 | 20.5 (55.7) | |
| Microphysics | 274 | 7.50 (6.62) | 1.69 | 6.25 (16.9) | $2.86 \times 10^{13}$ |
| Short time step | 238 | 9.50 (8.39) | 2.19 | 21.3 (57.9) | |
| Tracer time step | 15.0 | 5.85 (5.17) | 1.78 | 21.6 (58.7) | |
| Tracking | 87.9 | 15.3 (13.5) | 2.14 | 2.89 (10.5) | $8.91 \times 10^{13}$ (171) |
| Condensation | 32.6 | 18.2 (16.1) | 5.35 | 5.28 (14.3) | $2.40 \times 10^{14}$ (75.9) |
| Coalescence | 5.75 | 7.58 (6.69) | 2.96 | 17.5 (47.3) | $1.36 \times 10^{15}$ (5.57) |
| SD sorting | 79.2 | | | 12.5 (33.9) | |
| 3D to/from 2D conversion | 53.47 | | | | |

Thus, the elapsed time of tracer advection with 33 tracers was estimated to be 33*15*0.587/0.8 $\sim$ 363 min. We have modified L971 as follows:

Then, a possible optimization could refactor the codes, and we may improve the memory throughput performance of tracer advection to achieve up to 80% of the theoretical peak performance.

**6.1**

L990ff are too optimistic. First, instantaneous freezing and melting assumptions are quite restrictive, e.g. leads to complete neglect of wet growth in deep convection. Mixed-phase regimes -38 to 0 are very important and frequent, e.g. riming of ice particle is a main driver for cold phase precip. Hence, I think the coexistence of water and ice particles will be needed often and extension to mixed-phase particle should be the next step after Shima 09 and Shima 20. In conclusion, I feel that the comparison to mixed-phase bulk is not justified.#f

Thank you for your comment. We have deleted the paragraph starting from L990. In L979, we refer only to the case for temperatures above $0\,^{\circ}\mathrm{C}$ everywhere. In this paragraph, we retain the discussion on special cases in which the computational cost of mixed-phase SDM should be comparable to that of warm SDM if well implemented, whereas the computational cost of the mixed-phase two-moment bulk method increases for additional tracers. In L997 of the revised manuscript, we have provided an example of riming for a mixed-phase regime, based on your suggestion, and described the challenges of optimization.

For detailed changes we have made to the revised manuscript, see the paragraph starting from L1084 in the marked-up manuscript.

**6.3**

L1014ff Online postprocessing seems needed, for statistical analyses DSD on the scale of blocks should be sufficient. Of course the checkpoint/restart problem remains if the simulation time is longer than allowed by the compute queue. However, for such large scale applications it might be possible to discuss such rules with the computing center.

Thank you for your suggestion. Currently, statistical information about aerosols and clouds (number and water content) and activated SDs are outputted in a low-precision format. Analyses that need all information about SDs are possible only online. As you observed, we can make checkpoint/restart files in Fugaku, whose size is less than 400 TB, if we are permitted by the center. In the revised manuscript, we have mentioned that an online analysis should be performed if all information on SDs is necessary, and we may consult with the center for storing big data files. We would like to retain the discussion on data compression but in a simpler form.

For detailed changes we have made to the revised manuscript, see the sentences starting from L1033 in the marked-up manuscript.

**technical corrections**

L12: Add something like "possible" or "probably" for GPUs, as the effectiveness is not demonstrated in this paper.

Thank you for your suggestion. We have revised the sentence accordingly in the revised manuscript.

Fig 1. end of caption, dotted lines also in (d)

We have revised the statement in the revised manuscript.

L355 reformulate the sentence to avoid misinterpretation. E.g. move "except for the collision-coalescence process" to a new sentence or put it in parenthesis.

Thank you for your suggestion. We have revised the sentence accordingly.

L482 link to L375?

We have added as discussed in Sect. 3.3.1 in L482 to remind readers of L375.

L661 "resists" might not be the best word

We have revised the word to increases more gradually than linearly.

Fig 7. larger symbols in the legend

We have revised the symbols accordingly (see Figure 3 in this letter).

**References**

Decyk, V. K. and Singh, T. V.: Particle-in-cell algorithms for emerging computer architectures, Comput. Phys. Commun., 185, 708–719, https://doi.org/10.1016/j.cpc.2013.10.013, 2014.

Jarecka, D., Grabowski, W. W., Morrison, H., and Pawlowska, H.: Homogeneity of the subgrid-scale turbulent mixing in large-eddy simulation of shallow convection, J. Atmos. Sci., 70, 2751–2767, https://doi.org/10.1175/JAS-D-13-042.1, 2013.

Lam, M. D., Rothberg, E. E., and Wolf, M. E.: The cache performance and optimizations of blocked algorithms, Oper. Syst. Rev. (ACM), 25, 63–74, https://doi.org/10.1145/106973.106981, 1991.

Morrison, H. and Grabowski, W. W.: Modeling supersaturation and subgrid-scale mixing with two-moment bulk warm microphysics, J. Atmos. Sci., 65, 792–812, https://doi.org/10.1175/2007JAS2374.1, 2008.

Santos, S. P., Caldwell, P. M., and Bretherton, C. S.: Numerically relevant timescales in the MG2 microphysics model, J. Adv. Model. Earth Syst., 12, e2019MS001 972, https://doi.org/10.1029/2019MS001972, 2020.

Sato, Y., Nakajima, T., Suzuki, K., and Iguchi, T.: Application of a Monte Carlo integration method to collision and coagulation growth processes of hydrometeors in a bin-type model, J. Geophys. Res., 114, https://doi.org/10.1029/2008JD011247, 2009.

**Additional changes made by authors on article gmd-2023-26**

Toshiki Matsushima, Seiya Nishizawa, Shin-ichiro Shima

- We have revised our reference list to comply with the GMD standards.
- We have added the respective country names to the institutional affiliations.
- We have utilized an English editing service for the revision of our manuscript.

---

## Author Response (AR2)

**Reply to the comments of Dr. Simon Unterstrasser on article gmd-2023-26**

Toshiki Matsushima, Seiya Nishizawa, Shin-ichiro Shima

Thank you for taking the time to read through our manuscript and provide us with positive and helpful comments. Below are our point-by-point responses to your comments.

**1.)** From Fig. 6, I understand, that the SD spatial distribution is affected by the interpolation scheme, but what is the effect of it on the cloud as a whole. You could, e.g., add a panel to Fig. 7 showing the differences between CVI-1 and CVI-2. Other model adaptations should be described similarly to better understand the physical implications of your improvements.

You may even add an additional figure with time series of selected total cloud properties to make the comparison between the different model version more quantitative.

**Response:**

Thank you for your important question. During our research, we attempted to find how much the numerical accuracy of the CVI scheme affects the whole cloud system. In the experiments described in the manuscript, we could not find a clear difference in terms of LWC and the origins of the initial positions of droplets before the submission. To address your question, we investigated how the numerical accuracy of the CVI affects the time evolution of the tracer fields and how it might affect the clouds. In this letter, we would like to share the results with you.

We performed additional experiments to investigate the impact of the numerical accuracy of the CVI scheme on the time evolution of the tracer fields. These were modifications of the warm bubble experiments described in our manuscript. The domain size for these experiments is $300\,\mathrm{m} \times 8{,}000\,\mathrm{m} \times 5{,}000\,\mathrm{m}$ for $x$, $y$, and $z$ directions. The spatial resolution is $100\,\mathrm{m} \times 500\,\mathrm{m} \times 500\,\mathrm{m}$. We introduced a prescribed two-dimensional Benard-convection-like flow in which the cell positions oscillate periodically. The density and the mass stream function are represented as follows:

$$\rho_0 = 1\,[\mathrm{kg} \cdot \mathrm{m}^{-3}], \tag{1}$$

$$\psi(y, z, t) = \frac{50{,}000}{\pi} \sin\left(\frac{\pi z}{5{,}000\,\mathrm{m}}\right)\left[\sin\left(\frac{2\pi y}{8{,}000\,\mathrm{m}}\right) + \epsilon \cos\left(\frac{2\pi y}{8{,}000\,\mathrm{m}}\right)\cos\left(\frac{2\pi t}{1{,}800\,\mathrm{s}}\right)\right]\,[\mathrm{kg} \cdot \mathrm{m}^{-1}\mathrm{s}^{-1}], \tag{2}$$

$$\rho_0 v(y, z, t) = -\frac{\partial \psi}{\partial z}\,[\mathrm{m} \cdot \mathrm{s}^{-1}], \tag{3}$$

$$\rho_0 w(y, z, t) = \frac{\partial \psi}{\partial y}\,[\mathrm{m} \cdot \mathrm{s}^{-1}]. \tag{4}$$

Here, $\epsilon = 0.5$. Even for such a simple 2D flow, the time evolution of particles can be chaotic because the flow is unsteady (Malhotra et al. 1998). The momentum fields and the mass stream function at the initial time ($t = 0\,\mathrm{min}$) are shown in Fig. 1. The vorticity of the flow is given by

$$\zeta = -\left[\left(\frac{\pi}{5{,}000\,\mathrm{m}}\right)^2 + \left(\frac{2\pi}{8{,}000\,\mathrm{m}}\right)^2\right]\psi(y, z, t). \tag{5}$$

As described in our manuscript, the CVI-1 cannot incorporate the effect of shear and vortical flow within a cell to the SD movement, but CVI-2 can. Thus, the prescribed flow serves as a case in which the effects of the numerical accuracy in the CVI are likely to be verified.

First, we investigate the time evolutions of the particle distributions. An average of 3,200 particles per cell (128 for $100\,\mathrm{m}^3$ volume) was initially distributed in 3D space quasi-uniformly (Sobol sequences). The time integrations of particle positions using the CVI-1, CVI-2, and analytical expression of velocities were performed for $21{,}600\,\mathrm{s}$ by $\Delta t = 1.0\,\mathrm{s}$. The time evolutions of particle distributions are shown in Fig. 2. When the CVI-1 is used, a staircase-like pattern appears, and the particles mix

[Figure]

**Figure 1.** The momentum fields and the mass stream function at the initial time.

well, especially at the central positions of the vorticities centered around $y = 1,500\,\text{m}$ and $y = 5,500\,\text{m}$. In contrast, such patterns are not present in CVI-2 and ANL; Instead, small and thin filaments are depicted in CVI-2 and ANL. In our original manuscript, we could not compare the particle distributions obtained by CVI-1 and CVI-2 to the reference solution. However, our latest results confirm that switching from CVI-1 to CVI-2 improves particle distributions, aligning them more closely with the reference solution.

Now, we address how the numerical accuracy of particle distributions affects that of the tracer distributions (i.e., the statistics over the grid fields). In our calculations, we can track each particle's ID (or initial position of particles), enabling us to monitor the time evolutions of tracer fields for arbitrary initial tracer fields without the need to recalculate the time evolutions of the particle distributions. The initial conditions of the tracer field $\phi$ are defined as follows.

$$\phi(y, z, t = 0) = \sin\left(\frac{\pi k z}{5,000\,\text{m}}\right)\left[\sin\left(\frac{2\pi l y}{8,000\,\text{m}}\right) + \epsilon \cos\left(\frac{2\pi l y}{8,000\,\text{m}}\right)\right] + \sqrt{1 + \epsilon^2} \tag{6}$$

We construct the initial tracer fields by scaling and shifting the stream function described in Eq. (2) to ensure $\phi \geq 0$. Additionally, we introduce wavenumber parameters $(k, l)$ to control the spatial scale of the initial distributions. Instead of sampling particles to ensure consistency between the particle number density and tracer concentration, we sample numerous particles quasi-uniformly across the computational domain. We then assign a multiplicity to each particle, allowing us to manage tracer distribution time evolutions more efficiently. We examine the time evolution of the tracer fields for the initial distributions with the wavenumbers $(k, l) = (1, 1), (4, 3)$, and $(8, 5)$. The error is quantified as the $L^2$ norm of the difference between the tracer fields obtained using CVI-1 or CVI-2 and analytical velocities. These time evolutions of the errors are shown in Fig. 3. Note that the time evolutions of the tracer fields can be found in the appendix of this letter. In all cases, the errors when using CVI-2 are reduced compared to CVI-1, especially during the time $t < 180\,\text{min}$. The numerical accuracy of the CVI scheme has a more significant impact on the errors as the wavenumbers increase. The errors grow over time when $(k, l) = (1, 1)$, and the differences between CVI-1 and CVI-2 become statistically indistinguishable. However, CVI-2 still manages to reduce the errors even during extended simulated time, and the statistical differences between CVI-1 and CVI-2 remain apparent when $(k, l) = (8, 5)$.

These results provide insights into the effect of the numerical accuracy of the CVI scheme on cloud simulations.

- The particle distributions and transient time evolutions of the tracer fields are significantly improved in alignment with the reference solution when using the CVI-2 scheme instead of CVI-1.
- The impact of the CVI's numerical accuracy on errors is minimal for large-scale initial distributions. Since small-scale fluctuations of the DSD cannot be represented at the spatial resolutions often used in LES, it might be challenging to verify the CVI-2 scheme's impact on whole-cloud structures.
- In contrast, the effect of the CVI scheme's numerical accuracy is evident for small-scale tracer distributions. Although particle-based advection avoids numerical diffusion for each particle's information due to its algorithms, the CVI-1's effect might manifest as macroscopic and spurious numerical diffusion.
- If this occurs in meter-to-submeter scale cloud simulations, we might detect a statistical difference in the LWC between CVI-1 and CVI-2, as the LWC fluctuates due to small-scale turbulence mixing. This detection is analogous to how we discern the numerical accuracy of advection schemes through differences in effective resolution.
- The statistical difference in small-scale tracer fields might indirectly affect large-scale cloud fields through interactions between the eddies and microphysics.

[Figure]

**Figure 2.** Distributions of particle positions colored by the initial $y$ coordinate ($Y$) when CVI-1, CVI-2, and analytical velocities are used for particle movement.

In the revised manuscript, we will briefly outline the implications of our findings. We could expand our methodology to investigate further the impact of the CVI's numerical accuracy on clouds, using the warm bubble experiment as a basis. For instance, the CVI's numerical accuracy would be enhanced beyond 2nd-order spatial accuracy if we calculate the vector potential of the momentums using the spectral method. This would involve interpolating velocity at the particle position through the vector potential's interpolation, using the expansion of the basis function. Then we could compare the tracer distributions (moments of the DSD) obtained using CVI-1 and CVI-2 with a reference solution obtained using a spectral CVI. Unfortunately, implementing such a method in our model is not straightforward. Therefore, we intend to leave it to future work to provide a complete answer to your question. Additionally, we have decided to exclude the above discussion from the main body of the revised paper, as it is not directly related to the impacts on clouds. However, we will still provide this information in the supplemental material.

The differences in the time evolutions of the LWC when using different model versions and CVIs are provided in the Data set available at `https://doi.org/10.5281/zenodo.8103378`. In this Data set, the figures and data are public to readers.

[Figure]

**Figure 3.** The time evolutions of the errors between the tracer fields obtained using CVI-1 or CVI-2 and analytical velocities for each initial tracer distribution with the wavenumbers $(k, l) = (1, 1), (4, 3)$, and $(8, 5)$.

However, we have omitted these figures from the revised manuscript. The rationale for this decision is that the differences in clouds due to the numerical changes are trivially small. The slight differences detected are primarily attributable to sampling errors or the inherent chaos of the systems. We believe that such results do not offer meaningful insights for meter-to-submeter scale cloud simulations. Additionally, considering the already substantial length of the paper, we have deemed the priority of including these results in the manuscript to be low.

**2.)** Moreover, I understand that including SGS velocity needs a lot of memory (in the LCM of Sölch & Kärcher, SGS turbulence on SD movement is included. But clearly, those simulations do not run on such extremely large grids). But the open question is, if the inclusion of SGS velocity wouldn't be more important than switching from CVI-1 to CVI-2. How can you demonstrate that such sophisticated methods like CVI-2 are really necessary compared to neglecting GSG turbulence? Wouldn't SGS turbulence lead to more SD mixing and potentially affect your cloud evolution?

**Response:**

In the original manuscript, we mentioned why we have not adopted the SGS velocity into our model. However, we have modified the description based on feedback from the anonymous referee 2. We plan to implement the SGS velocity in the future.

Whether the impact of the numerical accuracy of the CVI or SGS velocity on clouds is more significant may depend on the spatial scale of the cloud simulations. We assume that for meter-to-submeter scale simulations such as this study and Mellado et al. (2018), the numerical accuracy of the advection schemes and the CVI may become relatively larger because the kinetic energy of the SGS turbulence (LES approach) or the viscous diffusion (DNS approach) becomes small. In the revised manuscript, we discuss the above points briefly.

We admit that the impact of CVI-2 may be less than that of the SGS velocity for low-resolution simulations. However, we would like to avoid stating which components are more important on clouds because our study is to improve the overall model performance but not determine the priority of the model development.

**3.)** You mention and stress that SD density should be similar to air density. This is a rather strong statement, which I believe is only valid under several assumptions and for a particular SD initialization (which may not be clear to readers).

The initial SD density may not be linked to air density, instead the multiplicities could be accordingly scaled. Moreover, one could initialize SDs only in a confined region of the whole domain. Unterstrasser & Sölch (2014) report about SD splitting and merging. In all those cases, SD density and air density are not linked any longer.

I understand that none of the cases is true in your case. Yet, your statements reads a bit like it is universally valid. This should be clarified to avoid any misunderstandings.

**Response:**

We agree with your comment and thank you for the important reference provided.

One desirable feature of the SD movement scheme is that the changes in the SD number density follow the changes in the air density. If we initialize SD positions in the computational domain so that the SD number density is proportional to the air density,

the consistency between the SD number density and air density is maintained throughout time evolution. This is expected to maintain adequate SD number density in convective clouds.

On the other hand, Unterstrasser and Sölch (2014) introduced SD splitting and merging, specifically to optimize collision–coalescence, a scheme that we understand performs a resampling of SDs. We may also consider introducing a resampling (such as used in particle filter) to maintain SD number density, even if SD positions are initialized independently of the air density.

Although these studies adopt different approaches, they share the purpose of maintaining the SD numbers in clouds. In our revised manuscript, we will first mention this purpose and outline the two approaches. We will then focus on the SD movement scheme that automatically adjusts the SD number density to air density, explaining the desired property of velocity interpolation for designing such schemes. Finally, we will discuss the necessity of numerical accuracy for SD movement in meter-to-submeter scale simulations.

In the attached document you can find many more comments on language, plot layout and so on.

**Response:**

Thank you for your many comments. We have revised the manuscript to conform to your comments as much as possible. Below are our responses for each.

1. Could you choose a less technical term in the abstract?
   We have changed it to throughput of particle calculations per second.
2. This implies that there is one particular state-of-the-art experiment. But I guess every researchers defines state-of-the- art slightly differently. You refer here to the sheer size of the model domain?
   We have changed it to highest resolution simulation performed so far.
3. The current formulation implies that you have developed more than one particle-based scheme. Is this intended? ” In particular, herein we focus on the particle-based super-droplet method (SDM) developed by Shima et al. (2009).” ?
   We have revised the sentence accordingly.
4. could you spell out FCT and provide a reference?
   flux corrected transport (FCT) scheme (Zalesak 1979)
5. In my opinion, you could keep the original version as the explanation was very clear.
   Thank you for your comment. We like the discussion on the convergence to DNS as $\xi \to 1$, which was added following feedback from anonymous referee 2. We would like to leave it as it is.
6. Is it some special notation to have subscripts on the left and right hand side of the letter? I do not know it. It looks peculiar so I believe the format has some special meaning.
   It is one of the notations to denote a number of combinations. We changed them to $C_2^N$.
7. the number or the length of the time steps?
   fixed
8. are
   fixed
9. that are
   fixed
10. does not vanish
    We have deleted it accordingly.
11. I do not understand why.
    To simplify the loop body for the SDs in a block, it is essential that the gridded values in a block are a collection of similar values, because similar operations or calculations may be applied to these values in such cases.
12. location
    We changed it to area.
13. the
    fixed

14. For this case, we can reduce the information per SD by subtracting information of partition by the MPI process and a block from the global position.

   We have added the explanation to the sentence. Note that amount of information for 2 possible states is $\log_2 2 = 1$ bit. Similarly, partitioning (or possible MPI process and a block for particles) introduce the information: the information that arises from the partitioning of the domain

15. The performance of your algorithm depends on your simulated clouds depending on which Case you encounter more frequently? You may state this clearly at the end of this subsection.

   The performance of our algorithms depends on mainly the number of Newton iterations (i.e., the closeness between the initial guess and solution) rather than whether Case 1 or Case 2 is more frequently encountered. We describe this in the revised manuscript.

16. `https://en.wikipedia.org/wiki/Rosenbrock_methods` ??

   fixed

17. described in Eq.

   fixed

18. the bulk methods solves individual droplets? Okay you write statistics, but the connection of bulk and individual is somehow counterintuitive.

   We have changed it to moments of the DSD.

19. groups

   We have changed it to moments of the DSD.

20. that of

   fixed

21. I am not sure if I understand this sentence. I understand that you deliberately included some additional modifications to the new SDM in order to achieve that the physical results remain similar to the old SDM.

   We made a modification from SCALE version 5.4.5 to generate an initial condition of water vapor mass mixing ratio that is similar to that generated by the original SCALE-SDM. This is just a bugfix for the SCALE version 5.4.5.

22. the

   fixed

23. in my understanding, x,y and z have units m. So please replace z by z/m. Or otherwise substract 1000m.

   We have added the units accordingly.

24. same applies

   fixed

25. statistical fluctuation

   The word, statistical fluctuation, is often used in our manuscript. We assume that what is lacking there is an additional explanation of how it is caused, so we have added the cause for the statistical fluctuation: caused by varying number of SDs in space.

26. are shorter

   fixed

27. SDM-new !!

   fixed

28. If it is easily changed you may turn the black background colour into white. Saves a lot of ink.

   When drawing the distributions of many tiny points, they are more clearly recognized for a black background. Please allow us to use a black background for the figure.

29. you may use white color for the smallest bin. I now see in the online version that there are thin white contour line. In my printed version they are not visible. The colours and the contour lines show both the same quantity?

   We have revised the figure to delete the thin white contour lines and to show the LWC higher than 0.001 g·m$^{-3}$.

30. I am not sure, if I understand correctly. You do not want to say "number of grid points"!? You really mean "number of grids". Do you define "grid" as the small subset of the local domain that fits into the cache? Sorry, if I have overlooked it.

We have changed them to the number of grid points.

31. please use BULK2MOM as in all other legends :-) Moreover, is it SDM-orig or SDM-new?

    fixed

32. N

    It is the values on the horizontal axis.

33. do you mean grid points?

    fixed

34. high as

    fixed

35. The font size in the figure is small. You may keep the x-axis description only in the bottom row. Similarly, it is enough to have the "z [km]" only at left-most position. As you use the same color bar in each columns you may move it to the top of each panel. These actions would all save space and allows to better increase the font size.

    We have revised the figures accordingly.

36. per grid

    We have added the sentence to describe more clearly: We represent the SD positions in a block by 50 bits using the mapping described in Eq. (6).

37. what does NH mean?

    node-hours

38. of 1h (In the beginning I was not sure what you mean with integration time; other also use the term "simulated time")

    We have changed them to simulated time.

39. Weak scaling performance of 2m

    We have changed the caption of the figure so that it is more proper to describe the figure: Elapsed times corresponding to the dynamics, microphysics, and the other processes for different number of nodes.

40. is included in a required physical process

    is included in the total elapsed time of the process that requires these conversions for calculations

41. this is only true if SD number is spatially uniform in the beginning, isnt it? would it be clearer to say that changes in the SD density should be consistent with changes in air density?

    We have revised them accordingly.

42. $(50m)^3$ ?

    fixed

43. thats not clear, do you refer to a particular Delta t VALUE?

    reference value of $\Delta t$ is defined prior to the sentence. So we have added the word larger or smaller to emphasize comparison with the prior sentence.

44. in the printed version the colour are hard to distinguish.

    We have changed the colors and line styles of the lines in the figure.

45. The caption should only list information that is needed to understand the plot. The marked sentences seem to list results.

    We have moved the results to the main body of the revised manuscript.

46. this formulation is quite generic. Could you state what feature of the cited paper you want to include. By the way the Sölch & Kärcher (2010) has a similar concept, as SDs only exist inside clouds.

    We have revised the sentence as follows: The computational performance of our numerical model may be further improved if we store only the activated cloud and rain droplets in memory, following methods such as those described in Sölch and Kärcher (2010) and Grabowski et al. (2018).

47. for those people reading only the conclusion, it would help to define the abbreviations again in this section. But thats up to you.

    We have changed them to include definitions of the abbreviations again.

48. You may leave out some information that is too technical and anyway no one can understand without reading the paper

    We have removed the explanation of Sinkhorn distances. We have also simplified the explanation of our newly developed initialization.

**References**

Malhotra, N., Mezić, I., and Wiggins, S.: Patchiness: A new diagnostic for Lagrangian trajectory analysis in time-dependent fluid flows, Int. J. Bifurc. Chaos Appl. Sci. Eng., 8, 1053–1093, https://doi.org/10.1103/RevModPhys.89.025007, 1998.

Sölch, I. and Kärcher, B.: A large-eddy model for cirrus clouds with explicit aerosol and ice microphysics and Lagrangian ice particle tracking, QJRMS, 136, 2074–2093, https://doi.org/10.1002/qj.689, 2010.

Unterstrasser, S. and Sölch, I.: Optimisation of the simulation particle number in a Lagrangian ice microphysical model, Geoscientific Model Development, 7, 695–709, https://doi.org/10.5194/gmd-7-695-2014, 2014.

Zalesak, S. T.: Fully multidimensional flux-corrected transport algorithms for fluids, J. Comput. Phys., 31, 335–362, https://doi.org/10.1016/0021-9991(79)90051-2, 1979.

**Appendix: time evolutions of the tracer distributions and the $L^2$ norm errors**

[Figure]

**Figure 4.** The case for initial tracer distributions with wavenumbers $(k, l) = (1, 1)$

[Figure]

**Figure 5.** The case for initial tracer distributions with wavenumbers $(k, l) = (4, 3)$

[Figure]

**Figure 6.** The case for initial tracer distributions with wavenumbers $(k, l) = (8, 5)$

**Additional changes made by authors on article gmd-2023-26**

Toshiki Matsushima, Seiya Nishizawa, Shin-ichiro Shima

- We have utilized an English editing service for the revision of our manuscript.

---

## Editor Decision (ED2)

[revised manuscript text omitted]

---

## Author Response (AR3)

**Reply to the comments of Dr. Simon Unterstrasser on article gmd-2023-26**

Toshiki Matsushima, Seiya Nishizawa, Shin-ichiro Shima

1. In the Supplement on the first page, last paragraph: 128 for $100^3$ m$^3$ volume. You missed the exponent 3 after 100.
2. Rosenbrock
3. quantities

**Response:**

Thank you for pointing out the typos. We have fixed them accordingly. Additionally, in the supplemental material, we removed the title page to meet GMD standards, changed the numbering for sections, equations, and figures, revised Fig. S2 for readers with colour vision deficiencies, and fixed the DOI in the references.